# It's `TIME`: Towards the Next Generation of Time Series Forecasting Benchmarks

**Zhongzheng Qiao** [1 2]  **Sheng Pan** [3]  **Anni Wang**  **Viktoriya Zhukova** [4]  **Yong Liu** [5]  **Xudong Jiang** [1]
**Qingsong Wen** [6]  **Mingsheng Long** [5]  **Ming Jin** [3]  **Chenghao Liu** [4 7]

## Abstract

Time series foundation models (TSFMs) are revolutionizing the forecasting landscape from specific dataset modeling to generalizable task evaluation. However, we contend that existing benchmarks exhibit common limitations in four dimensions: constrained data composition dominated by reused legacy sources, compromised data integrity lacking rigorous quality assurance, misaligned task formulations detached from real-world contexts, and rigid analysis perspectives that obscure generalizable insights. To bridge these gaps, we introduce `TIME`, a next-generation task-centric benchmark comprising 50 fresh datasets and 98 forecasting tasks, tailored for strict zero-shot TSFM evaluation free from data leakage. Integrating large language models and human expertise, we establish a human-in-the-loop benchmark construction pipeline to ensure high data integrity and redefine task formulation by aligning forecasting configurations with real-world operational requirements and variate predictability. Furthermore, we propose a novel pattern-level evaluation perspective that moves beyond traditional dataset-level evaluations based on static meta labels. By leveraging structural time series features to characterize intrinsic temporal properties, this approach offers generalizable insights into model capabilities across diverse patterns. We evaluate 12 TSFMs and establish a multi-granular leaderboard to facilitate in-depth analysis and visualized inspection. The leaderboard is available at https://huggingface.co/spaces/Real-TSF/TIME-leaderboard.

---

[1]Nanyang Technological University [2]CNRS@CREATE, Singapore [3]Griffith University, Australia [4]Datadog AI Research, Paris, France [5]Tsinghua University, Beijing, China [6]Squirrel Ai Learning [7]This work was completed prior to joining Datadog. Correspondence to: Zhongzheng Qiao <qiao0020@e.ntu.edu.sg>, Ming Jin <mingjinedu@gmail.com>, Chenghao Liu <twinsken@gmail.com>.

*Proceedings of the 43rd International Conference on Machine Learning*, Seoul, South Korea. PMLR 306, 2026. Copyright 2026 by the author(s).

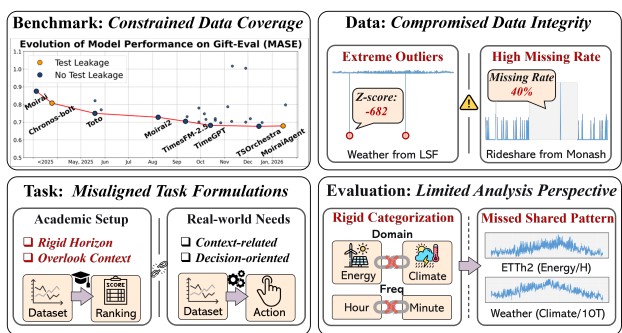

*Figure 1.* Common bottlenecks in prevalent TSF benchmarks.

## 1. Introduction

Time series foundation models (TSFMs) have reshaped the evaluation landscape of time series forecasting (TSF), shifting the dominant paradigm from a *dataset-centric regime*, where models are trained and assessed within individual datasets, to a *task-centric regime* that emphasizes zero-shot generalization across forecasting tasks. This paradigm shift has also exposed fundamental insufficiencies in widely adopted TSF benchmarks(Zhou et al., 2021; Wu et al., 2021). Recent studies (Hewamalage et al., 2023; Bergmeir, 2024; Brigato et al., 2026) have revealed critical limitations, including questionable data forecastability and misaligned evaluation protocols, whose effects can be obscured in task-centric benchmarking settings that evaluate models across a large number of heterogeneous datasets. In response, recent years have witnessed intensified efforts (Aksu et al., 2024; Cohen et al., 2025; Shchur et al., 2025) to develop more advanced TSF benchmarks that aim to move beyond traditional dataset-centric evaluation.

While these initiatives have introduced valuable refinements, a central challenge remains: *how can TSF benchmarking be fundamentally reoriented toward the task-centric evaluation paradigm required by TSFMs?* We argue that addressing this challenge requires confronting a set of persistent structural limitations in existing TSF benchmarks. As shown in Figure 1, we identify four critical dimensions: **(1)** ***Legacy-Constrained Data Coverage***. The data composition of modern benchmarks remains heavily constrained by *legacy* datasets (see Figure 2). This limited coverage is reflected in plateaued performance where model improvements on established benchmarks have visibly slowed down

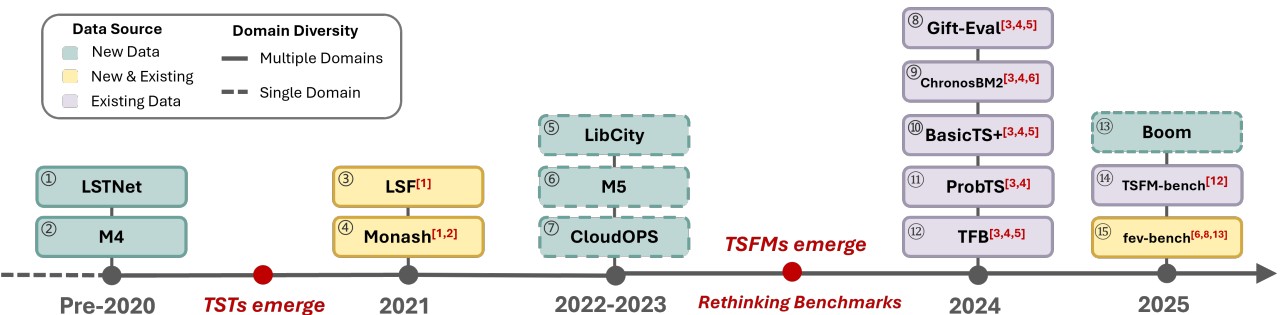

*Figure 2.* Timeline of time series forecasting (TSF) benchmarks. Each block represents a TSF benchmark, where color indicates the **data source** and outline style represents the **domain diversity**. Superscripts mark the prior benchmarks whose datasets are reused in each work. Major milestones and transitions in the field are highlighted in red along the timeline. TST denotes Time Series Transformer.

(e.g. Gift-Eval shown in Figure 1), while simultaneously amplifying the growing risk of benchmark contamination from data leakage.

**(2) *Compromised Data Integrity*.** Quality assurance procedures are often overlooked in dataset curation, including poor-quality variates with extreme outliers or excessive missing values (Wu et al., 2021; Godahewa et al., 2021).

**(3) *Misaligned Task Formulations*.** Classic benchmarks often use fixed prediction lengths, such as 720 steps, across diverse datasets. Such "one-size-fit-all" setups detach forecasting requirements from real-world contexts (Bergmeir, 2024), ignoring that the valid forecasting horizon depends on the specific application scenario and data frequency.

**(4) *Limited Analysis Perspective*.** Current evaluations categorize datasets by rigid meta labels (Aksu et al., 2024; Qiu et al., 2024) and aggregate dataset-level results as the performance indicator on the specific domain or frequency. While offering macro-level ranking, this perspective ignores that variates across domains/frequencies often exhibit similar temporal characteristics, resulting in limited insight into why a model performs well and offering weak guidance for selecting models in specific application scenarios.

Moreover, unlike modalities with intuitive performance proxies, time-series error metrics, such as MSE or MASE, only provide a scalar measurement of prediction error. A numerical improvement in these metrics does not guarantee that the model accurately captures temporal behaviors or that its predictions are reliable for actual deployment (See Section A.2 for details). For example, Figure 16 illustrates a prediction by TimesFM 2.5 (Das et al., 2024) on a test window. Although the metric appears favorable (MASE=0.662, MAE=1.12), the forecast fails to capture the distinct spike structure of the series. Thus, over-reliance on such numerical rankings risks decoupling benchmark conclusions from practical decision-making in real-world temporal settings.

To bridge these gaps, we introduce `TIME`, a next-generation task-centric TSF benchmark designed to support the evalua-

tion of modern TSFMs. Compared with predecessors, `TIME` advances the field in two fundamental dimensions: **benchmark construction** and **evaluation perspective**. Benchmark construction encompasses data curation and task formulation. We first curate a diverse collection of *fresh datasets* from public sources or industry partners, which have never/rarely been explored by existing TSF benchmarks. Secondly, to ensure data integrity, we establish a rigorous human-in-the-loop *data preparation pipeline*, which integrates automated screening with human decision to refine raw curated data into benchmark-ready quality. Critically, to ensure practical relevance, we formulate forecasting tasks by directly mapping configurations to the application context of each dataset. We leverage both domain-specific knowledge and LLM-based analysis to validate the rationality of each prediction task, ensuring that configurations (frequency and horizons) adhere to real-world operational requirements and variate predictability.

For evaluation perspective, we introduce a *pattern-level analysis* approach to facilitate generalizable and diagnostic benchmarking across heterogeneous datasets. Leveraging STL decomposition, we select a curated set of structural and interpretable time series features to characterize the intrinsic pattern of each variate. Through a binary encoding scheme, variates exhibiting identical pattern representations can be retrieved for a pattern-specific evaluation. For each TSFM, variate-level results are aggregated over retrieved variates, yielding high-level and generalizable insights into model capabilities on universal temporal behaviors.

Our contributions are summarized as follows:

- We introduce `TIME`, a task-centric benchmark comprising 50 fresh datasets and 98 forecasting tasks, where configurations are aligned with real-world operational requirements. By utilizing fresh data, `TIME` enables a strict zero-shot evaluation free from data leakage, ensuring a fair and unbiased assessment.

- We propose a pattern-level evaluation perspective for

generalizable, high-level, and cross-dataset performance benchmarking. By leveraging a curated set of interpretable temporal features with clear separability, `TIME` enables effective pattern-based stratification and retrieval, yielding generalizable and diagnostic insights into model performance.

- We evaluate 12 representative TSFMs on `TIME` and develop an interactive leaderboard supporting multi-granular analysis, combining quantitative results with qualitative visualization to improve the interpretability and actionability of benchmark outcomes. Code is available at https://github.com/zqiao11/TIME.

**Conflict of Interest Disclosure.** Two authors are employed by Datadog, the company that developed Toto (Cohen et al., 2025), which is one of the several models evaluated in this study. All benchmark evaluations were conducted objectively, and the remaining authors declare no financial or substantive conflicts of interest.

## 2. Related Work

### 2.1. Time Series Forecasting Benchmarks

We present the timeline of TSF benchmarks in Figure 2 . Early deep learning studies for TSF (Lai et al., 2018; Salinas et al., 2020) are not evaluated on standard benchmarks. Each work typically evaluates models on 4–6 datasets, which vary across studies and lack unified evaluation protocols.

A milestone is the M4 competition (Makridakis et al., 2020), which provides a large-scale univariate time series benchmark that enables evaluation of deep learning models such as MLPs and RNNs. With the rise of time series transformers (TST), LSF (Zhou et al., 2021; Wu et al., 2021) emerges as the first standardized benchmarks for long-horizon forecasting. Owing to its moderate scale and ease of use, LSF quickly becomes the most widely used in the TSF community. Concurrently, the Monash archive (Godahewa et al., 2021) introduced a diverse collection of datasets from various domains. Both LSF and Monash integrate existing and newly collected datasets, which serve as the primary data sources for many later TSF benchmarks. Afterwards, several domain-specific benchmarks are proposed, including M5 (Makridakis et al., 2022) for sales, CloudOPS (Woo et al., 2023) for cloud operation metrics, and LibCity (Wang et al., 2023) for urban transportation.

Since 2023, TSFMs have emerged as a prominent paradigm. Their rise drives the creation of large-scale benchmarks designed for zero-shot evaluation, characterized by diverse domains and extensive data coverage to assess universal forecasting capabilities (Aksu et al., 2024; Ansari et al., 2024; Li et al., 2025; Cohen et al., 2025; Shchur et al.,

2025). At the same time, the community reflects on the limitations of prevalent benchmarks (Hewamalage et al., 2023; Bergmeir, 2024) and calls for a new generation of TSF evaluation, leading to a new wave of benchmark development (Qiu et al., 2024; Shao et al., 2024; Zhang et al., 2024; Ni et al., 2025). However, these recent works still face main limitations. First, while they include a larger number of datasets, they mainly assemble previously released datasets instead of incorporating genuinely new data sources, failing to expand the boundaries of evaluation across novel and diverse temporal patterns. Second, both data integrity and task formulation are often compromised, characterized by unverified data flaws and mechanical setups detached from real-world contexts. Finally, their evaluation approaches remain conventional relying on coarse dataset-level metrics. While aggregating by metadata (e.g., domain) offers convenient categorization, it neglects intrinsic temporal patterns, failing to yield generalizable conclusions regarding model capabilities on universal dynamics.

### 2.2. Time Series Features

Unlike natural language or visual data, time series are inherently heterogeneous and lack intuitive semantics, making them difficult to interpret and categorize during analysis. Time series features (*tsfeatures*) are widely used to characterize the statistical and structural properties of time series. Early studies apply them primarily for classification tasks (Fulcher & Jones, 2014; Fulcher, 2018; Lubba et al., 2019) or synthetic data generation (Bahrpeyma et al., 2021). In the context of forecasting, a common practice (Kang et al., 2017; Spiliotis et al., 2020; Godahewa et al., 2021) selects several features and uses principal component analysis (PCA) to visualize the distribution of variates within a dataset. Recent benchmarks (Aksu et al., 2024; Qiu et al., 2024; Cohen et al., 2025) further employ tsfeatures to define several high-level properties (e.g., trend, seasonality) and analyze the overall coverage of their datasets. However, existing tsfeature-based analyses still remain at the dataset level, offering limited insights into pattern-specific model performance for variates across datasets.

## 3. Preliminary

**Time Series Forecasting Benchmarks.** A TSF benchmark serves as a unified platform for systematically evaluating forecasting methods. We formalize its internal structure as a hierarchy consisting of several levels: (1) A *benchmark* comprises multiple tasks that share a common evaluation protocol and provides multiple perspectives of analysis. (2) A *task* is defined by a specific dataset together with a dedicated prediction horizon. (3) A *dataset* contains one or more series, and the same underlying data sampled at different frequencies are treated as distinct datasets. (4) A *series*

can be *univariate* or *multivariate*; series within the same dataset may vary in temporal length but share the same set of variates. (5) A *variate* represents a single time-dependent variable, corresponding to one univariate time series (UTS) or a channel/variable within a multivariate series (MTS). In this benchmark, all variates are treated as prediction targets, excluding exogenous covariates. (6) A *testing window* denotes a continuous segment of a time series used as ground truth target for forecasting, typically with the length of prediction horizon.

**Forecasting Task.** We formulate a forecasting task as $\mathcal{T} = (\mathcal{D}, H)$. Here, $H$ denotes the prediction horizon, and $\mathcal{D} = \left\{ \mathbf{X}^{(i)} \right\}_{i=1}^{N}$ denotes a dataset comprising $N$ time series. Each series $\mathbf{X} \in \mathbb{R}^{L \times D}$ represents a complete temporal record of the sequence. Formally, the series is composed of $D$ individual variates, formulated as $\mathbf{X} = [\mathbf{x}_1, \mathbf{x}_2, \ldots, \mathbf{x}_D]$. Depending on the dimension, the series is categorized as univariate if $D = 1$ or multivariate if $D > 1$. For evaluation, we isolate the last $L_{test}$ time steps of each series as the test set. Forecasting samples are generated via a non-overlapping rolling window strategy: We employ a rolling window with a stride of $H$ over this test period, generating $W = \lfloor L_{test}/H \rfloor$ samples per series. For the $k$-th sample ($1 \leq k \leq W$), the benchmark defines the testing window as $\mathbf{X}_{t_k:t_k+H} \in \mathbb{R}^{H \times D}$, where the start index is $t_k = (k-1)H$ relative to the test set.

**Time Series Pattern.** A time series pattern denotes the intrinsic temporal characteristics of a single variate. By abstracting raw temporal data into these patterns, we establish a bridge for cross-dataset evaluation and enable generalizable, pattern-level analysis. For a given variate $\mathbf{x}$, we define its pattern representation as a feature vector $\boldsymbol{F} = (F_1, \ldots, F_K)$. This vector encodes a curated set of statistical properties that quantify the underlying temporal dynamics of the series. The detailed feature selection is provided in Section 5.

# 4. Benchmark Construction

## 4.1. Data Curation

**Data Sources.** Legacy datasets, many of which have been released for years, are likely to have been partially ingested into large-scale pre-training corpora. This exposure causes risks of both inadvertent and malicious contamination, undermining the reliability of benchmark-based evaluations. To mitigate the risk, we prioritize data novelty by curating fresh datasets. We aggregate data from four primary sources: official government portals, industrial and academic partnerships, open-access repositories, and forecasting competitions. Further details are provided in Section B.

**Manual Curation.** For each candidate dataset, we proceed with the following procedures: Firstly, we conduct an eligibility check covering both format and context. We verify that the data are *continuous* time series with valid timestamps and a regular sampling frequency, while simultaneously ensuring that its underlying application context supports well-defined forecastable processes. Subsequently, we perform metadata-based filtering by examining the frequency and time span of each series to ensure sufficient sequence length, discarding those that are too short relative to their frequency. We also review the semantic meaning of each variable, removing those deemed irrelevant to the application context. After that, we employ visualization-based inspection to identify and exclude variates lacking clear temporal patterns (e.g., constant sequences) or containing excessive missing values. In cases where quality is poor across the full span, we determine if a reliable shorter sub–time span can be extracted. Finally, the curated variates are organized into datasets following the structure defined in Section 3. Where appropriate, related variates are aggregated into multivariate series after aligning them to a common time span. Otherwise, they remain as multiple univariate series.

## 4.2. Automatic Screening

Given a raw dataset, we deploy an automated screening pipeline for fine-grained quality profiling. Instead of discarding series or variates immediately, this process generates a *Quality Summary* documenting the status of every series, proceeding through five sequential steps: *(1) Timestamp Rectification:* Curated data often exhibit missing or misaligned timestamps, which undermines temporal consistency. We first identify and fill missing timestamps, then rectify misaligned timestamps by calibrating them to the nearest standard timestamp based on the data frequency. *(2) Rule-based Validation:* As manual inspection is insufficient for fine-grained quality control at scale, we implement an automatic process to validate data against a set of predefined rules. We flag variates violating missing rate or length thresholds, and detect near-constant series lacking meaningful temporal dynamics by analyzing their value distributions. Specifically, a variate is flagged if its top-5 most frequent values constitute the majority of the observations or if its normalized entropy falls below a critical threshold. *(3) Statistical Test:* To eliminate unpredictable variates, we employ the Ljung-Box test (Ljung & Box, 1978) to detect white noise, which are flagged for removal. *(4) Extreme Outliers Removal:* We apply a local interquartile range (IQR) filter to detect and eliminate extreme erroneous values. For a time point $t$ within a local window $w$, a value is identified as an error if it falls outside $[m_w - k \cdot \text{IQR}_w, \ m_w + k \cdot \text{IQR}_w]$, where $m_w$ is the median and $\text{IQR}_w = Q_{0.75} - Q_{0.25}$. Detected errors are replaced by the preceding valid observation.

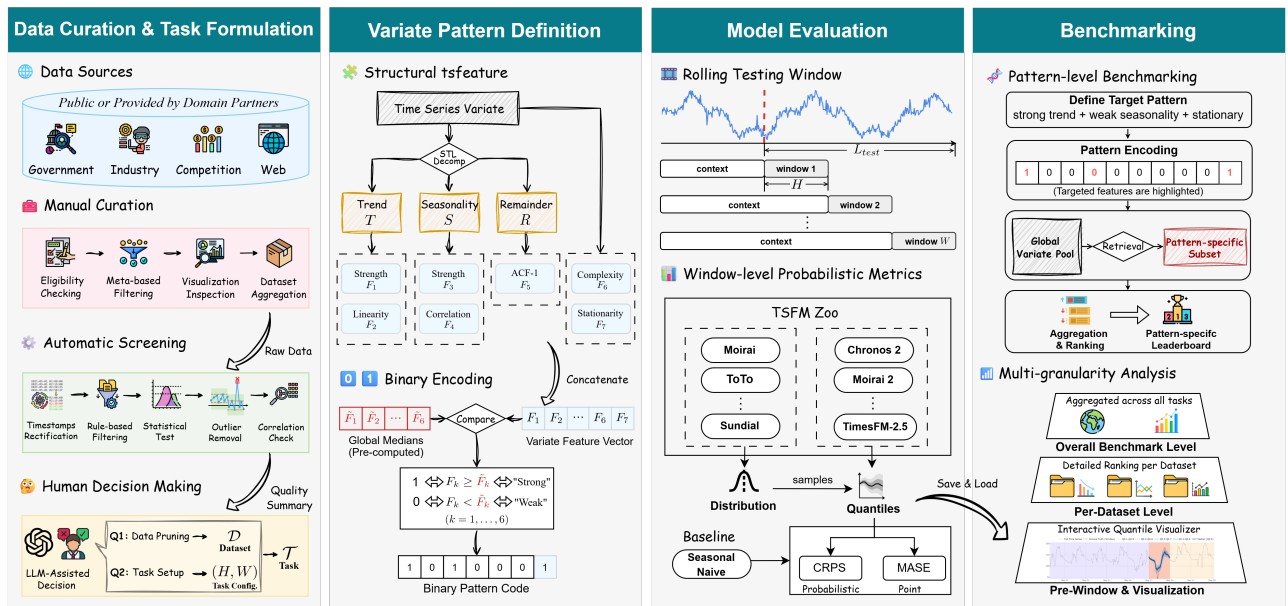

*Figure 3.* The overall workflow of TIME. (1) Fresh datasets are curated from diverse sources, integrating automated quality screening with LLM-assisted human decision-making to formulate context-aware forecasting tasks. (2) The intrinsic pattern of each variate is characterized by structural time series features derived via STL decomposition and mapped to binary encodings. (3) Representative TSFMs undergo rolling-window evaluations for probabilistic forecasting, where quantile-based metrics are computed and recorded for each window. (4) Performance analysis is conducted through pattern-targeted retrieval and multi-granular leaderboards.

We set $k = 9$ to preserve genuine spikes while filtering technical errors. *(5) Correlation Check:* Finally, we compute pairwise correlations across all variates within the dataset to detect potential redundancy. Pairs exhibiting a correlation coefficient exceeding a predefined threshold are flagged as highly collinear, marked for subsequent expert review.

After all these procedures (See Section C.1 for detailed algorithms), all diagnostic results are aggregated into a dataset-level Quality Summary, which is submitted for final human decision-making.

### 4.3. Human Decision Making

**Dataset Finalization via Review.**   Leveraging the Quality Summary, this stage introduces human judgment to finalize dataset structure, resolving quality ambiguities that automated protocols cannot address alone. Guided by domain knowledge and LLM-synthesized insights, we evaluate flagged variates within their specific application contexts to distinguish between genuine data corruption and expected domain characteristics. For instance, while high correlation among the availabilities of different car parks suggests redundancy warranting removal, similar correlations between macroeconomic indicators reflect inherent structural dependencies that must be preserved. This review dictates the precise granularity of data pruning: we determine whether to exclude specific problematic series (series-level) or eliminate entire variates (variate-level) that are fundamentally unsuitable for forecasting. This decision process ensures

each dataset $\mathcal{D}$ maintains high data integrity while remaining aligned with real-world application.

**Context-Aligned Task Formulation.**   Forecasting tasks should mirror real-world operational requirements. Therefore, task configurations must be directly determined by the specific application context. Instead of applying rigid settings across all datasets (e.g., universal horizons (Wu et al., 2021), frequency-proportional horizons (Aksu et al., 2024), or fixed split ratios (Qiu et al., 2024)), we formulate each task $\mathcal{T}$ based on domain expertise and LLM analysis. Specifically, we set prediction horizons $H$ according to the dataset's inherent frequency and operational constraints. We assign three horizons (*Short*, *Medium*, *Long*) for high-frequency datasets, and restrict datasets with low frequency or limited samples to a single operationally viable horizon. Similarly, we define the test length $L_{test}$ to cover complete seasonal cycles, ensuring the evaluation reflects practical deployment conditions. We detail the LLM usage and prompt templates in Section C.2.

## 5. Benchmarking Strategy

### 5.1. Structural tsfeatures for Pattern Definition

We curate a set of *structural* time series features to facilitate effective and interpretable categorization. For each variate $\mathbf{x} \in \mathbb{R}^L$, we decompose it into three additive components via STL decomposition (Seasonal and Trend decomposition

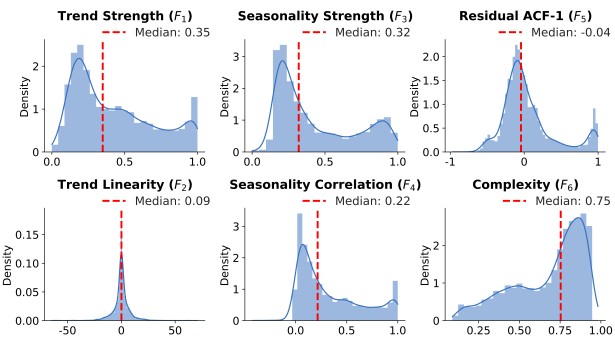

*Figure 4.* Empirical density distributions of our structural tsfeatures ($F_1$–$F_6$) across the benchmark variates. *Stationarity* ($F_7$) is omitted here as it is a binary indicator, with non-stationary variates accounting for 16.6% of the total.

using Loess) (Cleveland et al., 1990):

$$\mathbf{x} = T + S + R. \tag{1}$$

In contrast to previous benchmarks that compute metrics directly on raw variates $\mathbf{x}$ (Aksu et al., 2024; Qiu et al., 2024), our approach isolates distinct temporal components (Trend $T$, Seasonality $S$, and Remainder $R$). This results in features that are not only more interpretable and structured but also exhibit reduced overlap. We provide detailed definition of each tsfeature in Section D.2

For Trend $T$, we select two features to reflect its significance and structural patterns. *Trend Strength* ($F_1$) quantifies the proportion of the variate's variability that is explained by the trend component. *Trend Linearity* ($F_2$) capture the structural evolution of the trend, offering an intuitive metric to quantify the degree of its linear progression.

For Seasonality $S$, we similarly characterize its significance and structural patterns via two features. We compute *Seasonality Strength* ($F_3$) analogously to $F_1$ to quantify the seasonal contribution to the overall variability. *Seasonality Correlation* ($F_4$) measures the linear correlation between consecutive seasonal cycles, reflecting the stability and consistency of the seasonal evolution over time.

For Remainder $R$, we investigate whether any significant temporal structure persists. *Residual ACF-1* ($F_5$) capture residual dependencies by calculating the first-order autocorrelation of the remainder term.

Finally, we calculate two additional features on the raw variate $\mathbf{x}$ to characterize important global patterns. We include *Complexity* ($F_6$, spectral entropy of $\mathbf{x}$) to quantify the overall forecast difficulty, and *Stationarity* ($F_7$), a binary indicator derived from the Augmented Dickey-Fuller test (Dickey & Fuller, 1979) to distinguish between stationary ($p < 0.05$) and non-stationary processes.

To validate this feature set, we analyze the empirical probability distributions of these metrics across all variates in

| Model | Date | Arch. | Param. | Multi. | Output |
|---|---|---|---|---|---|
| TimesFM 2.5 | 10-25 | Dec. | 200M | ✗ | Q |
| Chronos-2 | 10-25 | Enc. | 120M | ✓ | Q |
| Kairos | 09-25 | E-D | 23M | ✗ | Q |
| Moirai 2.0 | 08-25 | Dec. | 11M | ✗ | Q |
| VisionTS++ | 08-25 | MAE | 460M | ✓ | Q |
| TiRex | 05-25 | xLSTM | 35M | ✗ | Q |
| Toto | 05-25 | Dec. | 151M | ✓ | D |
| Sundial | 05-25 | Dec. | 128M | ✗ | D |
| TimesFM 2.0 | 12-24 | Dec. | 500M | ✗ | Q |
| Chronos-bolt | 11-24 | E-D | 205M | ✗ | Q |
| Moirai | 06-24 | Enc. | 91M | ✓ | D |
| TimesFM 1.0 | 05-24 | Dec. | 200M | ✗ | Q |

*Table 1.* Properties of selected TSFMs. Dates are in MM-YY formats. Enc./Dec.= Encoder/Decoder-only, E-D=Encoder-Decoder; **Multi.**=Multivariate support; D=Distribution, Q=Quantiles.

our benchmark. As shown in Figure 4, these features exhibit diverse and informative distributions (See Section D.3 for quantitative analysis), enabling effective grouping and cross-domain analysis of time series patterns regardless of the underlying dataset or sampling frequency.

### 5.2. Pattern-Driven Benchmarking

Based on the structural features, we propose a pattern-driven strategy to stratify the benchmark and identify variates with shared characteristics for targeted evaluation. We employ a binary encoding scheme to transform the feature vector $\boldsymbol{F} = [F_1, \cdots, F_7]$ into a compact binary pattern code $\boldsymbol{B} \in \{0, 1\}^7$. Specifically, we first compute the feature vector $\boldsymbol{F}$ for every variate in the benchmark. For each continuous feature $F_k$ ($k = 1, \ldots, 6$), we calculate the population median $\tilde{F}_k$ across all the variates to serve as a global threshold. If the feature value of a variate exceeds this threshold ($F_k > \tilde{F}_k$), we consider the variate to possess a strong or dominant presence of that characteristic (encoded as 1). Otherwise, the characteristic is deemed weak or non-significant (encoded as 0). The binary feature *Stationarity* ($F_7$) is directly adopted without thresholding.

For each target pattern for evaluation, we can retrieve subsets of variates that exhibit identical binary codes from the whole benchmark. By using scale-invariant metrics (MASE and CRPS), we can aggregate the variate-level results for each TSFM to derive pattern-specific performance estimates. This granular evaluation allows us to generate distinct leaderboards for each pattern, revealing model capabilities across specific temporal dynamics.

## 6. Experiments

### 6.1. Global Setup

**Models and Datasets.** We evaluate 12 TSFMs with diverse architectural designs across 50 datasets and 98 tasks

(details in Table 2). The datasets span eight domains and feature a broad spectrum of frequencies. The evaluated models include: TimesFM (2.5/2.0/1.0) (Das et al., 2024), Moirai (2/1) (Liu et al., 2025a; Woo et al., 2024), Chronos (2/Bolt) (Ansari et al., 2025; 2024), Toto (Cohen et al., 2025), Sundial (Liu et al., 2025b), VisionTS++ (Shen et al., 2025), Kairos (Feng et al., 2025), and TiRex (Auer et al., 2025). Model properties are summarized in Table 1.

**Evaluation Protocol.** As described in Section 3, we adopt a rolling evaluation protocol (Aksu et al., 2024; Shchur et al., 2025). For metrics, we employ MASE and CRPS for point and probabilistic evaluation, respectively. To accommodate two distinct output formats of TSFMs (*distribution-based* and *quantile-based*), we derive quantiles via sampling for distribution-based models prior to metric computation, using a default sample size of 100. During evaluation, we save window-level metrics and quantile predictions per model, enabling the hierarchical analysis and visualization of the interactive leaderboard.

**Metric Normalization and Aggregation.** While MASE and CRPS are inherently scale-independent, evaluating TSFMs across diverse datasets presents challenges due to varying intrinsic forecastability. To ensure a robust and interpretable comparison, we adopt the protocol in (Aksu et al., 2024) and utilize a consistent relative evaluation framework where model metrics are normalized against a Seasonal Naive (S-Naive) baseline. This framework is applied across all experimental views, varying only in the granularity of the normalization unit.

For any given evaluation unit $u$ (e.g., a task or a variate), we compute the **normalized metric** as the ratio of the model's performance to the baseline:

$$\text{Metric}_{\text{model}}^{\text{norm}}(u) = \frac{\text{Metric}_{\text{model}}(u)}{\text{Metric}_{\text{S-Naive}}(u)}, \qquad (2)$$

where $\text{Metric}_{\text{model}}$ and $\text{Metric}_{\text{S-Naive}}$ are the raw metric values computed via arithmetic mean within unit $u$. A normalized score less than 1 indicates performance superior to S-Naive, while a value greater than 1 implies inferiority. We employ the **geometric mean** to aggregate these normalized metrics across units. This choice is critical for two reasons: (1) it offers greater robustness against outliers compared to the arithmetic mean, (2) it treats multiplicative relationships symmetrically (e.g., a model that is $2\times$ better on one task and $0.5\times$ on another yields a neutral mean of 1.0) and (3) it preserves the interpretability of the normalized scale.

### 6.2. Overall Performance

In this section, we assess the general capability of TSFMs by aggregating results at the **task level**, i.e, a unique pair of (dataset, horizon). The normalization unit here is the task

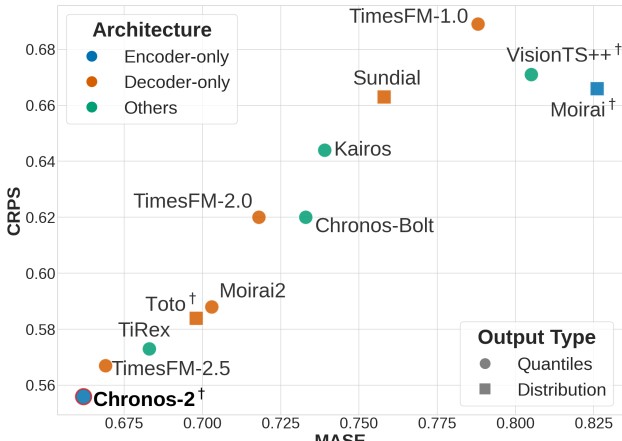

*Figure 5.* Overall performance across all tasks. Task-level results are normalized by the *Seasonal Naive* baseline and aggregated using the geometric mean. Lower values (bottom-left corner) indicate better performance. Models annotated with † support multivariate modeling, whereas others operate under channel independence.

itself: we first compute the raw metrics for a task (averaged across rolling windows, variates and series), normalize them by the corresponding S-Naive task score, and then aggregate these normalized scores across all 98 tasks using the geometric mean. Detailed numerical and Per Dataset results can be found in Table 4 and Table 6, respectively.

**Results.** Figure 5 illustrates the overall performance on point and probabilistic forecasting across all tasks. While using distinct architectures, Chronos-2, TimesFM-2.5, and TiRex emerge as the top-3 performers, consistently achieving the lowest MASE and CRPS scores across the benchmark. A distinct performance progression is observed where newer iterations (e.g., TimesFM-2.5 vs. 2.0/1.0, and Moirai2 vs. Moirai, Chronos-2 vs. Chronos-bolt) consistently outperform their predecessors. This confirms that recent advancements in TSFMs represent genuine capability improvements rather than overfitting to the data bias in established benchmarks. Analyzing the profile of the top-5 performing models reveals distinct design trends: decoder-only frameworks (TimesFM-2.5, Toto, Moirai2) and quantile-based output mechanisms are widely adopted. Notably, the leading positions are generally occupied by the most recently released models. This trend not only highlights the development of the TSFM field but also validates the rationality of our benchmark in accurately capturing these genuine advancements.

### 6.3. Pattern-specific Performance

Unlike the overall evaluation, pattern-driven analysis requires granular insights at the **variate level**. For a target pattern (e.g., high trend strength), we utilize our retrieval system to identify all variates matching this criterion from the benchmark. The performance for a specific pattern is

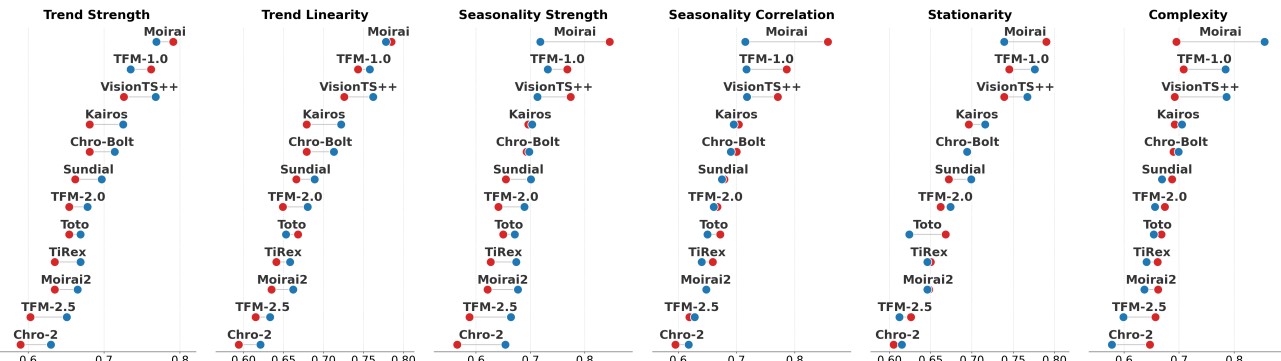

*Figure 6.* Comparison of **MASE** across different feature-specific variates. Each row represents a model's performance on variates with $F_k = 1$ (🔴) and $F_k = 0$ (🔵). The distance between dots indicates the performance difference of the model to that specific feature.

then computed as the geometric mean of the scaled metrics across all retrieved variates. In this section, we analyze six essential tsfeature independently. Figure 6 presents the Normalized MASE gap between time series exhibiting a specific feature ($F_k = 1$, red dots) and those that do not ($F_k = 0$, blue dots), whlie CRPS can be found in Figure 10. While joint patterns are possible, we focus on single-feature impacts to provide clear, actionable insights, encouraging readers to explore complex combinations via our leaderboard. Full numerical results are provided in Table 5.

**Trend.** The impact of trend attributes reveals a consistent performance profile. Regarding Trend Strength ($F_1$), for most models, we observe that the relative improvement over the S-Naive baseline is more pronounced on variates with stronger trends than those with weaker ones. (indicated by the red markers consistently outperforming the blue markers). The results for Trend Linearity ($F_2$) follow a similar pattern, where most models exhibit larger relative gains on sequences with high linearity. While TSFMs capitalize more effectively on strong and linear trending signals, the consistent performance enhancements observed across all trend-specific variates as models evolve suggest that the overall progress of TSFMs remains relatively independent of trend intensity or structure.

**Seasonality.** Regarding Seasonality Strength ($F_3$), we observe a disparity in model performance differentiation: On variates with weak seasonality, the performance gap between models is relatively narrow, where most TSFMs cluster around an MASE of 0.7. While TSFMs consistently provide a robust performance advantage over S-Naive, achieving significant further differentiation between models on such data remains challenging. In contrast, a significant divergence appears on variates exhibiting strong seasonality. The increased spread in performance on highly seasonal data suggests that a key advantage of superior models lies in their enhanced capacity to capture and model strong seasonal signals, whereas weaker models struggle to leverage

this information effectively.

In terms of Seasonality Correlation ($F_4$), the evolution of TSFMs brings performance gains across both stable and unstable seasonal patterns. For early/weaker models (Moirai, VisionTS++, and TimesFM-1.0), the modeling effectiveness on stable periods is relatively close to that of S-Naive, which results in a significant gap compared to the much larger gains seen on unstable sequences. With further development, this gap has narrowed significantly. Most recent TSFMs show a relatively small performance difference between stable and unstable sequences. For leading models like Chronos-2 and TimesFM-2.5, the performance gains on stable sequences actually surpass those on unstable ones. This suggests that recent TSFMs have become more robust to seasonality stability and provide consistent improvements.

**Stationarity.** For the majority of TSFMs, the relative gains are more pronounced on non-stationary sequences. This trend highlights the limitations of S-Naive, which struggle to adapt to the shifting statistical properties of non-stationary data. TSFMs, however, demonstrate a robust capacity to model these dynamic changes, resulting in a larger performance gap over the baseline in non-stationary scenarios. Notably, the model rankings exhibit high sensitivity to stationarity. For instance, Toto ranks third on non-stationary data but drops to sixth on stationary sequences. Conversely, Chronos-2 dominates stationary sequences as the top performer, but is surpassed by TimesFM-2.5 when dealing with non-stationary data. This shift suggests that different TSFMs possess unique inductive biases, with some optimized for handling non-stationary shifts and others prioritizing precision in stationary dynamics.

**Complexity.** Similar to $F_3$, we observe that the inter-model disparity is noticeably smaller on variates with high complexity compared to those with low complexity. This suggests that high spectral entropy acts as a performance equalizer, compressing the variance between models, whereas lower complexity allow superior models to more

clearly distinguish themselves from weaker counterparts. For weaker models, the relative gains on low-complexity data are smaller than on high-complexity sequences, indicating that their performance on simple signals shows no significant advantage over S-Naive. As models evolve, the relative improvement on low-complexity sequences increases significantly and eventually surpasses the gains seen on complex data. While high-complexity sequences remain difficult for all models, the most advanced TSFMs distinguish themselves by their superior ability to model and refine structured non-complex patterns.

**Summary.** Although model rankings exhibit specific variations across different pattern variates, the overall profile remains largely consistent. In most cases, Chronos-2 and TimesFM-2.5 maintain their status as the top two performers, while Moirai2, TiRex and Toto typically fluctuate within the third to fifth positions. Notably, we find that these feature-specific rankings do not align with the overall rankings presented in Section 6.2. While task-level aggregation is the common practice in existing benchmarks (Aksu et al., 2024; Shchur et al., 2025), our first attempt at variate-level aggregation reveals that rankings are significantly influenced by the level of aggregation. Determining the most appropriate aggregation level remains an open question, highlighting that a single leaderboard ranking is not the sole metric for success and requires more granular quantitative analysis.

### 6.4. Qualitatively Analysis

Beyond quantitative metrics, our platform enables qualitative analysis via prediction visualization. As shown in Section F, we observe that variates may exhibit distinct patterns in global and local views (Qiao et al., 2026). For instance, apparent global spikes can resolve into periodicities at a local scale. Additionally, variates often combine multiple structural patterns. Regarding model behavior, TSFMs generally predict clear seasonality and trends well. However, for series with high variations, TSFMs are prone to producing conservative predictions (i.e., relatively constant lines). In such cases, these flat forecasts may statistically yield competitive MASE or CRPS scores. However, quantitative metrics alone cannot distinguish whether the model has successfully captured the underlying dynamics or merely defaulted to a safe, conservative mean. This ambiguity highlights the necessity of visual verification to interpret the true nature of model behavior.

## 7. Conclusion

We introduce TIME, a task-centric benchmark designed to address the limitations in data freshness, integrity, and evaluation granularity plaguing current TSFM assessments. By implementing a rigorous data pipeline, we construct

a contamination-free repository of 50 fresh datasets with context-aligned task formulations. Our pattern-level evaluation perspective offers granular, generalizable insights into intrinsic model capabilities, revealing that TSFMs exhibit significant advantages over Seasonal Naive baseline in handling variates with non-stable seasonality or non-stationarity. Crucially, visualization serves as an essential complement, validating whether statistical predictive quality truly translates into actionable reliability for real-world applications. In future iterations, we plan to incorporate task-specific metrics tailored to different applications.

## Acknowledgements

This research is part of the programme DesCartes and is supported by the National Research Foundation, Prime Minister's Office, Singapore under its Campus for Research Excellence and Technological Enterprise (CREATE) programme. Zhongzheng Qiao is affiliated with Energy Research Institute @ NTU, Interdisciplinary Graduate Programme, Nanyang Technological University, Singapore.

## Impact Statement

This paper presents work whose goal is to advance the field of Machine Learning by introducing a real-world-aligned benchmark (TIME) for Time Series Foundation Models (TSFMs). There are many potential societal consequences of our work, none which we feel must be specifically highlighted as uniquely detrimental.

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

# A. Further Discussion of Our Motivation

## A.1. Plateaued Performance and Benchmark Contamination

Unlike natural language tasks, time-series forecasting lacks a human evaluation standard that can serve as an upper-bound reference. As a result, the validity of benchmarks can only be assessed indirectly through relative performance trends across models. In this context, as illustrated in the Figure 1, performance gains on current popular benchmarks have continued to slow down, making it increasingly difficult to disentangle genuine modeling progress from apparent improvements driven by the noise, implementation details, or data biases of evaluation benchmark.

A key factor contributing to this ambiguity is the growing risk of benchmark contamination in TSFMs. Most existing TSFM benchmarks rely on publicly available datasets, many of which have been released for years and are likely to have been partially absorbed into large-scale pretraining corpora. Given the demonstrated ability of foundation models to memorize training data (Carlini et al., 2022), both inadvertent and malicious contamination may occur, undermining the reliability of benchmark-based evaluation.

Notably, benchmark contamination in time series settings is arguably more difficult to detect and mitigate than in natural language. Time series data lacks intuitive semantic structure, making leakage hard to identify through manual inspection or standard similarity-based filtering. In addition, fragmented dataset versioning and inconsistent naming conventions often obscure shared data origins, leading to unintentional overlap across benchmarks. Finally, strong temporal dependencies imply that even datasets drawn from disjoint time periods may still exhibit implicit information leakage. Taken together, these factors suggest that contamination can silently inflate benchmark performance, further obscuring whether observed gains reflect genuine modeling advances or merely apparent improvements.

## A.2. Limited Evaluation Perspective

Benchmarks are not merely tools for ranking models, but are intended to serve as interpretable references that clarify how progress should be understood and acted upon. This role is particularly critical in time series forecasting, where commonly used error metrics, such as mean squared error, lack intuitive meaning in isolation. A numerical improvement on a benchmark offers little guidance on whether a model's predictions are reliable, robust, or suitable for real-world deployment, leaving a substantial gap between benchmark performance and practical decision-making.

This gap is further exacerbated by how existing general-purpose benchmarks are structured. Evaluation results are typically organized using predefined meta-categories, such as domain labels or dataset frequency, implicitly assuming that these categories align with forecasting task characteristics and model behavior. In practice, however, temporal patterns that govern predictability—such as trend strength, seasonality, and regime changes—often cut across domains, while datasets within the same domain can exhibit substantial heterogeneity. As a result, domain-level performance comparisons provide limited insight into why a model performs well and offer weak guidance for selecting models in concrete application scenarios.

Consequently, even when benchmark performance improves, it remains unclear whether such gains reflect meaningful advances in modeling capability or merely artifacts of evaluation design. This ambiguity is amplified by the inherent difficulty of mapping scalar error values to deployment readiness: unlike many natural language benchmarks, where metrics such as accuracy can be intuitively interpreted and sometimes anchored to human-level references, commonly used time-series error measures lack a clear operational meaning. Although recent benchmarks have proposed relative evaluation schemes, such as comparisons against naive baselines or win-rate statistics across models, these normalized metrics remain abstract and provide limited intuition about whether predictions are practically usable in real temporal settings. Without interpretability grounded in temporal behavior, benchmarks risk obscuring the relationship between measured performance and real-world usability.

To address this gap, we propose an interpretability-oriented benchmark analysis that complements aggregate metrics with pattern-aware aggregation and qualitative inspection. By extracting interpretable time-series features, such as trend and seasonality strength, and grouping test instances accordingly, our analysis enables more meaningful comparisons of model behavior across distinct temporal regimes. In addition, recognizing the inherent difficulty of mapping scalar error values to deployment readiness, we provide an interactive visualization interface that allows users to directly examine model forecasts at multiple temporal scales. While this analysis does not yield a definitive deployment criterion, it offers a transparent and intuitive lens for contextualizing benchmark results and assessing model behavior in realistic temporal settings.

*Table 2.* Individual statistics of tasks across all datasets. **Freq** denotes the sampling frequency: T (Minute), H (Hour), D (Day), B (Business day), W (Week), M (Month), and Q (Quarter). $H$ represents the forecast horizon length, and $W$ represents the number of rolling test windows. Values in parentheses indicate the actual physical time duration corresponding to the time steps (e.g., 1,440 (15D) denotes 1,440 steps spanning 15 days). Blank entries in the Medium/Long-term columns indicate that the dataset is evaluated under a single task configuration.

| Dataset | Freq | # Series | # Variate | # Obs | Series Length | | | | Short-term | | Med-term | | Long-term | | Domain | Source | License |
| | | | | | Avg | Min | Max | Test | H | W | H | W | H | W | | | |
|---|---|---|---|---|---|---|---|---|---|---|---|---|---|---|---|---|---|
| Water Quality-Darwin | 15T | 7 | 6 | 639,618 | 15,229 | 8,662 | 18,985 | 1,440 (15D) | 16 (4H) | 90 | 96 (D) | 15 | 288 (3D) | 5 | Nature | IMOS | CC BY 4.0 |
| Current Velocity | 5T | 1 | 6 | 158,916 | 26,486 | 26,486 | 26,486 | 4,320 (15D) | 36 (3H) | 120 | 288 (D) | 15 | 864 (3D) | 5 | Nature | IMOS | CC BY 4.0 |
| Current Velocity | 10T | 10 | 6 | 1,239,966 | 20,669 | 7,481 | 31,957 | 2,160 (15D) | 18 (3H) | 120 | 144 (D) | 15 | 432 (3D) | 5 | Nature | IMOS | CC BY 4.0 |
| Current Velocity | 15T | 5 | 6 | 255,096 | 8,503 | 3,710 | 17,953 | 1,440 (15D) | 12 (3H) | 120 | 96 (D) | 15 | 288 (3D) | 5 | Nature | IMOS | CC BY 4.0 |
| Current Velocity | 20T | 27 | 6 | 1,046,550 | 6,460 | 2,529 | 16,759 | 1,080 (15D) | 9 (3H) | 120 | 72 (D) | 15 | 216 (3D) | 5 | Nature | IMOS | CC BY 4.0 |
| Current Velocity | H | 21 | 6 | 441,288 | 3,502 | 2,208 | 5,587 | 672 (4W) | 24 (D) | 28 | 168 (W) | 4 | 336 (2W) | 2 | Nature | IMOS | CC BY 4.0 |
| CPHL | 15T | 2 | 1 | 20,752 | 10,831 | 9,956 | 10,394 | 1,440 (15D) | 12 (3H) | 120 | 96 (D) | 15 | 288 (3D) | 5 | Nature | IMOS | CC BY 4.0 |
| CPHL | 30T | 2 | 1 | 26,802 | 14,687 | 11,808 | 17,566 | 1,440 (30D) | 12 (3H) | 120 | 48 (D) | 30 | 144 (3D) | 10 | Nature | IMOS | CC BY 4.0 |
| CPHL | H | 4 | 1 | 19,191 | 4,971 | 2,490 | 8,783 | 672 (4W) | 24 (D) | 28 | 168 (W) | 4 | 336 (2W) | 2 | Nature | IMOS | CC BY 4.0 |
| Coastal T-S | 5T | 18 | 3 | 3,704,598 | 68,604 | 44,614 | 105,408 | 4,320 (15D) | 36 (3H) | 120 | 288 (D) | 15 | 864 (3D) | 5 | Nature | IMOS | CC BY 4.0 |
| Coastal T-S | 15T | 5 | 3 | 313,053 | 20,870 | 16,686 | 23,134 | 1,440 (15D) | 12 (3H) | 120 | 96 (D) | 15 | 288 (3D) | 5 | Nature | IMOS | CC BY 4.0 |
| Coastal T-S | 20T | 1 | 3 | 24,594 | 8,198 | 8,198 | 8,198 | 1,080 (15D) | 9 (3H) | 120 | 72 (D) | 15 | 216 (3D) | 5 | Nature | IMOS | CC BY 4.0 |
| Coastal T-S | H | 24 | 3 | 395,193 | 5,489 | 2,733 | 8,784 | 672 (4W) | 24 (D) | 28 | 168 (W) | 4 | 336 (2W) | 2 | Nature | IMOS | CC BY 4.0 |
| SG Weather | D | 6 | 4 | 73,488 | 2,953 | 2,953 | 2,953 | 366 (1Y) | 3 (3D) | 122 | 7 (W) | 53 | 14 (2W) | 27 | Nature | data.gov.sg | SG ODL 1.0 |
| SG PM 2.5 | H | 1 | 5 | 186,790 | 38,688 | 38,688 | 38,688 | 2,208 (3M) | 24 (D) | 92 | 72 (3D) | 30 | 168 (W) | 13 | Nature | data.gov.sg | SG ODL 1.0 |
| NE China Wind | H | 1 | 4 | 35,056 | 8,764 | 8,764 | 8,764 | 720 (30D) | 24 (D) | 30 | 72 (3D) | 10 | 168 (W) | 4 | Nature | Github | MIT |
| Australia Solar | H | 1 | 3 | 85,982 | 35,064 | 35,064 | 35,064 | 2,520 (15W) | 24 (D) | 105 | 72 (3D) | 35 | 168 (W) | 15 | Energy | Pvoutput | Terms of Service |
| EPF Electricity Price | H | 5 | 1 | 262,080 | 52,416 | 52,416 | 52,416 | 2,520 (15W) | 24 (D) | 105 | 72 (3D) | 35 | 168 (W) | 15 | Energy | Academic | Apache 2.0 |
| OpenElectricity NEM | 5T | 1 | 10 | 434,880 | 43,488 | 43,488 | 43,488 | 4,032 (2W) | 24 (2H) | 168 | 96 (8H) | 42 | 288 (D) | 14 | Energy | OpenElectricity | CC BY-NC 4.0 |
| EWELD Load | 15T | 1 | 10 | 205,440 | 20,544 | 20,544 | 20,544 | 1,344 (2W) | 24 (4H) | 56 | 96 (D) | 14 | 672 (W) | 2 | Energy | Academic | CC BY 4.0 |
| SG Carpark | 15T | 354 | 1 | 5,073,528 | 14,332 | 14,332 | 14,332 | 672 (1W) | 16 (4H) | 42 | 96 (D) | 7 | 672 (W) | 1 | Transportation | data.gov.sg | SG ODL 1.0 |
| Finland Traffic | 15T | 1 | 1 | 35,136 | 35,136 | 35,136 | 35,136 | 2,976 (1M) | 16 (4H) | 186 | 96 (D) | 31 | 672 (W) | 4 | Transportation | Digitraffic | CC BY 4.0 |
| Port Activity | D | 99 | 2 | 421,146 | 2,127 | 2,127 | 2,127 | 365 (A) | 30 (M) | 12 | | | | | Transportation | Competition | CC BY 4.0 |
| Port Activity | W | 99 | 2 | 60,192 | 304 | 304 | 304 | 52 (A) | 13 (Q) | 4 | | | | | Transportation | Competition | CC BY 4.0 |
| ECDC COVID | D | 9 | 1 | 9,681 | 1,117 | 794 | 1,229 | 150 (120D) | 30 (30D) | 5 | | | | | Healthcare | ECDC | CC BY 4.0 |
| ECDC COVID | W | 16 | 1 | 2,609 | 165 | 114 | 196 | 52 (52W) | 13 (Q) | 4 | | | | | Healthcare | ECDC | CC BY 4.0 |
| Global Influenza | W | 15 | 4 | 12,105 | 205 | 154 | 210 | 52 (52W) | 13 (Q) | 4 | | | | | Healthcare | WHO | Terms of Service |
| Crypto | D | 1 | 4 | 11,344 | 2,842 | 2,842 | 2,842 | 273 (9M) | 30 (M) | 9 | | | | | Finance | Public | Public Domain |
| US Term Structure | B | 1 | 40 | 357,400 | 9,327 | 9,327 | 9,327 | 717 (11Q) | 20 (4W) | 35 | | | | | Finance | U.S. Gov | Public Domain |
| Oil Price | B | 1 | 12 | 57,312 | 5,035 | 5,035 | 5,035 | 717 (11Q) | 20 (4W) | 35 | | | | | Finance | U.S. Gov | Public Domain |
| Job Claims | W | 1 | 2 | 392 | 196 | 196 | 196 | 52 (A) | 13 (Q) | 4 | | | | | Economics | U.S. Gov | Public Domain |
| Uncertainty-1M | M | 1 | 3 | 2,340 | 780 | 780 | 780 | 42 (42M) | 6 (6M) | 7 | | | | | Economics | Academic | Public Domain |
| Housing Inventory | M | 1 | 4 | 456 | 114 | 114 | 114 | 36 (3A) | 12 (A) | 3 | | | | | Economics | Realtor.com® | Terms of Service |
| JOLTS | M | 1 | 6 | 1,782 | 297 | 297 | 297 | 60 (5A) | 12 (A) | 5 | | | | | Economics | U.S. Gov | Public Domain |
| US Labor | M | 1 | 14 | 5,320 | 380 | 380 | 380 | 60 (5A) | 12 (A) | 5 | | | | | Economics | U.S. Gov | Public Domain |
| Vehicle Supply | M | 1 | 6 | 2,346 | 391 | 391 | 391 | 60 (5A) | 12 (A) | 5 | | | | | Economics | U.S. Gov | Public Domain |
| Auto Production-SF | M | 1 | 1 | 367 | 367 | 367 | 367 | 60 (5A) | 12 (A) | 5 | | | | | Economics | U.S. Gov | Public Domain |
| Commodity Production | M | 32 | 1 | 10,403 | 325 | 120 | 648 | 60 (5A) | 12 (A) | 5 | | | | | Economics | NBER | Public Use |
| Commodity Import | M | 8 | 1 | 5,578 | 697 | 254 | 1240 | 60 (5A) | 12 (A) | 5 | | | | | Economics | NBER | Public Use |
| WUI-Global | Q | 1 | 15 | 4,325 | 294 | 294 | 294 | 20 (5A) | 4 (A) | 5 | | | | | Economics | Public | CC BY 4.0 |
| Global Price | Q | 1 | 60 | 8,464 | 142 | 142 | 142 | 20 (5A) | 4 (A) | 5 | | | | | Economics | IMF | Terms of Service |
| Vehicle Sales | M | 1 | 10 | 5,960 | 596 | 596 | 596 | 60 (5A) | 12 | 5 | | | | | Sales | U.S. Gov | Public Domain |
| Online Retail II | D | 1 | 1 | 739 | 739 | 739 | 739 | 180 (6M) | 30 | 6 | | | | | Sales | Competition | CC BY 4.0 |
| Supply Chain-Customer | D | 1 | 36 | 72,252 | 2,007 | 2,007 | 2,007 | 365 (A) | 30 | 12 | | | | | Sales | Competition | Apache 2.0 |
| Supply Chain-Location | D | 1 | 51 | 102,357 | 2,007 | 2,007 | 2,007 | 365 (A) | 30 | 12 | | | | | Sales | Competition | Apache 2.0 |
| Azure2019-D | 5T | 989 | 3 | 25,596,372 | 8,627 | 8,073 | 8,639 | 864 (3D) | 288 (D) | 3 | | | | | CloudOPS | Github | CC BY 4.0 |
| Azure2019-I | 5T | 492 | 3 | 12,738,438 | 8,630 | 8,250 | 8,639 | 864 (3D) | 288 (D) | 3 | | | | | CloudOPS | Github | CC BY 4.0 |
| Azure2019-U | 5T | 78 | 3 | 329,019 | 1,406 | 882 | 2,006 | 288 (D) | 48 (4H) | 6 | | | | | CloudOPS | Github | CC BY 4.0 |
| Smart Manufacturing | H | 34 | 5 | 201,325 | 1,666 | 1,661 | 1,667 | 336 (2W) | 24 (D) | 14 | 168 (W) | 2 | 336 (2W) | 1 | Industry | Competition | CC0 1.0 |
| MetroPT-3 | 5T | 1 | 6 | 88,572 | 17,809 | 17,809 | 17,809 | 1,728 (6D) | 48 (4H) | 36 | 288 (D) | 6 | 576 (2D) | 3 | Industry | Competition | CC BY 4.0 |

# B. Datasets

## B.1. Nature

**Water Quality-Darwin**   This dataset consists of water quality measurements sourced from the National Reference Station (NRSDAR) located off the coast of Darwin, Northern Territory. It features multivariate time series from seven stations, downsampled to a 15-minute resolution. The dataset tracks six key water quality indicators: electrical conductivity (CNDC), temperature (TEMP), practical salinity (PSAL), chlorophyll mass concentration (CPHL), turbidity (TURB), and dissolved oxygen (DOX2). We ensured data integrity by filtering out bad-quality data points based on the provided quality flags. This dataset is publicly available under CC BY 4.0 License. We acknowledge the dataset as follows: Data was sourced from Australia's Integrated Marine Observing System (IMOS) - IMOS is enabled by the National Collaborative Research Infrastructure Strategy (NCRIS). The support of the Darwin Port Corporation is also acknowledged.

**Current Velocity**   This dataset consists of time-series observations of ocean current velocities sourced from the National Mooring Network Facility, designed to monitor oceanographic phenomena in Australian coastal waters. The data were collected using Acoustic Doppler Current Profilers (ADCP) hosted on moorings across New South Wales, the Northern Territory, Queensland, South Australia, and Western Australia. The observational period covers May 22, 2020, to August 16, 2024. We select the primary flow indicators including the zonal (UCUR), meridional (VCUR), and vertical (WCUR) current velocity components. Additionally, environmental parameters such as sea water temperature (TEMP), pressure (PRES), and sound speed (SSPD) are included to characterize the hydrographic conditions at the instrument depth. Data with different

minute-level sampling frequencies originate from distinct measurement stations. The hourly-level data were obtained by downsampling these minute-level records, filtering out sequences that were too short. This dataset is publicly available under CC BY 4.0 License.

**CPHL** This dataset monitors chlorophyll mass concentration across four marine stations in Australia. The data collection falls within the overall span of January 1, 2021 to January 1, 2023, though the specific temporal coverage varies for each station within this period. Regarding sampling frequency, two stations record at 15-minute intervals, while the remaining two operate at 30-minute intervals. The hourly-level dataset was derived by downsampling the minute-level records. This dataset is publicly available under CC BY 4.0 License.

**Coastal T-S** This dataset is derived from the National Mooring Network Facility, which maintains a series of reference stations for monitoring Australian coastal waters. We collected time-series observations of temperature and salinity from dozens of stations within this network. The dataset covers the period from January 1, 2023, to January 1, 2024, utilizing high-precision measurements from temperature loggers and Conductivity-Temperature-Depth (CTD) instruments to capture coastal oceanographic variations. Same as above, distinct minute-level frequencies originate from different stations, while hourly data are derived via downsampling. This dataset is publicly available under CC BY 4.0 License.

**SG Weather** This dataset contains information from "Realtime Weather Readings across Singapore", accessed on June 15, 2025 from data.gov.sg, which is made available under the terms of the Singapore Open Data Licence version 1.0. It include real-time daily weather readings collected from six monitoring stations across Singapore. We select four variates featuring temperature, humidity, wind direction, and wind speed. We curated the data recording from January 1, 2017, to January 31, 2025 via the official API. Observations from February 1, 2024, onwards are used as the test set.

**SG PM 2.5** This dataset contains information from "PM2.5", accessed on June 15, 2025 from data.gov.sg, which is made available under the terms of the Singapore Open Data Licence version 1.0. This dataset captures hourly concentrations of PM2.5 across five major geographical regions in Singapore: Central, East, North, South, and West. The observational data covers the period from January 1, 2021, to May 31, 2025, reflecting regional air quality dynamics. Data from the final year was utilized as the test set. We acknowledge the dataset as follows: National Environment Agency. (2024). PM2.5 (2025) [Dataset]. data.gov.sg. Retrieved June 15, 2025 from `https://data.gov.sg/datasets/d_e1058d6974c877257e32048ab128ad83/view`.

**NE China Wind** We utilize the meteorological recordings from (Zhang et al., 2025), covering Northeast China at an hourly frequency. Key variables include u-wind, v-wind, geopotential height, and temperature. While the original work (Zhang et al., 2025) utilized the latter two as auxiliary features, we treat all four variables as target series for forecasting. Due to data homogeneity, we only use data from a single region spanning one year. The dataset and its associated codebase are publicly available under the MIT License.

**B.2. Energy**

**Australia Solar** This dataset is curated from PVOutput[1] platform. It comprises hourly solar energy generation records from three distinct photovoltaic (PV) stations located in Australia. The temporal coverage spans exactly four years, from April 1, 2021, to March 31, 2025. We accessed this data via their Data Service API, in accordance with their respective Terms of Service.

**EPF Electricity Price** This dataset is sourced from (Lago et al., 2021) includes hourly electricity markets recording from five different areas, each of them comprising 6 years of data. In this benchmark, we discard all exogenous variables and exclusively retain the electricity price as the prediction target. The dataset and its associated toolbox are publicly available under the Apache License 2.0.

**OpenElectricity NEM** This dataset is curated from the OpenElectricity platform, a public repository tracking energy statistics across Australia. We focus specifically on the National Electricity Market (NEM), collecting electricity demand and price series from five distinct regions. The data covers the period from January 1, 2025, to May 31, 2025. The data is made available by The Superpower Institute under CC BY-NC 4.0 License.

---

[1] `https://pvoutput.org/`

**EWELD Load**    This dataset is sourced from (Liu et al., 2023), which includes 15-minute electricity consumption data from industrial and commercial users under extreme weather events. We select a subset of 10 users spanning from June 1, 2019, to December 31, 2019. We utilize only the univariate consumption series and exclude all weather covariates. The dataset is publicly available under CC BY 4.0 License.

### B.3. Transportation

**SG Carpark**    This dataset contains information from "Carpark Availability", accessed on June 15, 2025 from data.gov.sg, which is made available under the terms of the Singapore Open Data Licence version 1.0. We curated the data from website at a 15-minute resolution spanning from January 1, 2025, to June 1, 2025, to capture high-frequency urban parking utilization patterns. Following filtering and preprocessing, the final dataset includes 354 storey carparks. We use the last one week data as test set. We acknowegde the dataset as follow: Housing & Development Board. (2018). Carpark Availability (2025) [Dataset]. data.gov.sg. Retrieved June 15, 2025 from `https://data.gov.sg/datasets/d_ca933a644e55d34fe21f28b8052fac63/view`

**Finland Traffic**    This dataset comprises real-time traffic volume measurements sourced from Digitraffic platform. The data is acquired from fixed monitoring stations positioned throughout the Finnish road network. The observational period spans from January 1, 2024, to January 1, 2025. To capture fine-grained temporal dynamics, the raw traffic counts were aggregated into a 15-minute resolution, providing a granular representation of traffic flow intensity and congestion patterns at specific road junctions. The data is publicly available under the CC BY 4.0 License.

**Port Activity**    This dataset offers a high-resolution view of global maritime trade by tracking daily port activities and cargo estimates provided by the IMF. It encompasses diverse vessel types and trade volumes across multiple geographic regions. In addition to the original daily records, a weekly aggregated version is also generated for longitudinal analysis. The dataset is sourced from The World Bank and is publicly available under the CC BY 4.0 License, subject to The World Bank's specific terms of use.

### B.4. Health Care

**ECDC COVID**    This dataset captures the daily dynamics of healthcare utilization sourced from the European Centre for Disease Prevention and Control (ECDC). It features time-series observations of daily hospital occupancy, defined as the total number of COVID-19 patients occupying hospital beds on a given day. The dataset covers a comprehensive period from January 5, 2020, to November 26, 2023, across multiple European regions. The data is publicly available under the CC BY 4.0 License. Following the provider's mandate, we acknowledge the source as follows: "Dataset provided by ECDC based on data provided by public health authorities, scientific institutes or health care providers in the relevant reporting countries and/or by WHO." Please note that the original data has been adapted and preprocessed by us to fit the specific input requirements of our benchmark.

**Global Influenza**    This data is curated from FluNet, a global web-based platform for influenza virological surveillance, maintained by the WHO's GISRS network since 1997. It provides weekly-updated, country-level data on influenza subtypes. We select 15 representative countries and regions including the USA, UK, India, and Singapore for this dataset. The dataset is publicly made available by the World Health Organization (WHO). We utilize this anonymized data in accordance with the WHO Policy on the Use and Sharing of Data, which permits non-commercial, not-for-profit use. We formally acknowledge the source as: WHO, FluNet (Global Influenza Virological Surveillance), data provided by National Influenza Centres (NICs) of the Global Influenza Surveillance and Response System (GISRS).

### B.5. Finance

**Crypto**    This dataset comprises daily price records for four major cryptocurrencies—Bitcoin, Ethereum, Litecoin, and Bitcoin Cash—covering the period from December 20, 2017, to September 30, 2025. Data from the final nine months of 2025 are designated as the test set. Consistent with the methodology established by the Monash Time Series Forecasting Archive (Godahewa et al., 2021), the raw data is curated by extracting the historical daily series from the interactive web graphs publicly available at BitInfoCharts using an automated script.

**US Term Structure**   This dataset tracks daily term premiums and instantaneous forward rates derived from the Kim and Wright arbitrage-free model (Kim & Wright, 2005). The data spans from January 2, 1990, to September 30, 2025, on a business-day frequency. We use data from the years 2024 and 2025 as the test set. The raw data is accessed directly from the Board of Governors of the Federal Reserve System, which is in the public domain and available for open-source redistribution and research use.

**Oil Price**   This dataset records the daily trading prices of key energy commodities, specifically Crude Oil and various Refined Products. The data covers a business-day frequency from June 14, 2006, to September 30, 2025. Same as above, the observations from the year 2025 are reserved as the test set. The raw data is obtained from the U.S. Energy Information Administration (EIA); as a U.S. government publication, it resides in the public domain. In accordance with their guidelines, we include the required attribution: "Source: U.S. Energy Information Administration (May 2026)"

### B.6. Economics

**Job Claims**   This dataset tracks two key indicators of the U.S. labor market, specifically the 4-week moving averages of Initial Claims and Continued Claims. Sampled on a weekly basis, the data spans from January 1, 2022, to September 27, 2025. The final 52 weeks of the series are reserved as the test set. The raw data is accessed directly from the U.S. Department of Labor's Employment and Training Administration, falling into the public domain as a federal work available for open redistribution.

**Uncertainty-1M**   This multivariate dataset comprises three key monthly uncertainty indices of the US—macroeconomic, financial, and real uncertainty, sourced from the JLN and LMN frameworks (Jurado et al., 2015; Ludvigson et al., 2021). These series quantify the 1-month-ahead unpredictability of the economic state, derived from a broad spectrum of real time series. The dataset covers the period from July 1960 to June 2025. We designate the data from January 2022 to June 2025 as the test set. The data is publicly provided by the authors for research purposes and is accessed directly from their academic repository.

**Housing Inventory**   This dataset tracks monthly supply-side dynamics of the U.S. real estate market. It includes four key indicators: Active Listing Count, Median Days on Market, Median Listing Price per Square Foot, and Median Home Size. The data spans from July 2016 to December 2025. The raw data is publicly provided by the Realtor.com® Real Estate Data Library. In accordance with their data usage guidelines, we acknowledge Realtor.com® Economic Research as the source of this dataset.

**JOLTS**   This dataset, derived from the Job Openings and Labor Turnover Survey (JOLTS), provides a comprehensive view of U.S. labor market dynamics. It tracks key monthly indicators such as Job Openings, Hires, and Quits, covering the period from December 2000 to August 2025. The raw data is retrieved from the U.S. Bureau of Labor Statistics (BLS), a federal agency whose publications are in the public domain.

**US Labor**   This dataset provides a comprehensive overview of the U.S. labor market, aggregating monthly statistics on labor force participation, employment levels, and unemployment rates. The data spans from January 1994 to August 2025. The raw data is accessed directly from the BLS database under the same public-domain status for unrestricted research use.

**Vehicle Supply**   This dataset monitors supply-side dynamics in the U.S. automotive industry. It encompasses key metrics including domestic production, inventory levels, the inventory-to-sales ratio, and cross-border trade flows (imports from Canada/Mexico and exports). The data spans monthly from January 1993 to July 2025. The raw data is accessed directly from the U.S. Bureau of Economic Analysis (BEA), specifically via the "Motor vehicles" dataset provided within their supplemental GDP information. As a federal government publication, it is in the public domain and open for redistribution.

**Auto Production-SF**   This dataset contains the regular seasonal adjustment factors for U.S. auto production. These statistical components represent recurring temporal patterns driven by annual events such as model changeovers and holiday shutdowns. The raw data is accessed directly from the Board of Governors of the Federal Reserve System, inheriting the public-domain status for open-source redistribution.

**Commodity Production**   This dataset is a collection of historical monthly univariate time series tracking U.S. commodity production (e.g., steel, silver). The series vary in length and predominantly cover the industrial dynamics of the early to mid-20th century. The raw data is accessed directly from the National Bureau of Economic Research (NBER) Macrohistory Database, hosted within their Public Use Data Archive for open-source academic redistribution.

**Commodity Import**   Similar to Commodity Production, this dataset includes historical monthly univariate time series of U.S. commodity imports, including bananas, coffee, and crude rubber. The raw data is also drawn from the NBER Public Use Data Archive, making these historical factual series freely accessible for unrestricted applications.

**WUI-Global**   This dataset includes quarterly World Uncertainty Indices (WUI) (Ahir et al., 2022) for 15 countries. The dataset spans from Q1 1952 to Q2 2025. We forecast the future 4 quarters (1 year), using the final 5 years as the test set. The raw data is publicly provided via the Economic Policy Uncertainty repository under a CC BY 4.0 License.

**Global Price**   This dataset tracks quarterly global market prices for a broad spectrum of commodities. It encompasses three major sectors: Energy (e.g., Crude Oil, Coal, Natural Gas), Metals (e.g., Aluminum, Copper, Iron Ore), and Agriculture (e.g., Coffee, Soybeans, Wheat). Additionally, it features aggregate indices for food, beverages, and industrial materials. The data spans from 1990 Q1 to 2025 Q2. The raw data is sourced from the International Monetary Fund (IMF). Per their usage terms permitting derivative works, we explicitly state the original statistics were materially transformed for time-series forecasting. Following their guidelines, we acknowledge the data source: "Source: International Monetary Fund, Primary Commodity Price System (PCPS), `https://data.imf.org/en/datasets/IMF.RES:PCPS`."

### B.7. Sales

**Vehicle Sales**   This dataset encompasses monthly U.S. vehicle sales across diverse categories from 1976 through 2025. Consistent with our experimental setup, the final five-year period is reserved as a hold-out test set. The raw data is also accessed from the BEA "Motor vehicles" database, sharing the same public-domain status for unrestricted research and open redistribution use.

**Online Retail II**   This dataset encompasses all transactions for a UK-based, non-store online retailer from December 1, 2009, to December 9, 2011 (Asuncion et al., 2007). We aggregate individual transactions into daily totals to derive a univariate time series representing the total daily sales volume. The dataset is sourced from the UCI Machine Learning Repository and is publicly available under the Creative Commons Attribution 4.0 International (CC BY 4.0) License.

**Supply Chain (Customer & Location)**   This dataset comprises five years of simulated delivery records spanning from January 2, 2020, to June 30, 2025. To capture both micro and macro demand patterns, the raw transactional data—including daily order counts, piece volumes, and total revenue—are aggregated into two distinct multivariate time series. The customer-level version treats each individual client as a separate variate, while the location-level version aggregates these metrics by geographic city. This dual-aggregation approach allows for a comprehensive analysis of supply chain dynamics from both relational and spatial perspectives. This dataset is hosted on Kaggle and is publicly available under the Apache License 2.0.

### B.8. CloudOPS

**Azure2019**   Following the methodology of (Woo et al., 2023), we curated and processed CPU utilization traces from the Azure cloud platform (Cortez et al., 2017). The dataset captures average, minimum, and maximum CPU metrics at a 5-minute resolution across three workload categories: Delay-insensitive (D), Interactive (I), and Unknown (U), where categories D and I exhibit longer temporal durations compared to U. Due to the vast volume of raw data, we randomly sampled 2,000 instances; after applying quality filters, a refined subset was retained, comprising 989 instances for D, 492 for I, and 78 for U. The dataset is made publicly available by Microsoft Azure under the Creative Commons Attribution 4.0 International (CC BY 4.0) License.

### B.9. Industry

**Smart Manufacturing**   Designed for predictive maintenance and anomaly detection, this dataset simulates IoT sensor readings from industrial machinery. Due to the inherent irregular sampling rate of the raw logs, we downsample the series to

an hourly level. The dataset tracks five multivariate features critical to machine health: temperature, vibration, humidity, pressure, and energy consumption. This dataset is hosted on Kaggle and is dedicated to the public domain under the CC0 1.0 Universal (CC0 1.0) Public Domain Dedication.

**MetroPT-3** This dataset comprises multivariate time series recorded from the Air Production Unit (APU) of a metro train's compressor between February and August 2020 (Davari et al., 2021; Veloso et al., 2022). Sourced from an operational environment, the original data captures 15 distinct signals—including pressure, motor current, oil temperature, and valve states—logged at a frequency of 1Hz. We downsample the data into 5-minute resolution for practical predictive maintenance purposes. The dataset is publicly available under the Creative Commons Attribution 4.0 International (CC BY 4.0) License.

## C. Implementation Details

### C.1. Data Screening Pipeline

As described in Section 4.2, we implemented an automated quality assurance pipeline to ensure the data integrity of the time series in our benchmark. It systematically examines each series at both the variate and dataset levels. The pipeline operates in four phases, as outlined in Algorithm 1. For each variate, we apply a set of rules to assess data quality and impute extreme outliers during the screening process. The detailed variate-level checks are presented in Algorithm 2.

The pipeline outputs a comprehensive quality summary $\mathcal{Q}$ that documents all check results, along with a cleaned dataset $\mathcal{D}'$ where extreme outliers have been imputed via forward-filling. These outputs are then passed to the subsequent human decision-making stage for data finalization and task formulation.

---

**Algorithm 1** Time Series Quality Assurance Pipeline

---

**Require:** Dataset $\mathcal{D} = \{\mathbf{X}^{(i)}\}_{i=1}^{N}$ where each $\mathbf{X}^{(i)} \in \mathbb{R}^{L_i \times D_i}$
**Require:** Parameters: $\tau_{\text{miss}}$ (missing rate threshold), $\tau_{\text{corr}}$ (correlation threshold), $\tau_{\text{len}}^{(f)}$ (minimum length per frequency $f$)
**Ensure:** Quality summary $\mathcal{Q}$ and cleaned dataset $\mathcal{D}'$
 1: **Phase 1: Per-Series Processing**
 2: **for** each series $\mathbf{X}^{(i)} \in \mathcal{D}$ **do**
 3:     Normalize timestamp column to first position
 4:     $f \leftarrow \text{INFERFREQUENCY}(\mathbf{X}^{(i)})$
 5:     $\tau_{\text{len}} \leftarrow \tau_{\text{len}}^{(f)}$
 6:     Fill missing timestamps via reindexing with $f$
 7:     **Phase 2: Per-Variate Quality Checks**
 8:     **for** each variate $\mathbf{x}_d \in \mathbf{X}^{(i)}, d = 1, \ldots, D_i$ **do**
 9:         $\text{quality}_d \leftarrow \text{UNIVARIATEQUALITYCHECK}(\mathbf{x}_d, \tau_{\text{miss}}, \tau_{\text{len}})$
10:     **end for**
11:     **Phase 3: Multivariate Correlation Check**
12:     **if** $D_i > 1$ **then**
13:         Compute Pearson correlation matrix $\mathbf{R} \in \mathbb{R}^{D_i \times D_i}$
14:         Identify pairs $(\mathbf{x}_j, \mathbf{x}_k)$ where $|r_{jk}| > \tau_{\text{corr}}$
15:     **end if**
16: **end for**
17: **Phase 4: Cross-Series Correlation (for univariate datasets)**
18: **if** all $D_i = 1$ **and** all $L_i$ equal **then**
19:     Compute inter-series correlation matrix
20:     Flag highly correlated series pairs
21: **end if**
22: **return** Quality summary $\mathcal{Q}$, cleaned dataset $\mathcal{D}'$

---

---

**Algorithm 2** Univariate Quality Check

---

**Require:** Variate $\mathbf{x} = (x_1, x_2, \ldots, x_L) \in \mathbb{R}^L$, thresholds $\tau_{\text{miss}}, \tau_{\text{len}}$
**Ensure:** Quality result $\mathcal{R} = \{\texttt{predictable}, \tilde{\mathbf{x}}\}$

1: **Check 1: Data Type**
2: **if** $\mathbf{x}$ is not numeric **then**
3:     **return** $\{\texttt{predictable} = \text{False}\}$
4: **end if**
5: **Check 2: Data Integrity**
6: **if** $L < \tau_{\text{len}}$ **then**
7:     $\texttt{predictable} \leftarrow \text{False}$
8: **end if**
9: $\rho_{\text{miss}} \leftarrow |\{t : x_t = \text{NaN}\}|/L$
10: **if** $\rho_{\text{miss}} > \tau_{\text{miss}}$ **then**
11:     $\texttt{predictable} \leftarrow \text{False}$
12: **end if**
13: $\bar{\mathbf{x}} \leftarrow$ Forward-fill then backward-fill NaN values
14: **Check 3: Signal Existence (Constant Series Detection)**
15: $\texttt{topk\_dom} \leftarrow \sum_{i=1}^{5} p_{(i)}$ {Top-5 value frequencies}
16: $H \leftarrow -\sum_v p_v \log p_v / \log(|\mathcal{V}|)$ {Normalized entropy}
17: **if** $\texttt{topk\_dom} \geq 0.5$ **or** $H < 0.1$ **then**
18:     **return** $\{\texttt{predictable} = \text{False}\}$
19: **end if**
20: **Check 4: White Noise Test (Ljung-Box)**
21: $p_{\text{LB}} \leftarrow \textsc{LjungBoxTest}(\bar{\mathbf{x}}, \text{lags} = [10, 20])$
22: **if** $\min(p_{\text{LB}}) > 0.05$ **then**
23:     **return** $\{\texttt{predictable} = \text{False}\}$ {White Noise}
24: **end if**
25: **Check 5: Outlier Detection and Cleaning (Sliding Window IQR)**
26: **for** $t = 1$ to $L$ **do**
27:     $m_t \leftarrow \text{median}(\mathcal{W}_t)$, $\text{IQR}_t \leftarrow Q_{0.75}(\mathcal{W}_t) - Q_{0.25}(\mathcal{W}_t)$
28:     $d_t \leftarrow |x_t - m_t|/\text{IQR}_t$
29: **end for**
30: $\mathcal{I}_{\text{trans}} \leftarrow \{t : k_{\text{trans}} < d_t < k_{\text{ext}}\}$, $\mathcal{I}_{\text{ext}} \leftarrow \{t : d_t \geq k_{\text{ext}}\}$
31: **if** $|\mathcal{I}_{\text{ext}}|/L > \tau_{\text{ext}}$ **then**
32:     **return** $\{\texttt{predictable} = \text{False}\}$
33: **end if**
34: $\tilde{\mathbf{x}} \leftarrow$ Replace $x_t$ for $t \in \mathcal{I}_{\text{ext}}$ with forward-filled values
35: **return** $\{\texttt{predictable} = \text{True}, \tilde{\mathbf{x}}\}$

---

### C.2. Details of LLM usage

To bridge the gap between the automated data screening and the final task formulation, we utilized Gemini 3 Pro (via the standard web interface) as a domain-expert assistant. Specifically, utilizing the prompt templates detailed in Figures 7 and 8, the LLM was employed to perform the following critical tasks:

- **Dataset Pruning**: Serving as a domain-aware safeguard, the LLM processes the automated quality summary $\mathcal{Q}$ to evaluate flagged variates within their specific application context. This approach allows it to confirm or override automatic deletions, preserving indispensable series that exhibit expected statistical imperfections.

- **Horizon Configuration**: The LLM analyzes the sampling frequency and real-world domain of each cleaned dataset $\mathcal{D}'$. This ensures the configured short, medium, and long-term forecasting windows accurately reflect operational decision cycles, underlying periodicity, and logical predictability limits.

**Prompt for LLM-assisted Screening Review and Data Filtering**

**USER**: You are a domain-aware data quality reviewer for time series forecasting benchmarks. An automatic screening pipeline has already flagged potentially problematic variates and series based on statistical rules (e.g., missing rate thresholds, constant-value detection). Your role is **not** to repeat these rules, but to serve as a **final safeguard** before any deletion is executed, applying domain knowledge to confirm, override, or refine the automatic recommendations.

**Context**

- **Dataset Description**: {data_description}
- **Sampling Frequency**: {freq}

**Screening Report**

{paste_screening_report_here}

**Your Task**

For each flagged variate (and its affected series), evaluate whether the recommended deletion is appropriate by reasoning through the following dimensions:

1. Does the variate have the value for forecasting in this application? If not, deletion should be recommended (e.g. depth of sensor when monitoring water quality is not a target for forecasting). Conversely, if variate is indispensable despite its statistical imperfections, deletion should be avoided or minimized.

2. Is it safe or common practice to exclude this variate in this application (e.g. different dissolved oxygen indicators like DO, BOD, COD, etc. are almost roughly equivalent, thus it is safe and common to exclude most of them)?

3. Review the root cause of each flag (e.g., high missing rate, white noise, anomalous spikes) in the context of the application domain. Some patterns that appear problematic statistically may be expected domain behavior (e.g., sensor saturation, seasonal shutdowns, scheduled maintenance windows) and still have predictive value.

4. Does the sampling frequency affect the deletion or removal of this variate?

**Output Format**

For each flagged variate, provide:

```
Variate: <name>
Decision: Drop Series <list> | Drop Variate | Keep (override)
Rationale: <2-3 sentences explicitly referencing the
           domain context, root cause, and frequency.>
```

If multiple variates are flagged, evaluate each independently. Conclude with an overall summary of all final actions.

**ASSISTANT**: ...

*Figure 7.* Prompt template for LLM-assisted data quality screening and domain-aware pruning.

---

**Prompt for LLM-assisted Horizon Selection**

USER: You are an expert AI researcher specializing in time series forecasting and real-world industry applications. Your task is to determine three reasonable **prediction horizons** (short-term, medium-term, and long-term) for a given dataset based on its sampling frequency and application context.

**Context**

- **Sampling Frequency**: {freq}
- **Application Context**: {application_context}

**Guidelines**

1. **Short-term**: Provide a small range of candidate values (not a single number) that reflect the immediate operational decision window in the application. Ensure the lower bound is not trivially small.

2. **Medium-term**: Should span one to several dominant periodic cycles of the data relevant to the application.

3. **Long-term** (most critical): Must balance *predictability* against *utility*. Apply the following principles:
   - **Avoid "too long":** Do not set a horizon where the signal degrades to noise. For chaotic or physics-driven processes (e.g., wind speed, turbulence), cap the horizon at the physical predictability limit.
   - **Avoid "too short":** The horizon must be long enough to test the model's ability to capture long-range dependencies and trends.
   - **Periodicity vs. Chaos:** Human-behavior-driven data (e.g., traffic, electricity load) exhibits strong weekly periodicity and can support longer horizons. Pure physics-driven data (e.g., wind, turbulence) is chaotic, and horizons must be conservative.

**Output Format**

For each horizon, provide:

```
Short:  <steps_low>-<steps_high> (<real time range>)
Rationale: <1-2 sentences on the application scenario.>
Medium: <steps> (<real time equivalent>)
Rationale: ...
Long:   <steps> (<real time equivalent>)
Rationale: ...
```

*Example 1* — Frequency: 15-minute; Context: Parking Lot Availability.
- **Short**: 4–16 steps (1–4 hrs). Covers immediate navigation and queuing time for drivers.
- **Medium**: 96 steps (24 hrs). Captures the full diurnal cycle; useful for commuters planning the next day.
- **Long**: 672 steps (1 week). Parking demand is driven by human routines with strong weekday-vs-weekend periodicity; 1 week is the minimum period to capture this cycle.

*Example 2* — Frequency: Hourly; Context: Wind Speed (Meteorology / Renewable Energy).
- **Short**: 12–24 steps (12–24 hrs). Covers intra-day to day-ahead wind power dispatch.
- **Medium**: 72 steps (3 days). Covers typical synoptic weather events (e.g., passage of a cold front); vital for turbine maintenance scheduling.
- **Long**: 168 steps (1 week). Wind is chaotic with limited inertia; beyond 7 days, hourly accuracy degrades to noise. This caps at the physical predictability boundary.

Now, determine the three horizons for the following dataset:

...

ASSISTANT: ...

*Figure 8.* Prompt template used for determining practical forecasting horizons.

## C.3. Models

We evaluate 12 time series foundation models using their official checkpoints from HuggingFace. Table 3 summarizes the key hyperparameters used in our evaluation. All models are evaluated in a zero-shot setting without any fine-tuning. Experiments are run on a single NVIDIA RTX 3090 GPU (24GB VRAM).

*Table 3.* Model configurations used in evaluation. Context length refers to the maximum number of historical time steps used as input. Input mode indicates whether the model natively supports multivariate time series or processes each variate independently (univariate). Output type specifies whether models produce quantile forecasts or distribution-based probabilistic forecasts.

| Model | Checkpoint | Context Length | Input Mode | Output Type |
|---|---|---|---|---|
| TimesFM 2.5 | `google/timesfm-2.5-200m-pytorch` | 4096 | Univariate | Quantile |
| Chronos-2 | `amazon/chronos-2` | 8192 | Multivariate | Quantile |
| Kairos | `mldi-lab/Kairos_50m` | 2048 | Univariate | Quantile |
| Moirai 2.0 | `Salesforce/moirai-2.0-R-base` | 4000 | Univariate | Quantile |
| VisionTS++ | `Lefei/VisionTSpp` | 4000 | Multivariate | Quantile |
| TiRex | `NX-AI/TiRex` | 2048 | Univariate | Quantile |
| Toto | `Datadog/Toto-Open-Base-1.0` | 4096 | Multivariate | Distribution |
| Sundial | `thuml/sundial-base-128m` | 2880 | Univariate | Distribution |
| TimesFM 2.0 | `google/timesfm-2.0-500m-pytorch` | 2048 | Univariate | Quantile |
| Chronos-bolt | `amazon/chronos-bolt-base` | 2048 | Univariate | Quantile |
| Moirai | `Salesforce/moirai-1.1-R-base` | 4000 | Multivariate | Distribution |
| TimesFM 1.0 | `google/timesfm-1.0-200m-pytorch` | 512 | Univariate | Quantile |

**Output Types.** Models with quantile output produce 9 quantile forecasts at levels $\{0.1, 0.2, \ldots, 0.9\}$. Models with dstribution-based output generate 100 Monte Carlo samples for probabilistic forecasting.

**Multivariate Handling.** Chronos-2, VisionTS++, Toto, and Moirai (1.1) natively support multivariate time series input. For univariate models, we flatten multivariate series into independent univariate sequences and forecast each variate separately. For Moirai, when GPU memory is insufficient for multivariate inference, the model automatically falls back to univariate mode to complete the evaluation.

## C.4. Metrics

**Mean Absolute Scaled Error (MASE).** MASE provides a scale-independent assessment of forecast accuracy by normalizing the prediction error against the mean absolute error of a seasonal naive baseline. This normalization ensures interpretability across variates with varying magnitudes. It is defined as:

$$\text{MASE} = \frac{1}{H} \sum_{i=1}^{H} \frac{|\mathbf{Y}i - \widehat{\mathbf{Y}}i|}{\frac{1}{H-s} \sum_{j=s+1}^{H} |\mathbf{Y}j - \mathbf{Y}j - s|},$$

where $s$ denotes the periodicity of the data (e.g., season length), and $H$ represents the forecasting horizon. $\mathbf{Y}, \widehat{\mathbf{Y}} \in \mathbb{R}^{H \times D}$ denote the ground truth and the predicted values, respectively, over $D$ dimensions. The term $\mathbf{Y}_i$ refers to the value at the $i$-th future time step.

**Continuous Ranked Probability Score (CRPS)** To evaluate the quality of the predicted distribution, we employ CRPS, which measures the compatibility between the predicted cumulative distribution function (CDF) $F$ and the observed ground truth $\mathbf{Y}$. The integral form is given by:

$$\text{CRPS}(F, \mathbf{Y}) = \int_0^1 2\Lambda_\alpha(F^{-1}(\alpha), \mathbf{Y}) d\alpha,$$
$$\Lambda_\alpha(q, \mathbf{Y}) = (\alpha - \mathbb{1}\mathbf{Y} < q)(\mathbf{Y} - q),$$

where $\Lambda\alpha$ represents the quantile loss (or pinball loss) at the quantile level $\alpha$. To ensure computational tractability and provide a normalized metric for cross-dataset comparison, we utilize the normalized discrete approximation of CRPS. This is calculated as the average of the weighted Quantile Loss (wQL) across a set of discrete quantiles:

$$\text{CRPS} \approx \frac{1}{K}\sum_{k=1}^{K} \text{wQL}[\alpha_k],$$

$$\text{wQL}[\alpha] = 2\frac{\sum_i \Lambda_\alpha(\hat{q}_i(\alpha), \mathbf{Y}i)}{\sum i|\mathbf{Y}i|}.$$

Here, $\hat{q}_i(\alpha)$ denotes the predicted $\alpha$-quantile at time step $i$. In our evaluation, we employ $K = 9$ equidistant quantiles, specifically $\alpha_k \in \{0.1, 0.2, \ldots, 0.9\}$.

## D. Time Series Features

### D.1. Detailed Related Work

Early studies (Kang et al., 2017; Spiliotis et al., 2020) quantified statistical and meta-features—such as spectral entropy, trend and seasonality strength, seasonal period, ACF-1, kurtosis, skewness, non-linear autoregressive structure, and the optimal Box–Cox transformation parameter—primarily for PCA and dataset-level distribution visualization. Similarly, Monash (Godahewa et al., 2021) utilized ACF-1, trend and seasonality strength, entropy, and the Box-Cox parameter ($\lambda$) for distribution visualization. GIFT-EVAL (Aksu et al., 2024) characterized the most recent 500 time steps to reflect three temporal dimension: forecastability (trend and seasonal strength), regularity (entropy and Hurst exponent), and variability (stability and lumpiness). TFB (Qiu et al., 2024) expanded this scope by considering stationarity (via the modified Augmented Dick-Fuller test), correlation, and custom-defined metrics for shifting and transition. More recently, ProbTS (Zhang et al., 2024) analyzed trend, seasonality, and distribution complexity within fixed-length windows, while Boom (Cohen et al., 2025) selected six key features—including first-lag ACF, ARCH-LM statistic, spectral entropy, KPSS statistic, flat spots, and skewness—to characterize individual variates.

Despite this extensive characterization, prior works primarily utilize these features for data coverage analysis, visualization, and comparing distributional divergence against other benchmarks. Few studies adopt feature patterns as a distinct evaluation perspective. While TFB employs a single representative dataset per feature to reflect model performance, it lacks a systematic cross-dataset evaluation rooted in specific feature patterns.

In this work, we firstly introduce the pattern-driven analysis for cross-dataset evaluation. By archiving the pattern vector of each variate, we can use the pattern coding of a target pattern to retrieve variates exhibiting specific characteristics. The resulting aggregated metrics provide generalizable and high-level insights into model performance regarding each specific pattern.

### D.2. Definition of Selected Feature

In this section, we provide detailed definitions and mathematical formulations for each structural time series feature described in Section 5.1. Consistent with the main text, we use $\mathbf{x} \in \mathbb{R}^L$ to represent a time series variate (e.g. a UTS) of length $L$. Our feature extraction is implemented based on the `tsfeatures` library (Garza et al., 2022). By default, we use the test split to compute the tsfeatures for the variate. However, for variates with a test length below 500, the full variate is used instead.

The computation of structural features ($F_1 - F_5$) relies on Seasonal and Trend decomposition using Loess (STL) (Cleveland et al., 1990). While the underlying implementation extends standard STL to handle multiple seasonalities, we configure the decomposition to focus on the dominant seasonal period to ensure consistent feature comparability across diverse datasets.

Accordingly, as shown in Equation 1, we decompose the variate $\mathbf{x}$ into three additive component vectors:

- $T$ (Trend) represents the smoothed trend-cycle component, estimated via a Loess smoother.

- $S$ (Seasonality) corresponds to the periodic seasonal component.

- $R$ (Remainder) denotes the residual component, capturing the variability not explained by the trend or seasonality.

Based on this decomposition, the computation of our structural time series features is defined as follows.

**Trend Strength ($F_1$)**    This feature quantifies the strength of the trend component relative to the remainder. It is computed as:

$$F_1 = \max\left(0, 1 - \frac{\text{Var}(R)}{\text{Var}(T + R)}\right) \tag{3}$$

where $\text{Var}(\cdot)$ denotes the variance. A value close to 1 indicates that the trend variance dominates the remainder variance, signifying a strong trend, while a value close to 0 implies the trend is negligible compared to the noise.

**Trend Linearity ($F_2$)**    To characterize the structural shape of the trend, we fit an orthogonal quadratic regression model to the estimated trend component $T$:

$$T_t = \beta_0 + \beta_1 P_1(t) + \beta_2 P_2(t) + \varepsilon_t, \quad \text{for } t = 1, \ldots, L \tag{4}$$

where $P_1$ and $P_2$ are the first-order (linear) and second-order (quadratic) orthogonal polynomials of the time index, respectively. Trend Linearity ($F_2$) is defined as the coefficient of the linear term:

$$F_2 = \beta_1 \tag{5}$$

This metric captures the overall direction and steepness of the linear progression within the trend, independent of any curvature effects due to the orthogonality of the regressors.

**Seasonality Strength ($F_3$)**    Analogous to Trend Strength, this feature quantifies the relative importance of the seasonal component compared to the noise. It is computed as:

$$F_3 = \max\left(0, 1 - \frac{\text{Var}(R)}{\text{Var}(S + R)}\right) \tag{6}$$

A value close to 1 implies that the seasonal variation is much larger than the residual variability, indicating a distinct and strong seasonal pattern.

**Seasonality Correlation ($F_4$)**    This feature measures the stability and consistency of the seasonal shape across different cycles. Let $m$ be the seasonal period. We first truncate the seasonal component $S$ to contain only full cycles and reshape it into $K = \lfloor L/m \rfloor$ segments, denoted as vectors $\mathbf{s}_1, \ldots, \mathbf{s}_K \in \mathbb{R}^m$. $F_4$ is defined as the average Pearson correlation coefficient between all pairs of seasonal cycles:

$$F_4 = \frac{2}{K(K-1)} \sum_{i=1}^{K} \sum_{j=i+1}^{K} \text{Corr}(\mathbf{s}_i, \mathbf{s}_j) \tag{7}$$

High values indicate that the seasonal pattern repeats with a consistent shape over time, while low values suggest an evolving or unstable seasonal structure.

**Residual ACF-1 ($F_5$)**    This feature captures the linear dependence between consecutive values in the remainder component. It is calculated as the first-order autocorrelation coefficient:

$$F_5 = \text{Corr}(R_t, R_{t-1}) = \frac{\sum_{t=2}^{L}(R_t - \bar{R})(R_{t-1} - \bar{R})}{\sum_{t=1}^{L}(R_t - \bar{R})^2} \tag{8}$$

where $\bar{R}$ is the mean of the remainder vector. A value significantly different from zero suggests that there is still temporal structure (information) left in the residuals that was not captured by the trend or seasonality components.

**Complexity ($F_6$)**    This global feature quantifies the overall forecast difficulty of the raw variate $\mathbf{x}$. Similar to $F_6$, it is computed using the spectral entropy of the original time series:

$$F_6 = -\int_{-\pi}^{\pi} \hat{f}_{\mathbf{x}}(\lambda) \log \hat{f}_{\mathbf{x}}(\lambda) d\lambda \tag{9}$$

where $\hat{f}_{\mathbf{x}}(\lambda)$ is the normalized spectral density of $\mathbf{x}$. Higher complexity values indicate a series with high entropy (more noise-like), suggesting greater difficulty in forecasting.

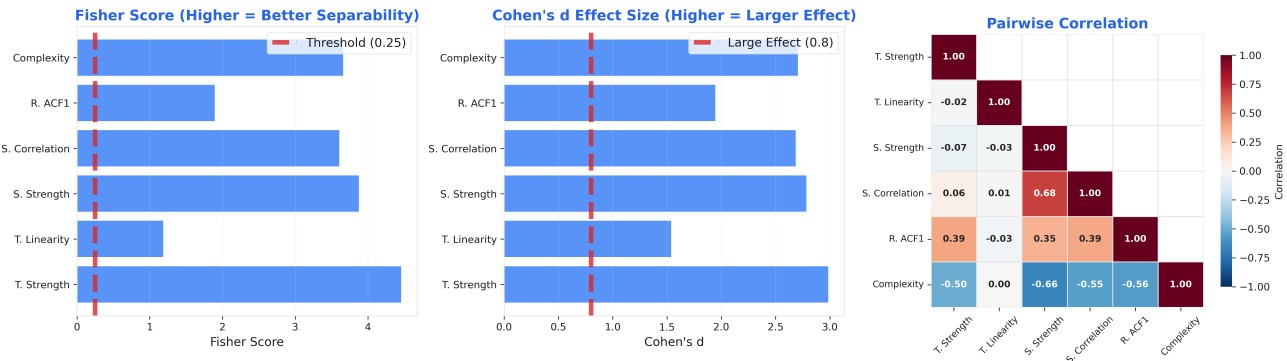

*Figure 9.* Quantitative analysis of feature informativeness for median-based binary grouping. **Left**: Fisher Score measures the ratio of between-group to within-group variance; all features exceed the threshold of 0.25 (red dashed line), indicating strong discriminative power. **Middle**: Cohen's $d$ quantifies effect size; all features surpass the large effect threshold of 0.8 (red dashed line), with values ranging from 1.54 to 2.98. **Right**: Pairwise correlation matrix shows that most features exhibit low to moderate correlations ($|r| < 0.7$), confirming they capture complementary time series characteristics

**Stationarity ($F_7$)** This binary feature indicates whether the raw time series $\mathbf{x}$ is stationary. It is derived from the Augmented Dickey-Fuller (ADF) unit root test (Dickey & Fuller, 1979). Let $p_{\text{ADF}}$ be the p-value obtained from the test. The feature is defined as:

$$F_7 = \mathbb{I}(p_{\text{ADF}} < 0.05) \tag{10}$$

where $\mathbb{I}(\cdot)$ is the indicator function. $F_7 = 1$ implies the series is stationary (no unit root) with 95% confidence, whereas $F_7 = 0$ indicates non-stationarity.

### D.3. Feature Distribution Analysis

To validate the effectiveness of median-based binary grouping for time series characterization, we conduct a quantitative analysis of the selected features. Our goal is to demonstrate that each feature provides meaningful discriminative power when partitioning time series into two groups based on its median value. Specifically, we aim to show that the resulting groups are statistically distinct, justifying the use of these features for stratified evaluation and analysis.

We employ two complementary metrics: **Fisher Score** for measuring class separability, and **Cohen's $d$** for quantifying effect size. Together, these metrics provide both a variance-normalized measure of group separation and an interpretable standardized effect size.

**Fisher Score.** The Fisher Score is a classical feature selection criterion that measures the ratio of between-class variance to within-class variance. For a feature $F_k$ with two groups defined by a threshold (median), the Fisher Score is computed as:

$$\text{Fisher Score} = \frac{(\mu_1 - \mu_2)^2}{\sigma_1^2 + \sigma_2^2} \tag{11}$$

where $\mu_1, \mu_2$ are the means and $\sigma_1^2, \sigma_2^2$ are the variances of the two groups, respectively. A higher Fisher Score indicates better separability between groups. Following common practice in feature selection literature, we consider a Fisher Score $> 0.25$ as indicative of meaningful discriminative power.

**Cohen's $d$.** Cohen's $d$ is a widely-used standardized effect size measure that quantifies the magnitude of difference between two group means in units of pooled standard deviation:

$$\text{Cohen's } d = \frac{\mu_1 - \mu_2}{\sigma_{\text{pooled}}} \tag{12}$$

where the pooled standard deviation is:

$$\sigma_{\text{pooled}} = \sqrt{\frac{(n_1 - 1)\sigma_1^2 + (n_2 - 1)\sigma_2^2}{n_1 + n_2 - 2}} \tag{13}$$

Cohen's $d$ provides interpretable benchmarks: $|d| < 0.2$ indicates a small effect, $0.2 \leq |d| \leq 0.8$ a medium effect, and $|d| > 0.8$ a large effect (Cohen, 2013).

We analyze six features extracted from 6,625 time series variates across our benchmark. Figure 9 summarizes the quantitative results. Several key observations emerge from the analysis:

**(1) All features demonstrate strong discriminative power.** Every feature achieves a Fisher Score substantially above the 0.25 threshold, ranging from 1.19 (Trend Linearity) to 4.46 (Trend Strength). This confirms that median-based partitioning creates groups with meaningfully different distributions for each feature.

**(2) Effect sizes are uniformly large.** All features exhibit Cohen's $d$ values well above 0.8, with the minimum being 1.54 (Trend Linearity) and the maximum reaching 2.98 (Trend Strength). According to standard interpretation guidelines, these represent *very large* effect sizes, indicating substantial practical differences between the high and low groups.

**(3) Features capture complementary information.** The pairwise correlation analysis reveals that most features exhibit low to moderate correlations ($|r| < 0.7$), indicating they capture distinct aspects of time series characteristics. Notably, Trend Linearity shows near-zero correlation with all other features ($|r| \leq 0.03$), confirming it measures an independent property. The strongest correlation occurs between Seasonality Strength and Seasonality Correlation ($r = 0.68$), which is expected as both characterize seasonal patterns. Complexity shows moderate negative correlations with seasonality-related features ($r = -0.66$ with Seasonality Strength, $r = -0.55$ with Seasonality Correlation) and Residual ACF1 ($r = -0.56$), suggesting that more complex time series tend to exhibit weaker seasonal patterns and lower residual autocorrelation. Overall, the correlation structure confirms that our feature set provides a diverse and complementary characterization of time series properties.

**(4) Statistical significance.** Mann-Whitney U tests confirm that all median-based groupings produce statistically significant differences ($p < 0.001$ for all features), ruling out the possibility that observed separations are due to chance.

These results validate our feature selection and demonstrate that median-based binary grouping produces meaningful stratifications of time series data. The consistently large effect sizes across all features support the use of this approach for analyzing model performance across different time series characteristics.

## E. Full Results

This section presents the comprehensive numerical results of our experiments. We begin with the overall aggregated performance in Table 4, followed by a detailed pattern-level breakdown in Figure 10 and Table 5. Finally, we provide the granular dataset-level results for all 50 datasets in Table 6. To ensure a multifaceted evaluation, we report six metrics in total. For both MASE and CRPS, we provide the raw values, scores normalized by the Seasonal Naive baseline, and task-level rankings.

*Table 4.* Detailed overall performance of TSFMs on TIME benchmark. Task-level results are normalized by Seasonal Naive and aggregated via geometric mean. **Bold** indicates the best performance, and underlined indicates the second best.

| Metric | TimesFM 2.5 | Chronos-2 | Kairos | Moirai 2.0 | VisionTS++ | TiRex | Toto | Sundial | TimesFM 2.0 | Chronos-bolt | Moirai | TimesFM 1.0 |
|---|---|---|---|---|---|---|---|---|---|---|---|---|
| **MASE** | 0.669 | **0.662** | 0.739 | 0.703 | 0.805 | 0.683 | 0.698 | 0.758 | 0.718 | 0.733 | 0.826 | 0.788 |
| **CRPS** | 0.567 | **0.556** | 0.644 | 0.588 | 0.671 | 0.573 | 0.584 | 0.663 | 0.620 | 0.620 | 0.666 | 0.689 |
| **MASE Rank** | 3.15 | **2.44** | 7.59 | 5.78 | 8.83 | 4.01 | 5.55 | 7.87 | 6.60 | 7.17 | 10.11 | 9.74 |
| **CRPS Rank** | 3.28 | **2.32** | 8.30 | 5.30 | 8.21 | 3.66 | 5.03 | 9.41 | 6.81 | 7.37 | 8.74 | 9.95 |

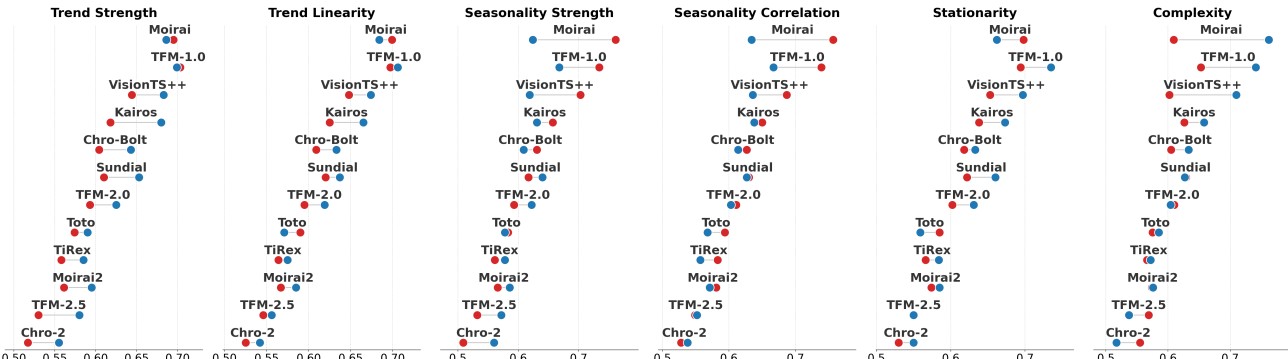

*Figure 10.* Comparison of **CRPS** across different feature-specific variates. Each row represents a model's performance on variates with $F_k = 1$ (🔴) and $F_k = 0$ (🔵). The distance between dots indicates the performance difference of the model to that specific feature.

*Table 5.* Pattern-specific performance (Normalized MASE and CRPS). Comparison between variates exhibiting a specific pattern (=1) and those that do not (=0).

| Feature | Group | Metric | TimesFM 2.5 | Chronos2 | Kairos | Moirai 2.0 | VisionTS++ | TiRex | Toto | Sundial | TimesFM 2.0 | Chronos-bolt | Moirai | TimesFM 1.0 |
|---|---|---|---|---|---|---|---|---|---|---|---|---|---|---|
| **Trend** | $F_k = 1$ | MASE | 0.603 | **0.59** | 0.681 | 0.635 | 0.726 | 0.635 | 0.654 | 0.662 | 0.654 | 0.681 | 0.791 | 0.762 |
| | | CRPS | 0.53 | **0.517** | 0.618 | 0.561 | 0.644 | 0.558 | 0.574 | 0.61 | 0.593 | 0.604 | 0.695 | 0.703 |
| | $F_k = 0$ | MASE | 0.651 | **0.63** | 0.725 | 0.665 | 0.768 | 0.669 | 0.669 | 0.697 | 0.678 | 0.714 | 0.769 | 0.735 |
| | | CRPS | 0.58 | **0.555** | 0.68 | 0.595 | 0.683 | 0.585 | 0.59 | 0.653 | 0.625 | 0.643 | 0.686 | 0.699 |
| **T. Linearity** | $F_k = 1$ | MASE | 0.615 | **0.594** | 0.679 | 0.635 | 0.726 | 0.641 | 0.668 | 0.666 | 0.649 | 0.679 | 0.785 | 0.743 |
| | | CRPS | 0.546 | **0.525** | 0.625 | 0.567 | 0.648 | 0.564 | 0.59 | 0.62 | 0.595 | 0.609 | 0.699 | 0.697 |
| | $F_k = 0$ | MASE | 0.633 | **0.621** | 0.722 | 0.662 | 0.762 | 0.658 | 0.653 | 0.689 | 0.68 | 0.713 | 0.778 | 0.758 |
| | | CRPS | 0.556 | **0.542** | 0.665 | 0.585 | 0.674 | 0.575 | 0.571 | 0.637 | 0.619 | 0.633 | 0.684 | 0.706 |
| **Seasonality** | $F_k = 1$ | MASE | 0.588 | **0.565** | 0.696 | 0.621 | 0.774 | 0.627 | 0.65 | 0.655 | 0.641 | 0.693 | 0.846 | 0.768 |
| | | CRPS | 0.532 | **0.509** | 0.657 | 0.566 | 0.703 | 0.561 | 0.583 | 0.617 | 0.593 | 0.631 | 0.761 | 0.734 |
| | $F_k = 0$ | MASE | 0.664 | **0.654** | 0.703 | 0.677 | 0.713 | 0.674 | 0.671 | 0.701 | 0.689 | 0.698 | 0.718 | 0.732 |
| | | CRPS | 0.572 | **0.56** | 0.631 | 0.586 | 0.619 | 0.578 | 0.578 | 0.64 | 0.622 | 0.609 | 0.624 | 0.668 |
| **S. Correlation** | $F_k = 1$ | MASE | 0.619 | **0.595** | 0.704 | 0.647 | 0.771 | 0.659 | 0.672 | 0.679 | 0.667 | 0.7 | 0.857 | 0.786 |
| | | CRPS | 0.549 | **0.528** | 0.65 | 0.581 | 0.687 | 0.583 | 0.594 | 0.63 | 0.611 | 0.627 | 0.757 | 0.739 |
| | $F_k = 0$ | MASE | 0.628 | **0.618** | 0.695 | 0.648 | 0.718 | 0.64 | 0.65 | 0.675 | 0.661 | 0.69 | 0.715 | 0.717 |
| | | CRPS | 0.552 | **0.538** | 0.638 | 0.571 | 0.636 | 0.557 | 0.568 | 0.627 | 0.603 | 0.614 | 0.634 | 0.667 |
| **R. ACF1** | $F_k = 1$ | MASE | 0.58 | **0.559** | 0.671 | 0.613 | 0.725 | 0.616 | 0.654 | 0.65 | 0.636 | 0.674 | 0.831 | 0.779 |
| | | CRPS | 0.505 | **0.486** | 0.603 | 0.536 | 0.639 | 0.535 | 0.568 | 0.591 | 0.564 | 0.591 | 0.724 | 0.707 |
| | $F_k = 0$ | MASE | 0.656 | **0.642** | 0.72 | 0.673 | 0.757 | 0.673 | 0.665 | 0.696 | 0.684 | 0.71 | 0.749 | 0.731 |
| | | CRPS | 0.585 | **0.569** | 0.675 | 0.605 | 0.676 | 0.595 | 0.59 | 0.655 | 0.638 | 0.642 | 0.669 | 0.697 |
| **Stationarity** | $F_k = 1$ | MASE | 0.626 | **0.605** | 0.696 | 0.648 | 0.739 | 0.65 | 0.668 | 0.672 | 0.662 | 0.695 | 0.79 | 0.745 |
| | | CRPS | 0.551 | **0.53** | 0.638 | 0.574 | 0.653 | 0.566 | 0.585 | 0.622 | 0.602 | 0.618 | 0.698 | 0.694 |
| | $F_k = 0$ | MASE | **0.612** | 0.615 | 0.716 | 0.646 | 0.767 | 0.646 | 0.624 | 0.699 | 0.674 | 0.694 | 0.739 | 0.776 |
| | | CRPS | **0.55** | **0.55** | 0.673 | 0.585 | 0.697 | 0.584 | 0.559 | 0.66 | 0.631 | 0.633 | 0.662 | 0.735 |
| **Complexity** | $F_k = 1$ | MASE | 0.657 | **0.647** | 0.692 | 0.662 | 0.692 | 0.661 | 0.668 | 0.687 | 0.674 | 0.69 | 0.695 | 0.708 |
| | | CRPS | 0.569 | **0.555** | 0.626 | 0.574 | 0.602 | 0.566 | 0.575 | 0.629 | 0.61 | 0.605 | 0.609 | 0.653 |
| | $F_k = 0$ | MASE | 0.599 | **0.578** | 0.705 | 0.637 | 0.786 | 0.641 | 0.654 | 0.669 | 0.656 | 0.699 | 0.855 | 0.784 |
| | | CRPS | 0.537 | **0.517** | 0.658 | 0.576 | 0.71 | 0.572 | 0.585 | 0.627 | 0.604 | 0.633 | 0.762 | 0.741 |

*Table 6.* Full Results of Per Dataset.

| Dataset | Metric | TimesFM 2.5 | Chronos-2 | Kairos | Moirai 2.0 | VisionTS++ | TiRex | Toto | Sundial | TimesFM 2.0 | Chronos-bolt | Moirai | TimesFM 1.0 |
|---|---|---|---|---|---|---|---|---|---|---|---|---|---|
| **Water Quality-Darwin** | MASE (raw) | 0.993 | **0.938** | 1.220 | 1.092 | 1.270 | 1.024 | 1.007 | 1.292 | 1.182 | 1.036 | 1.428 | 1.098 |
| | CRPS (raw) | 0.059 | **0.054** | 0.081 | 0.069 | 0.090 | 0.065 | 0.058 | 0.089 | 0.076 | 0.066 | 0.077 | 0.075 |
| | MASE (norm.) | 0.739 | **0.706** | 0.928 | 0.816 | 0.989 | 0.766 | 0.754 | 0.953 | 0.888 | 0.779 | 1.072 | 0.827 |
| | CRPS (norm.) | 0.504 | **0.466** | 0.694 | 0.584 | 0.795 | 0.553 | 0.500 | 0.745 | 0.645 | 0.563 | 0.675 | 0.630 |
| | MASE (rank) | 2 | **1.333** | 10.667 | 6 | 10 | 4.333 | 3.333 | 11 | 9.333 | 4.667 | 12.333 | 6.667 |
| | CRPS (rank) | 2.667 | **1** | 9.667 | 6.333 | 10.667 | 4.333 | 2.333 | 10.667 | 8.667 | 5 | 8.333 | 8.333 |
| **Current Velocity/5T** | MASE (raw) | 0.729 | **0.710** | 0.956 | 0.929 | 1.183 | 0.789 | 0.833 | 0.907 | 0.804 | 1.008 | 1.281 | 0.897 |
| | CRPS (raw) | 0.277 | **0.270** | 0.337 | 0.326 | 0.367 | 0.284 | 0.296 | 0.320 | 0.306 | 0.334 | 0.386 | 0.329 |
| | MASE (norm.) | 0.601 | **0.588** | 0.787 | 0.752 | 1.005 | 0.647 | 0.678 | 0.738 | 0.662 | 0.819 | 1.046 | 0.734 |
| | CRPS (norm.) | 0.649 | **0.632** | 0.786 | 0.757 | 0.862 | 0.666 | 0.691 | 0.749 | 0.712 | 0.780 | 0.903 | 0.766 |
| | MASE (rank) | 2 | **1** | 9 | 8 | 11.333 | 3 | 4.667 | 7 | 4.333 | 10 | 12.333 | 6.667 |
| | CRPS (rank) | 2 | **1** | 9.333 | 7.667 | 9.667 | 3 | 4.333 | 7 | 5.333 | 9 | 12 | 8 |
| **Current Velocity/10T** | MASE (raw) | 0.695 | **0.686** | 0.815 | 0.734 | 0.884 | 0.703 | 0.738 | 0.768 | 0.738 | 0.746 | 0.904 | 0.814 |
| | CRPS (raw) | 0.333 | **0.332** | 0.354 | 0.342 | 0.358 | 0.333 | 0.343 | 0.360 | 0.350 | 0.343 | 0.363 | 0.364 |
| | MASE (norm.) | 0.648 | **0.640** | 0.754 | 0.682 | 0.840 | 0.655 | 0.685 | 0.714 | 0.685 | 0.693 | 0.836 | 0.752 |
| | CRPS (norm.) | 0.659 | **0.655** | 0.697 | 0.675 | 0.708 | 0.658 | 0.677 | 0.710 | 0.689 | 0.677 | 0.716 | 0.717 |
| | MASE (rank) | 2.333 | **1** | 10 | 5.667 | 10.667 | 2.667 | 5 | 8 | 5 | 7 | 11.667 | 9.667 |
| | CRPS (rank) | 2.333 | **2** | 9 | 5.667 | 7.667 | 2 | 6.333 | 10 | 6.333 | 6 | 10.333 | 10.333 |
| **Current Velocity/15T** | MASE (raw) | 0.762 | **0.746** | 0.866 | 0.860 | 1.017 | 0.779 | 0.814 | 0.897 | 0.796 | 1.017 | 1.094 | 0.864 |
| | CRPS (raw) | 0.316 | **0.306** | 0.350 | 0.339 | 0.368 | 0.319 | 0.324 | 0.338 | 0.329 | 0.358 | 0.367 | 0.355 |
| | MASE (norm.) | 0.644 | **0.631** | 0.736 | 0.723 | 0.898 | 0.656 | 0.687 | 0.758 | 0.673 | 0.852 | 0.927 | 0.726 |
| | CRPS (norm.) | 0.648 | **0.627** | 0.716 | 0.693 | 0.760 | 0.655 | 0.665 | 0.673 | 0.673 | 0.734 | 0.754 | 0.722 |
| | MASE (rank) | 2 | **1** | 7.333 | 6.667 | 9.667 | 3 | 5 | 9.333 | 4 | 11.333 | 12.333 | 7.667 |
| | CRPS (rank) | 2 | **1** | 9 | 6.667 | 8.333 | 3.333 | 5 | 7.667 | 5 | 9.667 | 11 | 9.333 |
| **Current Velocity/20T** | MASE (raw) | **0.567** | 0.603 | 0.790 | 0.787 | 0.876 | 0.639 | 0.665 | 0.621 | 0.608 | 0.743 | 0.848 | 0.719 |
| | CRPS (raw) | **0.276** | 0.286 | 0.343 | 0.315 | 0.293 | 0.278 | 0.289 | 0.291 | 0.314 | 0.319 | 0.289 | 0.361 |
| | MASE (norm.) | **0.545** | 0.573 | 0.733 | 0.715 | 0.875 | 0.608 | 0.624 | 0.596 | 0.583 | 0.699 | 0.811 | 0.675 |
| | CRPS (norm.) | **0.641** | 0.661 | 0.769 | 0.720 | 0.683 | 0.645 | 0.670 | 0.678 | 0.720 | 0.727 | 0.672 | 0.815 |
| | MASE (rank) | **1** | 3 | 10.333 | 9.667 | 10 | 4.667 | 4.667 | 4.333 | 3.333 | 9 | 11 | 7.667 |
| | CRPS (rank) | **2** | 4 | 10 | 8 | 6.333 | 2 | 5.333 | 8 | 7 | 8.333 | 6.333 | 11.333 |
| **Current Velocity/H** | MASE (raw) | **1.062** | 1.063 | 1.373 | 1.121 | 1.482 | 1.092 | 1.175 | 1.775 | 1.258 | 1.313 | 1.405 | 1.457 |
| | CRPS (raw) | 0.308 | **0.304** | 0.374 | 0.323 | 0.363 | 0.305 | 0.325 | 0.330 | 0.359 | 0.355 | 0.355 | 0.409 |
| | MASE (norm.) | 0.650 | **0.648** | 0.844 | 0.684 | 0.930 | 0.662 | 0.715 | 1.082 | 0.760 | 0.798 | 0.869 | 0.887 |
| | CRPS (norm.) | 0.633 | **0.622** | 0.765 | 0.661 | 0.748 | 0.625 | 0.666 | 0.677 | 0.729 | 0.725 | 0.729 | 0.832 |
| | MASE (rank) | **1.667** | 2 | 8.667 | 4 | 10.667 | 2.333 | 5 | 12.333 | 6.667 | 6.667 | 9 | 10 |
| | CRPS (rank) | 2.667 | **1.667** | 10 | 4 | 8.667 | 1.667 | 5 | 6.333 | 8.333 | 9 | 9 | 11.667 |
| **CPHL/15T** | MASE (raw) | 0.977 | 0.958 | 0.992 | 0.994 | 0.973 | 0.957 | 0.938 | **0.922** | 0.944 | 0.975 | 0.998 | 1.100 |
| | CRPS (raw) | 0.213 | 0.212 | 0.225 | 0.216 | 0.213 | 0.213 | **0.208** | 0.216 | 0.212 | 0.218 | 0.222 | 0.252 |
| | MASE (norm.) | 0.657 | 0.646 | 0.666 | 0.665 | 0.661 | 0.645 | 0.634 | **0.623** | 0.637 | 0.655 | 0.675 | 0.735 |
| | CRPS (norm.) | 0.615 | 0.611 | 0.647 | 0.620 | 0.622 | 0.614 | **0.601** | 0.626 | 0.611 | 0.625 | 0.643 | 0.724 |
| | MASE (rank) | 8 | 4.333 | 10 | 7.333 | 7.333 | 4.667 | 3.667 | **1.333** | 4 | 5.667 | 10 | 11.667 |
| | CRPS (rank) | 4.667 | 4 | 10 | 5.333 | 5.333 | 5.667 | **3.333** | 6 | 4.667 | 7.667 | 9.333 | 12 |
| **CPHL/30T** | MASE (raw) | **1.406** | 1.407 | 1.482 | 1.499 | 1.490 | 1.446 | 1.432 | 1.451 | 1.451 | 1.471 | 1.583 | 1.515 |
| | CRPS (raw) | **0.248** | 0.250 | 0.270 | 0.267 | 0.266 | 0.262 | 0.255 | 0.287 | 0.262 | 0.268 | 0.285 | 0.286 |
| | MASE (norm.) | **0.765** | 0.765 | 0.805 | 0.816 | 0.808 | 0.787 | 0.779 | 0.789 | 0.790 | 0.799 | 0.862 | 0.822 |
| | CRPS (norm.) | **0.689** | 0.695 | 0.748 | 0.740 | 0.737 | 0.727 | 0.709 | 0.796 | 0.727 | 0.742 | 0.789 | 0.792 |
| | MASE (rank) | **1.333** | 1.667 | 8.667 | 9.333 | 7.667 | 4.667 | 3 | 6.333 | 6.333 | 7 | 12 | 10 |
| | CRPS (rank) | **1.333** | 1.667 | 8.333 | 7.333 | 6 | 4.667 | 3.333 | 11.333 | 5.667 | 6.667 | 10.667 | 11 |
| **CPHL/H** | MASE (raw) | **1.645** | 1.679 | 1.699 | 1.699 | 1.828 | 1.714 | 1.748 | 1.841 | 1.705 | 1.744 | 1.801 | 1.869 |
| | CRPS (raw) | **0.282** | 0.294 | 0.312 | 0.303 | 0.315 | 0.299 | 0.302 | 0.351 | 0.308 | 0.318 | 0.312 | 0.353 |
| | MASE (norm.) | **0.800** | 0.812 | 0.825 | 0.824 | 0.890 | 0.829 | 0.841 | 0.888 | 0.827 | 0.847 | 0.875 | 0.900 |
| | CRPS (norm.) | **0.743** | 0.770 | 0.815 | 0.795 | 0.829 | 0.783 | 0.791 | 0.916 | 0.805 | 0.833 | 0.820 | 0.918 |
| | MASE (rank) | **1.667** | 2.667 | 4 | 4.667 | 10 | 5.333 | 6 | 10.333 | 5 | 7.333 | 9.667 | 11.333 |
| | CRPS (rank) | **1** | 2 | 8 | 5.333 | 8.333 | 3.333 | 4.667 | 11.333 | 7 | 8.333 | 7.333 | 11.333 |
| **Coastal T-S/5T** | MASE (raw) | 1.261 | **1.256** | 1.263 | 1.277 | 1.299 | 1.259 | 1.303 | 1.272 | 1.284 | **1.256** | 1.322 | 1.291 |
| | CRPS (raw) | 0.072 | **0.068** | 0.081 | 0.070 | 0.076 | **0.068** | 0.069 | 0.093 | 0.080 | 0.075 | 0.069 | 0.082 |
| | MASE (norm.) | 0.819 | **0.816** | 0.821 | 0.830 | 0.846 | 0.818 | 0.845 | 0.827 | 0.834 | 0.817 | 0.858 | 0.839 |
| | CRPS (norm.) | 0.666 | **0.627** | 0.742 | 0.641 | 0.700 | 0.628 | 0.634 | 0.857 | 0.725 | 0.689 | 0.637 | 0.750 |
| | MASE (rank) | 3 | 2.333 | 5 | 7 | 9 | 3 | 10.667 | 5.667 | 8.333 | 3.333 | 11.333 | 9.333 |
| | CRPS (rank) | 6.667 | **1.667** | 10 | 4.667 | 9.333 | 2 | 2.667 | 11.333 | 7.667 | 8 | 4 | 10 |
| **Coastal T-S/15T** | MASE (raw) | **0.881** | 0.958 | 1.271 | 1.131 | 1.276 | 1.013 | 1.102 | 1.488 | 1.268 | 1.579 | 2.107 | 1.074 |
| | CRPS (raw) | **0.004** | **0.004** | 0.006 | 0.005 | 0.006 | **0.004** | 0.005 | 0.006 | 0.005 | 0.008 | 0.011 | 0.005 |
| | MASE (norm.) | **0.547** | 0.608 | 0.809 | 0.715 | 0.878 | 0.621 | 0.673 | 0.873 | 0.762 | 0.948 | 1.258 | 0.674 |
| | CRPS (norm.) | **0.512** | 0.571 | 0.877 | 0.708 | 0.995 | 0.593 | 0.710 | 0.860 | 0.717 | 1.082 | 1.456 | 0.697 |
| | MASE (rank) | **1** | 2.667 | 9.333 | 6.667 | 7.333 | 3.333 | 4.333 | 10 | 8 | 10.667 | 12.333 | 5.333 |
| | CRPS (rank) | **1** | 2.333 | 10 | 6.333 | 8.667 | 2.667 | 6 | 10 | 6.333 | 11 | 12.333 | 5.333 |
| **Coastal T-S/20T** | MASE (raw) | **1.019** | 1.024 | 1.135 | 1.078 | 1.127 | 1.051 | 1.033 | 1.171 | 1.071 | 1.093 | 1.199 | 1.139 |
| | CRPS (raw) | 0.043 | **0.042** | 0.055 | 0.043 | 0.046 | 0.042 | 0.044 | 0.057 | 0.054 | 0.048 | 0.045 | 0.056 |
| | MASE (norm.) | **0.733** | 0.738 | 0.810 | 0.774 | 0.816 | 0.754 | 0.743 | 0.830 | 0.768 | 0.784 | 0.856 | 0.818 |
| | CRPS (norm.) | 0.504 | **0.486** | 0.635 | 0.507 | 0.534 | 0.489 | 0.507 | 0.669 | 0.616 | 0.562 | 0.527 | 0.645 |
| | MASE (rank) | **1.333** | 2 | 9.333 | 5.667 | 8.667 | 4 | 2.667 | 10.333 | 5.333 | 7.667 | 11.333 | 10 |
| | CRPS (rank) | 4.333 | **1** | 10.333 | 4.333 | 7.333 | 2 | 4 | 10.667 | 8.667 | 8 | 6.333 | 11 |
| **Coastal T-S/H** | MASE (raw) | 1.618 | 1.612 | 1.732 | 1.637 | 1.797 | 1.718 | **1.602** | 1.955 | 1.859 | 1.710 | 1.896 | 1.909 |
| | CRPS (raw) | 0.029 | **0.029** | 0.033 | 0.030 | 0.033 | 0.031 | 0.029 | 0.034 | 0.035 | 0.031 | 0.033 | 0.038 |
| | MASE (norm.) | 0.778 | 0.778 | 0.842 | 0.791 | 0.902 | 0.820 | **0.774** | 0.925 | 0.888 | 0.828 | 0.920 | 0.920 |
| | CRPS (norm.) | **0.576** | 0.588 | 0.648 | 0.597 | 0.667 | 0.617 | **0.576** | 0.682 | 0.679 | 0.615 | 0.667 | 0.739 |
| | MASE (rank) | 2.667 | 2.667 | 7.333 | 4 | 7.667 | 5.667 | **2.333** | 11 | 8.667 | 6 | 10 | 10.667 |

*Table 6.* Full Results of Per Dataset (Continued).

| Dataset | Metric | TimesFM 2.5 | Chronos-2 | Kairos | Moirai 2.0 | VisionTS++ | TiRex | Toto | Sundial | TimesFM 2.0 | Chronos-bolt | Moirai | TimesFM 1.0 |
|---|---|---|---|---|---|---|---|---|---|---|---|---|---|
| | CRPS (rank) | 2 | 3.667 | 7.667 | 4 | 8.333 | 5 | **1.333** | 10.667 | 9.333 | 5.667 | 9.333 | 11 |
| **SG Weather** | MASE (raw) | 0.881 | **0.876** | 0.899 | 0.899 | 1.037 | 0.886 | 0.888 | 0.906 | 0.913 | 0.900 | 0.961 | 0.932 |
| | CRPS (raw) | 0.177 | **0.175** | 0.181 | 0.185 | 0.215 | 0.177 | 0.178 | 0.200 | 0.188 | 0.186 | 0.201 | 0.196 |
| | MASE (norm.) | 0.739 | **0.735** | 0.754 | 0.754 | 0.870 | 0.743 | 0.745 | 0.760 | 0.766 | 0.755 | 0.806 | 0.782 |
| | CRPS (norm.) | 0.548 | **0.542** | 0.562 | 0.573 | 0.665 | 0.550 | 0.552 | 0.620 | 0.582 | 0.579 | 0.623 | 0.608 |
| | MASE (rank) | 2 | **1** | 6 | 6 | 12 | 3 | 4 | 7.667 | 9 | 6.333 | 11 | 10 |
| | CRPS (rank) | 2.333 | **1** | 5 | 6 | 12 | 3 | 3.667 | 10.333 | 7.667 | 7.333 | 10.667 | 9 |
| **SG PM 2.5** | MASE (raw) | 0.935 | **0.915** | 0.920 | 0.937 | 0.933 | 0.919 | 0.927 | 0.920 | 0.926 | 0.949 | 0.946 | 0.978 |
| | CRPS (raw) | 0.320 | **0.311** | 0.317 | 0.319 | 0.317 | 0.312 | 0.315 | 0.335 | 0.320 | 0.327 | 0.326 | 0.339 |
| | MASE (norm.) | 0.750 | **0.734** | 0.738 | 0.752 | 0.749 | 0.738 | 0.744 | 0.738 | 0.743 | 0.761 | 0.758 | 0.784 |
| | CRPS (norm.) | 0.694 | **0.673** | 0.687 | 0.692 | 0.688 | 0.676 | 0.682 | 0.727 | 0.693 | 0.709 | 0.706 | 0.736 |
| | MASE (rank) | 7.667 | **2** | 3.333 | 8.667 | 6.333 | 2.667 | 6 | 3.333 | 6.667 | 10.333 | 9 | 12 |
| | CRPS (rank) | 6.333 | **1.333** | 4 | 6.667 | 5.333 | 3 | 4.333 | 11 | 6 | 9 | 9.333 | 11.667 |
| **NE China Wind** | MASE (raw) | **0.846** | 0.864 | 0.871 | 0.887 | 1.232 | 0.848 | 0.869 | 0.865 | 0.862 | 0.912 | 0.851 | 0.946 |
| | CRPS (raw) | 0.414 | **0.410** | 0.431 | 0.428 | 0.556 | 0.414 | 0.411 | 0.431 | 0.420 | 0.426 | 0.425 | 0.460 |
| | MASE (norm.) | 0.715 | 0.722 | 0.731 | 0.742 | 1.084 | **0.712** | 0.727 | 0.733 | 0.724 | 0.765 | 0.716 | 0.794 |
| | CRPS (norm.) | 0.618 | **0.613** | 0.643 | 0.639 | 0.839 | 0.620 | 0.614 | 0.643 | 0.626 | 0.635 | 0.636 | 0.689 |
| | MASE (rank) | 3.333 | 4.333 | 6.333 | 7.667 | 13 | **2.333** | 5.667 | 6.333 | 5.333 | 10 | 3.667 | 11 |
| | CRPS (rank) | 3.667 | **2** | 8.333 | 8 | 12 | 4 | 3 | 8.333 | 4.333 | 6.333 | 7 | 11 |
| **Australia Solar** | MASE (raw) | 0.606 | 0.600 | 0.594 | 0.620 | **0.560** | 0.585 | 0.604 | 0.625 | 0.650 | 0.584 | 0.666 | 0.663 |
| | CRPS (raw) | 0.232 | 0.231 | 0.233 | 0.253 | **0.229** | 0.236 | 0.239 | 0.268 | 0.243 | 0.245 | 0.247 | 0.259 |
| | MASE (norm.) | 0.854 | 0.844 | 0.837 | 0.873 | **0.789** | 0.824 | 0.851 | 0.881 | 0.916 | 0.823 | 0.939 | 0.934 |
| | CRPS (norm.) | 0.741 | 0.740 | 0.746 | 0.811 | **0.732** | 0.754 | 0.763 | 0.858 | 0.775 | 0.785 | 0.793 | 0.829 |
| | MASE (rank) | 6.333 | 5.667 | 5 | 7.667 | **1.333** | 3.333 | 5 | 8.667 | 10 | 2.333 | 11.333 | 11.333 |
| | CRPS (rank) | 2.667 | 3 | 3.333 | 10.333 | **2** | 4.333 | 6.333 | 12 | 7 | 7.667 | 8.667 | 10.667 |
| **EPF Electricity Price** | MASE (raw) | 1.221 | 1.189 | 1.272 | 1.207 | 1.474 | **1.163** | 1.229 | 1.206 | 1.232 | 1.235 | 1.243 | 1.289 |
| | CRPS (raw) | 0.149 | 0.147 | 0.156 | 0.150 | 0.184 | **0.144** | 0.156 | 0.159 | 0.153 | 0.154 | 0.151 | 0.158 |
| | MASE (norm.) | 0.688 | 0.669 | 0.715 | 0.678 | 0.836 | **0.655** | 0.692 | 0.680 | 0.693 | 0.694 | 0.701 | 0.726 |
| | CRPS (norm.) | 0.637 | 0.629 | 0.665 | 0.643 | 0.793 | **0.618** | 0.671 | 0.683 | 0.654 | 0.656 | 0.647 | 0.677 |
| | MASE (rank) | 5 | 2.333 | 10.333 | 4 | 12 | **1** | 6.667 | 4.667 | 6.333 | 6.667 | 8.333 | 10.667 |
| | CRPS (rank) | 4 | 2 | 8.667 | 4 | 12 | **1** | 8 | 9.667 | 6.333 | 6.333 | 6.333 | 9.667 |
| **OpenElectricity NEM** | MASE (raw) | 0.582 | **0.564** | 0.618 | 0.609 | 0.825 | 0.685 | 0.583 | 0.638 | 0.743 | 0.691 | 0.814 | 1.060 |
| | CRPS (raw) | 0.174 | **0.158** | 0.191 | 0.183 | 0.240 | 0.185 | 0.165 | 0.193 | 0.186 | 0.183 | 0.205 | 0.238 |
| | MASE (norm.) | 0.538 | **0.522** | 0.574 | 0.565 | 0.790 | 0.627 | 0.538 | 0.587 | 0.675 | 0.638 | 0.748 | 0.972 |
| | CRPS (norm.) | 0.507 | **0.463** | 0.555 | 0.532 | 0.716 | 0.538 | 0.480 | 0.563 | 0.544 | 0.537 | 0.607 | 0.695 |
| | MASE (rank) | 2.667 | **1** | 5.667 | 4.333 | 9.667 | 7.667 | 2.333 | 5.333 | 9 | 8 | 11 | 12.333 |
| | CRPS (rank) | 3 | **1** | 7.667 | 5.667 | 9.333 | 6.333 | 2 | 9.333 | 6.667 | 5.667 | 9.667 | 12 |
| **EWELD Load** | MASE (raw) | 0.667 | 0.659 | 0.686 | 0.677 | 0.776 | **0.653** | 0.696 | 0.667 | 0.669 | 0.674 | 0.853 | 0.994 |
| | CRPS (raw) | 0.226 | 0.210 | 0.276 | 0.228 | 0.382 | **0.208** | 0.209 | 0.329 | 0.280 | 0.245 | 0.294 | 0.528 |
| | MASE (norm.) | 0.673 | 0.662 | 0.695 | 0.682 | 0.789 | **0.660** | 0.702 | 0.673 | 0.678 | 0.680 | 0.864 | 1.009 |
| | CRPS (norm.) | 0.335 | 0.316 | 0.390 | 0.342 | 0.485 | **0.312** | 0.319 | 0.436 | 0.391 | 0.358 | 0.439 | 0.685 |
| | MASE (rank) | 4.333 | 3 | 6.667 | 6 | 9.667 | **2** | 9 | 4 | 4.667 | 5.667 | 11 | 12.667 |
| | CRPS (rank) | 3.667 | 2 | 6.667 | 6.333 | 10 | **1.667** | 5.333 | 9 | 6.333 | 5.333 | 9.667 | 12 |
| **SG Carpark** | MASE (raw) | 0.579 | **0.529** | 0.625 | 0.588 | 0.644 | 0.641 | 0.804 | 0.597 | 0.635 | 0.648 | 1.021 | 0.923 |
| | CRPS (raw) | 0.040 | **0.036** | 0.045 | 0.041 | 0.045 | 0.043 | 0.052 | 0.044 | 0.046 | 0.045 | 0.067 | 0.067 |
| | MASE (norm.) | 0.520 | **0.477** | 0.566 | 0.527 | 0.591 | 0.575 | 0.678 | 0.532 | 0.569 | 0.569 | 0.936 | 0.818 |
| | CRPS (norm.) | 0.465 | **0.422** | 0.532 | 0.474 | 0.541 | 0.507 | 0.586 | 0.513 | 0.531 | 0.514 | 0.797 | 0.773 |
| | MASE (rank) | 2.667 | **1** | 6.667 | 4 | 6.667 | 7.333 | 9.667 | 4 | 7 | 6.333 | 11.667 | 11.333 |
| | CRPS (rank) | 2.333 | **1** | 8.333 | 3.333 | 6.667 | 5.667 | 9 | 6 | 7.333 | 5.667 | 11.333 | 11.333 |
| **Finland Traffic** | MASE (raw) | 0.522 | **0.510** | 0.540 | 0.525 | 0.560 | 0.608 | 0.594 | 0.563 | 0.596 | 0.559 | 0.895 | 0.639 |
| | CRPS (raw) | 0.167 | **0.163** | 0.188 | **0.163** | 0.175 | 0.180 | 0.181 | 0.188 | 0.200 | 0.166 | 0.286 | 0.197 |
| | MASE (norm.) | 0.555 | **0.542** | 0.576 | 0.558 | 0.600 | 0.647 | 0.633 | 0.590 | 0.636 | 0.595 | 0.946 | 0.682 |
| | CRPS (norm.) | 0.463 | 0.452 | 0.515 | **0.451** | 0.481 | 0.499 | 0.502 | 0.522 | 0.550 | 0.461 | 0.787 | 0.547 |
| | MASE (rank) | 2.667 | **1** | 4.667 | 3.333 | 6 | 8.667 | 8 | 6 | 8.333 | 6.333 | 12.333 | 11 |
| | CRPS (rank) | 3 | **1.667** | 7.333 | 2.667 | 5.333 | 7 | 6.667 | 8 | 10 | 4.333 | 12 | 10 |
| **Port Activity/D** | MASE (raw) | 0.660 | **0.659** | 0.663 | 0.660 | 0.666 | 0.660 | 0.665 | 0.675 | 0.662 | 0.662 | 0.667 | 0.663 |
| | CRPS (raw) | 0.591 | 0.599 | 0.619 | **0.588** | 0.619 | 0.589 | 0.594 | 0.699 | 0.596 | 0.600 | 0.601 | 0.595 |
| | MASE (norm.) | 0.575 | **0.574** | 0.578 | 0.575 | 0.580 | 0.576 | 0.580 | 0.589 | 0.577 | 0.577 | 0.581 | 0.578 |
| | CRPS (norm.) | 0.176 | 0.178 | 0.184 | **0.175** | 0.184 | **0.175** | 0.177 | 0.208 | 0.177 | 0.178 | 0.179 | 0.177 |
| | MASE (rank) | 3 | **1** | 8 | 2 | 10 | 4 | 9 | 12 | 5 | 6 | 11 | 7 |
| | CRPS (rank) | 3 | 7 | 10 | **1** | 11 | 2 | 4 | 12 | 6 | 8 | 9 | 5 |
| **Port Activity/W** | MASE (raw) | 0.770 | 0.779 | 0.785 | 0.771 | 0.773 | 0.775 | 0.787 | 0.800 | **0.765** | 0.783 | 0.788 | 0.794 |
| | CRPS (raw) | 0.259 | 0.260 | 0.266 | 0.258 | 0.260 | 0.261 | 0.262 | 0.274 | **0.255** | 0.267 | 0.266 | 0.265 |
| | MASE (norm.) | 0.748 | 0.757 | 0.763 | 0.749 | 0.751 | 0.753 | 0.764 | 0.777 | **0.742** | 0.761 | 0.765 | 0.771 |
| | CRPS (norm.) | 0.524 | 0.527 | 0.539 | 0.522 | 0.527 | 0.529 | 0.530 | 0.554 | **0.517** | 0.541 | 0.539 | 0.537 |
| | MASE (rank) | 2 | 6 | 8 | 3 | 4 | 5 | 9 | 12 | **1** | 7 | 10 | 11 |
| | CRPS (rank) | 3 | 5 | 10 | 2 | 4 | 6 | 7 | 12 | **1** | 7 | 11 | 8 |
| **ECDC COVID/D** | MASE (raw) | 3.364 | 3.229 | 3.943 | 3.392 | 3.441 | **2.916** | 3.061 | 3.176 | 3.423 | 3.499 | 4.267 | 3.643 |
| | CRPS (raw) | 0.297 | 0.308 | 0.568 | **0.282** | 0.351 | 0.298 | 0.286 | 0.315 | 0.472 | 0.310 | 0.343 | 0.442 |
| | MASE (norm.) | 0.940 | 0.902 | 1.102 | 0.948 | 0.962 | **0.815** | 0.856 | 0.888 | 0.957 | 0.978 | 1.193 | 1.018 |
| | CRPS (norm.) | 0.479 | 0.496 | 0.916 | **0.456** | 0.566 | 0.481 | 0.462 | 0.509 | 0.762 | 0.501 | 0.552 | 0.713 |
| | MASE (rank) | 5 | 4 | 12 | 6 | 8 | **1** | 2 | 3 | 7 | 9 | 13 | 11 |
| | CRPS (rank) | 3 | 5 | 12 | **1** | 9 | 4 | 2 | 7 | 11 | 6 | 8 | 10 |
| **ECDC COVID/W** | MASE (raw) | 1.426 | 1.441 | 2.337 | 1.470 | 1.451 | 1.312 | 1.501 | **1.302** | 1.884 | 1.340 | 1.633 | 2.250 |
| | CRPS (raw) | 0.621 | 0.547 | 0.883 | 0.591 | 0.567 | 0.541 | 0.589 | **0.423** | 0.661 | 0.493 | 0.609 | 0.833 |
| | MASE (norm.) | 0.878 | 0.887 | 1.439 | 0.905 | 0.893 | 0.808 | 0.924 | **0.801** | 1.160 | 0.825 | 1.005 | 1.385 |
| | CRPS (norm.) | 0.780 | 0.688 | 1.110 | 0.743 | 0.713 | 0.680 | 0.740 | **0.532** | 0.831 | 0.620 | 0.765 | 1.047 |

*Table 6.* Full Results of Per Dataset (Continued).

| Dataset | Metric | TimesFM 2.5 | Chronos-2 | Kairos | Moirai 2.0 | VisionTS++ | TiRex | Toto | Sundial | TimesFM 2.0 | Chronos-bolt | Moirai | TimesFM 1.0 |
|---|---|---|---|---|---|---|---|---|---|---|---|---|---|
|  | MASE (rank) | 4 | 5 | 13 | 7 | 6 | 2 | 8 | **1** | 11 | 3 | 10 | 12 |
|  | CRPS (rank) | 9 | 4 | 13 | 7 | 5 | 3 | 6 | **1** | 10 | 2 | 8 | 12 |
| Global_Influenza | MASE (raw) | **2.453** | 2.702 | 2.531 | 2.614 | 2.699 | 2.807 | 2.641 | 3.196 | 2.507 | 2.902 | 2.669 | 2.713 |
|  | CRPS (raw) | 0.374 | 0.364 | 0.395 | **0.344** | 0.438 | 0.390 | 0.397 | 0.430 | 0.358 | 0.451 | 0.351 | 0.431 |
|  | MASE (norm.) | **0.567** | 0.624 | 0.585 | 0.604 | 0.624 | 0.649 | 0.610 | 0.739 | 0.579 | 0.671 | 0.617 | 0.627 |
|  | CRPS (norm.) | 0.281 | 0.273 | 0.296 | **0.258** | 0.329 | 0.292 | 0.298 | 0.322 | 0.268 | 0.338 | 0.263 | 0.323 |
|  | MASE (rank) | **1** | 8 | 3 | 4 | 7 | 10 | 5 | 12 | 2 | 11 | 6 | 9 |
|  | CRPS (rank) | 5 | 4 | 7 | **1** | 11 | 6 | 8 | 9 | 3 | 12 | 2 | 10 |
| Crypto | MASE (raw) | 6.559 | 6.229 | 7.779 | 6.491 | 11.111 | **6.205** | 6.936 | 6.437 | 6.831 | 6.651 | 7.171 | 6.435 |
|  | CRPS (raw) | 0.080 | **0.075** | 0.084 | 0.078 | 0.132 | 0.076 | 0.085 | 0.080 | 0.080 | 0.081 | 0.082 | 0.080 |
|  | MASE (norm.) | 1.041 | 0.989 | 1.235 | 1.030 | 1.764 | **0.985** | 1.101 | 1.022 | 1.084 | 1.056 | 1.138 | 1.022 |
|  | CRPS (norm.) | 0.961 | **0.898** | 1.005 | 0.939 | 1.584 | 0.915 | 1.019 | 0.961 | 0.965 | 0.975 | 0.981 | 0.957 |
|  | MASE (rank) | 7 | 2 | 12 | 6 | 13 | **1** | 10 | 5 | 9 | 8 | 11 | 4 |
|  | CRPS (rank) | 6 | **1** | 11 | 3 | 13 | 2 | 12 | 5 | 7 | 8 | 9 | 4 |
| US Term Structure | MASE (raw) | **1.458** | 1.460 | 1.602 | 1.542 | 3.484 | 1.479 | 1.563 | 1.510 | 1.506 | 1.535 | 1.712 | 1.542 |
|  | CRPS (raw) | 0.210 | **0.209** | 0.224 | 0.215 | 0.406 | 0.210 | 0.221 | 0.218 | 0.214 | 0.221 | 0.212 | 0.218 |
|  | MASE (norm.) | **0.837** | 0.839 | 0.920 | 0.886 | 2.001 | 0.849 | 0.897 | 0.867 | 0.865 | 0.882 | 0.983 | 0.886 |
|  | CRPS (norm.) | 0.846 | **0.842** | 0.903 | 0.865 | 1.634 | 0.844 | 0.889 | 0.876 | 0.863 | 0.891 | 0.853 | 0.876 |
|  | MASE (rank) | **1** | 2 | 10 | 7 | 13 | 3 | 9 | 5 | 4 | 6 | 11 | 8 |
|  | CRPS (rank) | 3 | **1** | 11 | 6 | 13 | 2 | 9 | 8 | 5 | 10 | 4 | 7 |
| Oil Price | MASE (raw) | 1.307 | 1.300 | 1.343 | 1.383 | 2.290 | 1.333 | 1.319 | **1.293** | 1.344 | 1.376 | 1.662 | 1.476 |
|  | CRPS (raw) | **0.045** | 0.045 | 0.046 | 0.047 | 0.088 | 0.046 | **0.045** | 0.046 | 0.046 | 0.046 | 0.057 | 0.050 |
|  | MASE (norm.) | 0.871 | 0.866 | 0.895 | 0.921 | 1.526 | 0.889 | 0.879 | **0.862** | 0.896 | 0.917 | 1.107 | 0.984 |
|  | CRPS (norm.) | **0.836** | 0.836 | 0.862 | 0.882 | 1.651 | 0.859 | 0.849 | 0.869 | 0.854 | 0.863 | 1.070 | 0.939 |
|  | MASE (rank) | 3 | 2 | 6 | 9 | 13 | 5 | 4 | **1** | 7 | 8 | 12 | 10 |
|  | CRPS (rank) | 2 | **1** | 6 | 9 | 13 | 5 | 3 | 8 | 4 | 7 | 12 | 10 |
| Job Claims | MASE (raw) | 2.732 | 3.144 | 3.381 | 2.799 | 2.978 | 2.431 | **2.392** | 5.572 | 2.899 | 4.147 | 2.641 | 3.432 |
|  | CRPS (raw) | 0.018 | 0.018 | 0.021 | 0.017 | 0.018 | 0.016 | **0.015** | 0.032 | 0.018 | 0.022 | 0.017 | 0.021 |
|  | MASE (norm.) | 0.960 | 1.105 | 1.188 | 0.984 | 1.047 | 0.854 | **0.841** | 1.958 | 1.019 | 1.458 | 0.928 | 1.206 |
|  | CRPS (norm.) | 0.934 | 0.947 | 1.063 | 0.855 | 0.948 | 0.848 | **0.782** | 1.625 | 0.921 | 1.145 | 0.871 | 1.074 |
|  | MASE (rank) | 4 | 9 | 10 | 5 | 8 | 2 | **1** | 13 | 7 | 12 | 3 | 11 |
|  | CRPS (rank) | 6 | 7 | 10 | 3 | 8 | 2 | **1** | 13 | 5 | 12 | 4 | 11 |
| Uncertainty-1M | MASE (raw) | 0.376 | 0.348 | **0.339** | 0.374 | 0.463 | 0.371 | 0.382 | 0.422 | 0.448 | 0.415 | 0.420 | 0.551 |
|  | CRPS (raw) | 0.031 | **0.027** | 0.028 | 0.029 | 0.040 | 0.027 | 0.028 | 0.035 | 0.038 | 0.033 | 0.032 | 0.043 |
|  | MASE (norm.) | 0.355 | 0.329 | **0.321** | 0.354 | 0.438 | 0.351 | 0.361 | 0.399 | 0.424 | 0.392 | 0.397 | 0.520 |
|  | CRPS (norm.) | 0.392 | **0.340** | 0.356 | 0.363 | 0.505 | 0.342 | 0.356 | 0.443 | 0.473 | 0.414 | 0.402 | 0.546 |
|  | MASE (rank) | 5 | 2 | **1** | 4 | 11 | 3 | 6 | 9 | 10 | 7 | 8 | 12 |
|  | CRPS (rank) | 6 | **1** | 3 | 5 | 11 | 2 | 4 | 9 | 10 | 8 | 7 | 12 |
| Housing Inventory | MASE (raw) | 0.428 | 0.530 | 0.512 | 0.526 | 0.469 | 0.483 | **0.412** | 1.035 | 0.566 | 0.528 | 0.543 | 0.496 |
|  | CRPS (raw) | 0.043 | 0.053 | 0.047 | 0.044 | 0.045 | 0.044 | **0.042** | 0.100 | 0.052 | 0.054 | 0.049 | 0.055 |
|  | MASE (norm.) | 0.595 | 0.737 | 0.712 | 0.732 | 0.652 | 0.672 | **0.573** | 1.440 | 0.787 | 0.735 | 0.756 | 0.690 |
|  | CRPS (norm.) | 0.644 | 0.791 | 0.701 | 0.663 | 0.671 | 0.658 | **0.635** | 1.499 | 0.787 | 0.806 | 0.733 | 0.821 |
|  | MASE (rank) | 2 | 9 | 6 | 7 | 3 | 4 | **1** | 13 | 11 | 8 | 10 | 5 |
|  | CRPS (rank) | 2 | 9 | 6 | 4 | 5 | 3 | **1** | 13 | 8 | 10 | 7 | 11 |
| JOLTS | MASE (raw) | 1.017 | **0.904** | 1.244 | 1.041 | 1.205 | 0.995 | 1.071 | 1.546 | 1.109 | 1.055 | 1.501 | 1.480 |
|  | CRPS (raw) | 0.063 | **0.058** | 0.075 | 0.065 | 0.076 | 0.062 | 0.068 | 0.094 | 0.069 | 0.065 | 0.090 | 0.093 |
|  | MASE (norm.) | 0.622 | **0.554** | 0.761 | 0.637 | 0.738 | 0.609 | 0.656 | 0.946 | 0.679 | 0.646 | 0.919 | 0.906 |
|  | CRPS (norm.) | 0.628 | **0.572** | 0.742 | 0.642 | 0.750 | 0.614 | 0.672 | 0.931 | 0.682 | 0.645 | 0.897 | 0.924 |
|  | MASE (rank) | 3 | **1** | 9 | 4 | 8 | 2 | 6 | 12 | 7 | 5 | 11 | 10 |
|  | CRPS (rank) | 3 | **1** | 8 | 4 | 9 | 2 | 6 | 12 | 7 | 5 | 10 | 11 |
| US Labor | MASE (raw) | 0.779 | 1.001 | 1.056 | 0.811 | 0.908 | 0.790 | **0.772** | 1.452 | 0.833 | 0.946 | 0.903 | 0.829 |
|  | CRPS (raw) | **0.042** | 0.058 | 0.060 | 0.046 | 0.053 | 0.044 | 0.044 | 0.062 | 0.045 | 0.056 | 0.055 | 0.044 |
|  | MASE (norm.) | 0.436 | 0.561 | 0.592 | 0.454 | 0.509 | 0.443 | **0.433** | 0.813 | 0.466 | 0.530 | 0.506 | 0.464 |
|  | CRPS (norm.) | **0.380** | 0.517 | 0.543 | 0.409 | 0.476 | 0.393 | 0.397 | 0.553 | 0.400 | 0.504 | 0.497 | 0.391 |
|  | MASE (rank) | 2 | 10 | 11 | 4 | 8 | 3 | **1** | 12 | 6 | 9 | 7 | 5 |
|  | CRPS (rank) | **1** | 10 | 11 | 6 | 7 | 3 | 4 | 12 | 5 | 9 | 8 | 2 |
| Vehicle Supply | MASE (raw) | 0.798 | **0.709** | 0.884 | 0.757 | 1.137 | 0.792 | 0.764 | 0.717 | 0.710 | 0.811 | 0.791 | 0.973 |
|  | CRPS (raw) | 0.188 | 0.167 | 0.236 | 0.190 | 0.327 | 0.193 | 0.172 | **0.157** | **0.157** | 0.195 | 0.174 | 0.278 |
|  | MASE (norm.) | 0.719 | **0.639** | 0.796 | 0.682 | 1.024 | 0.714 | 0.688 | 0.646 | **0.639** | 0.730 | 0.712 | 0.877 |
|  | CRPS (norm.) | 0.541 | 0.482 | 0.679 | 0.546 | 0.941 | 0.556 | 0.496 | **0.452** | 0.452 | 0.560 | 0.501 | 0.800 |
|  | MASE (rank) | 8 | **1** | 10 | 4 | 13 | 7 | 5 | 3 | 2 | 9 | 6 | 11 |
|  | CRPS (rank) | 6 | 3 | 10 | 7 | 12 | 8 | 4 | **1** | 2 | 9 | 5 | 11 |
| Auto Production-SF | MASE (raw) | 0.949 | **0.604** | 1.228 | 0.905 | 0.617 | 0.746 | 1.014 | 1.403 | 0.992 | 1.032 | 1.968 | 1.330 |
|  | CRPS (raw) | 0.029 | **0.019** | 0.038 | 0.028 | 0.020 | 0.023 | 0.031 | 0.046 | 0.030 | 0.032 | 0.061 | 0.042 |
|  | MASE (norm.) | 0.971 | **0.618** | 1.257 | 0.926 | 0.631 | 0.763 | 1.038 | 1.435 | 1.015 | 1.056 | 2.014 | 1.361 |
|  | CRPS (norm.) | 0.898 | **0.597** | 1.180 | 0.882 | 0.621 | 0.720 | 0.987 | 1.451 | 0.957 | 1 | 1.922 | 1.307 |
|  | MASE (rank) | 5 | **1** | 10 | 4 | 2 | 3 | 8 | 12 | 7 | 9 | 13 | 11 |
|  | CRPS (rank) | 5 | **1** | 10 | 4 | 2 | 3 | 7 | 12 | 6 | 9 | 13 | 11 |
| Commodity Production | MASE (raw) | 0.914 | 0.905 | 0.928 | 0.950 | 0.914 | 0.914 | 0.949 | 1.205 | **0.794** | 0.939 | 0.960 | 1.027 |
|  | CRPS (raw) | 0.124 | 0.124 | 0.123 | 0.128 | 0.124 | 0.124 | 0.128 | 0.151 | **0.109** | 0.124 | 0.128 | 0.138 |
|  | MASE (norm.) | 0.734 | 0.726 | 0.744 | 0.763 | 0.734 | 0.733 | 0.761 | 0.967 | **0.637** | 0.753 | 0.771 | 0.824 |
|  | CRPS (norm.) | 0.698 | 0.695 | 0.693 | 0.717 | 0.699 | 0.695 | 0.721 | 0.851 | **0.613** | 0.699 | 0.720 | 0.776 |
|  | MASE (rank) | 5 | 2 | 6 | 9 | 4 | 3 | 8 | 12 | **1** | 7 | 10 | 11 |
|  | CRPS (rank) | 5 | 3 | 2 | 8 | 7 | 4 | 10 | 12 | **1** | 6 | 9 | 11 |
| Commodity Import | MASE (raw) | 1.567 | 1.459 | 1.554 | **1.453** | 1.507 | 1.498 | 1.499 | 1.813 | 1.678 | 1.532 | 1.642 | 1.751 |
|  | CRPS (raw) | 0.214 | 0.238 | 0.213 | 0.190 | 0.223 | 0.226 | 0.201 | 0.199 | **0.185** | 0.214 | 0.221 | 0.197 |
|  | MASE (norm.) | 0.839 | 0.782 | 0.832 | **0.778** | 0.807 | 0.802 | 0.803 | 0.971 | 0.899 | 0.821 | 0.880 | 0.938 |

*Table 6.* Full Results of Per Dataset (Continued).

| Dataset | Metric | TimesFM 2.5 | Chronos-2 | Kairos | Moirai 2.0 | VisionTS++ | TiRex | Toto | Sundial | TimesFM 2.0 | Chronos-bolt | Moirai | TimesFM 1.0 |
|---|---|---|---|---|---|---|---|---|---|---|---|---|---|
| | CRPS (norm.) | 0.643 | 0.714 | 0.641 | 0.571 | 0.672 | 0.678 | 0.603 | 0.598 | **0.555** | 0.644 | 0.663 | 0.591 |
| | MASE (rank) | 8 | 2 | 7 | **1** | 5 | 3 | 4 | 12 | 10 | 6 | 9 | 11 |
| | CRPS (rank) | 7 | 12 | 6 | 2 | 10 | 11 | 5 | 4 | **1** | 8 | 9 | 3 |
| **WUI-Global** | MASE (raw) | 0.961 | **0.949** | 0.976 | 0.971 | 1.005 | 0.966 | 0.975 | 1.022 | 0.971 | 0.998 | 0.959 | 0.963 |
| | CRPS (raw) | **0.513** | 0.534 | 0.533 | 0.531 | 0.541 | 0.521 | 0.553 | 0.573 | 0.564 | 0.550 | 0.525 | **0.513** |
| | MASE (norm.) | 0.718 | **0.708** | 0.729 | 0.725 | 0.751 | 0.721 | 0.728 | 0.763 | 0.725 | 0.745 | 0.716 | 0.719 |
| | CRPS (norm.) | **0.623** | 0.647 | 0.646 | 0.644 | 0.656 | 0.631 | 0.671 | 0.695 | 0.683 | 0.667 | 0.636 | **0.623** |
| | MASE (rank) | 3 | 1 | 9 | 7 | 11 | 5 | 8 | 12 | 6 | 10 | 2 | 4 |
| | CRPS (rank) | 2 | 7 | 6 | 5 | 8 | 3 | 10 | 12 | 11 | 9 | 4 | **1** |
| **Global Price** | MASE (raw) | 1.416 | 1.427 | 1.512 | 1.478 | 1.555 | 1.496 | **1.412** | 1.748 | 1.497 | 1.568 | 1.464 | 1.482 |
| | CRPS (raw) | 0.132 | 0.133 | 0.141 | 0.140 | 0.144 | 0.144 | **0.131** | 0.172 | 0.148 | 0.146 | 0.139 | 0.136 |
| | MASE (norm.) | 0.777 | 0.783 | 0.829 | 0.810 | 0.853 | 0.820 | **0.774** | 0.958 | 0.821 | 0.860 | 0.803 | 0.812 |
| | CRPS (norm.) | 0.735 | 0.742 | 0.788 | 0.779 | 0.805 | 0.800 | **0.732** | 0.957 | 0.825 | 0.815 | 0.772 | 0.759 |
| | MASE (rank) | 2 | 3 | 9 | 5 | 10 | 7 | **1** | 12 | 8 | 11 | 4 | 6 |
| | CRPS (rank) | 2 | 3 | 7 | 6 | 9 | 8 | **1** | 12 | 11 | 10 | 5 | 4 |
| **Vehicle Sales** | MASE (raw) | 0.861 | 0.835 | 0.937 | 0.759 | 1.311 | 0.803 | **0.750** | 0.903 | 0.882 | 0.772 | 0.786 | 0.865 |
| | CRPS (raw) | 0.070 | 0.065 | 0.073 | 0.065 | 0.122 | 0.066 | **0.064** | 0.072 | 0.068 | 0.065 | 0.070 | 0.075 |
| | MASE (norm.) | 0.611 | 0.592 | 0.664 | 0.538 | 0.930 | 0.570 | **0.532** | 0.640 | 0.626 | 0.548 | 0.557 | 0.614 |
| | CRPS (norm.) | 0.610 | 0.570 | 0.640 | 0.570 | 1.062 | 0.573 | **0.559** | 0.630 | 0.594 | 0.566 | 0.612 | 0.652 |
| | MASE (rank) | 7 | 6 | 11 | 2 | 12 | 5 | **1** | 10 | 9 | 3 | 4 | 8 |
| | CRPS (rank) | 7 | 3 | 10 | 4 | 13 | 5 | **1** | 9 | 6 | 2 | 8 | 11 |
| **Online Retail II** | MASE (raw) | **0.590** | 0.609 | 0.652 | 0.630 | 0.733 | 0.609 | 0.700 | 0.765 | 0.643 | 0.620 | 0.671 | 0.653 |
| | CRPS (raw) | **0.234** | 0.242 | 0.255 | 0.247 | 0.272 | 0.244 | 0.274 | 0.302 | 0.253 | 0.248 | 0.268 | 0.258 |
| | MASE (norm.) | **0.276** | 0.285 | 0.305 | 0.294 | 0.343 | 0.284 | 0.327 | 0.357 | 0.300 | 0.289 | 0.313 | 0.305 |
| | CRPS (norm.) | **0.234** | 0.242 | 0.255 | 0.247 | 0.272 | 0.245 | 0.274 | 0.302 | 0.253 | 0.248 | 0.268 | 0.258 |
| | MASE (rank) | **1** | 3 | 7 | 5 | 11 | 2 | 10 | 12 | 6 | 4 | 9 | 8 |
| | CRPS (rank) | **1** | 2 | 7 | 4 | 10 | 3 | 11 | 12 | 6 | 5 | 9 | 8 |
| **Supply Chain-Customer** | MASE (raw) | 0.581 | **0.553** | 0.614 | 0.598 | 0.636 | 0.604 | 0.669 | 0.666 | 0.613 | 0.613 | 0.661 | 0.661 |
| | CRPS (raw) | 0.275 | **0.259** | 0.293 | 0.276 | 0.300 | 0.281 | 0.311 | 0.326 | 0.283 | 0.292 | 0.304 | 0.308 |
| | MASE (norm.) | 0.343 | **0.327** | 0.363 | 0.353 | 0.375 | 0.356 | 0.395 | 0.393 | 0.362 | 0.362 | 0.390 | 0.390 |
| | CRPS (norm.) | 0.258 | **0.242** | 0.274 | 0.258 | 0.281 | 0.263 | 0.291 | 0.305 | 0.265 | 0.273 | 0.284 | 0.288 |
| | MASE (rank) | 2 | 1 | 7 | 3 | 8 | 4 | 12 | 11 | 5 | 6 | 9 | 10 |
| | CRPS (rank) | 2 | 1 | 7 | 3 | 8 | 4 | 11 | 12 | 5 | 6 | 9 | 10 |
| **Supply Chain-Location** | MASE (raw) | 0.627 | **0.605** | 0.650 | 0.649 | 0.668 | 0.649 | 0.699 | 0.705 | 0.649 | 0.650 | 0.683 | 0.694 |
| | CRPS (raw) | 0.289 | **0.271** | 0.305 | 0.290 | 0.309 | 0.292 | 0.319 | 0.337 | 0.291 | 0.304 | 0.309 | 0.318 |
| | MASE (norm.) | 0.393 | **0.379** | 0.407 | 0.406 | 0.418 | 0.406 | 0.438 | 0.441 | 0.406 | 0.407 | 0.428 | 0.434 |
| | CRPS (norm.) | 0.273 | **0.257** | 0.288 | 0.274 | 0.292 | 0.276 | 0.302 | 0.319 | 0.275 | 0.287 | 0.293 | 0.300 |
| | MASE (rank) | 2 | 1 | 6 | 5 | 8 | 3 | 11 | 12 | 4 | 7 | 9 | 10 |
| | CRPS (rank) | 2 | 1 | 7 | 3 | 8 | 5 | 11 | 12 | 4 | 6 | 9 | 10 |
| **Azure2019-D** | MASE (raw) | 0.731 | **0.718** | 0.834 | 0.758 | 0.899 | 0.763 | 0.741 | 0.776 | 0.767 | 0.807 | 0.869 | 0.817 |
| | CRPS (raw) | 0.144 | **0.139** | 0.194 | 0.152 | 0.184 | 0.154 | 0.145 | 0.166 | 0.161 | 0.164 | 0.175 | 0.182 |
| | MASE (norm.) | 0.668 | **0.656** | 0.762 | 0.693 | 0.822 | 0.697 | 0.677 | 0.709 | 0.701 | 0.738 | 0.795 | 0.747 |
| | CRPS (norm.) | 0.492 | **0.474** | 0.662 | 0.518 | 0.630 | 0.528 | 0.494 | 0.568 | 0.550 | 0.561 | 0.599 | 0.621 |
| | MASE (rank) | 2 | 1 | 10 | 4 | 12 | 5 | 3 | 7 | 6 | 8 | 11 | 9 |
| | CRPS (rank) | 2 | 1 | 12 | 4 | 11 | 5 | 3 | 8 | 6 | 7 | 9 | 10 |
| **Azure2019-I** | MASE (raw) | 0.698 | **0.674** | 0.788 | 0.726 | 0.793 | 0.728 | 0.702 | 0.761 | 0.782 | 0.788 | 0.917 | 1.127 |
| | CRPS (raw) | 0.129 | **0.123** | 0.153 | 0.133 | 0.145 | 0.133 | 0.128 | 0.146 | 0.146 | 0.146 | 0.150 | 0.184 |
| | MASE (norm.) | 0.690 | **0.667** | 0.779 | 0.718 | 0.784 | 0.719 | 0.694 | 0.753 | 0.773 | 0.779 | 0.906 | 1.114 |
| | CRPS (norm.) | 0.565 | **0.537** | 0.673 | 0.584 | 0.637 | 0.585 | 0.563 | 0.639 | 0.641 | 0.640 | 0.658 | 0.805 |
| | MASE (rank) | 2 | 1 | 9 | 4 | 10 | 5 | 3 | 6 | 7 | 8 | 11 | 13 |
| | CRPS (rank) | 3 | 1 | 11 | 4 | 6 | 5 | 2 | 7 | 9 | 8 | 10 | 12 |
| **Azure2019-U** | MASE (raw) | 0.693 | **0.661** | 0.763 | 0.677 | 0.849 | 0.669 | 0.684 | 0.783 | 0.721 | 0.773 | 0.900 | 0.852 |
| | CRPS (raw) | 0.161 | **0.147** | 0.187 | 0.148 | 0.199 | 0.150 | 0.153 | 0.168 | 0.163 | 0.179 | 0.189 | 0.182 |
| | MASE (norm.) | 0.602 | **0.574** | 0.662 | 0.588 | 0.737 | 0.581 | 0.594 | 0.680 | 0.626 | 0.671 | 0.781 | 0.740 |
| | CRPS (norm.) | 0.283 | **0.257** | 0.328 | 0.260 | 0.349 | 0.263 | 0.269 | 0.295 | 0.286 | 0.314 | 0.331 | 0.320 |
| | MASE (rank) | 5 | 1 | 7 | 3 | 10 | 2 | 4 | 9 | 6 | 8 | 12 | 11 |
| | CRPS (rank) | 5 | 1 | 10 | 2 | 12 | 3 | 4 | 7 | 6 | 8 | 11 | 9 |
| **Smart Manufacturing** | MASE (raw) | 0.713 | **0.711** | 0.713 | 0.714 | 0.716 | 0.716 | 0.720 | 0.734 | 0.717 | 0.715 | 0.728 | 0.717 |
| | CRPS (raw) | 0.167 | **0.166** | 0.175 | 0.167 | 0.167 | 0.167 | 0.169 | 0.188 | 0.178 | 0.170 | 0.172 | 0.180 |
| | MASE (norm.) | 0.685 | **0.684** | 0.685 | 0.686 | 0.688 | 0.688 | 0.692 | 0.706 | 0.689 | 0.687 | 0.700 | 0.689 |
| | CRPS (norm.) | 0.610 | **0.608** | 0.641 | 0.613 | 0.612 | 0.611 | 0.653 | 0.689 | 0.623 | 0.629 | 0.657 | 0.657 |
| | MASE (rank) | 2.333 | 1 | 3.333 | 3.667 | 7 | 7 | 9.667 | 11.667 | 6.333 | 6.333 | 11.333 | 8.333 |
| | CRPS (rank) | 3.667 | 1 | 7.333 | 4.333 | 3.667 | 3.667 | 7 | 11.333 | 9.667 | 7 | 8.667 | 10.667 |
| **MetroPT-3** | MASE (raw) | 0.625 | **0.610** | 0.666 | 0.665 | 0.673 | 0.616 | 0.645 | 0.669 | 0.690 | 0.641 | 0.673 | 0.711 |
| | CRPS (raw) | 0.256 | **0.244** | 0.288 | 0.253 | 0.264 | 0.246 | 0.262 | 0.313 | 0.300 | 0.264 | 0.261 | 0.324 |
| | MASE (norm.) | 0.740 | **0.722** | 0.788 | 0.787 | 0.797 | 0.729 | 0.764 | 0.792 | 0.817 | 0.760 | 0.797 | 0.842 |
| | CRPS (norm.) | 0.677 | **0.647** | 0.762 | 0.670 | 0.698 | 0.652 | 0.693 | 0.829 | 0.790 | 0.698 | 0.691 | 0.854 |
| | MASE (rank) | 2.667 | 1.333 | 8 | 8 | 8.667 | 2.667 | 5.333 | 8.333 | 10 | 4.667 | 7 | 11.333 |
| | CRPS (rank) | 3.667 | 1 | 8.667 | 4 | 7 | 2 | 6.667 | 11 | 9 | 6.667 | 6.667 | 11.667 |

# F. Visualization

Our platform features an interactive visualization tool for inspecting predictions across different test windows. Key functionalities include zooming and the display of probabilistic quantiles to assess forecast confidence. The visualization

employs a distinct color schema for clarity: the blue region indicates the training history, the yellow region marks the overall test set, and the red region highlights the specific target window currently under evaluation. The following predictions are all from TimesFM 2.5.

To provide a comprehensive analysis of the forecasting behaviors, each case study is presented in two granularities: a *Global* view (a) that illustrates the complete time series, and a *Local* view (b) that zooms into the specific target window to detail the fine-grained prediction alignment and confidence intervals. The following curated examples evaluate TimesFM 2.5 across various distinct time-series phenomena. In all subsequent figures, the title denotes the specific source of the sequence, strictly following the format of `Dataset – Series – Variate`.

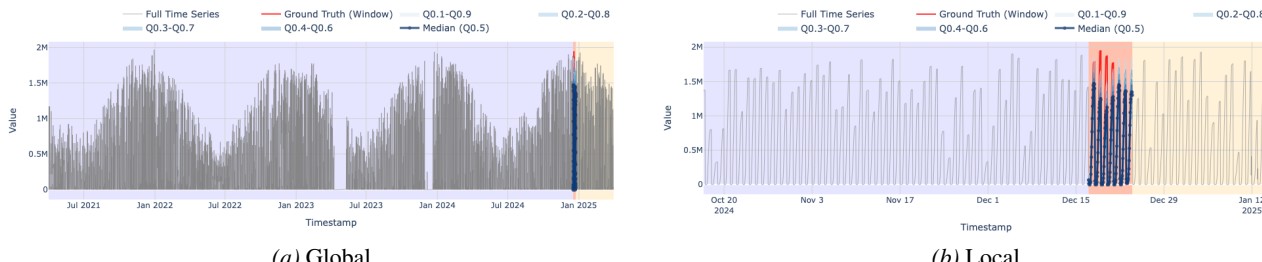

*(a)* Global          *(b)* Local

*Figure 11.* Australia Solar - item0 - solar_63726. This series exhibits distinct multiple seasonalities. The model accurately captures the fine-grained daily periodic spikes while aligning with the broader annual trend.

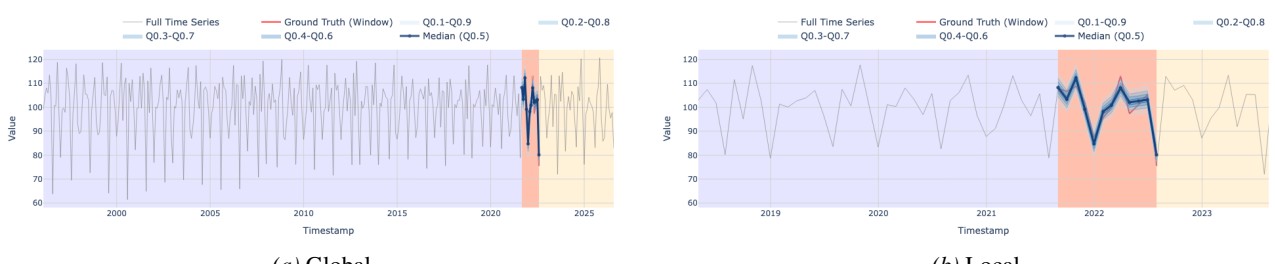

*(a)* Global          *(b)* Local

*Figure 12.* Auto Production SF - RSF AutoProduction. This stationary series exhibits highly regular, sharp periodic dips. The model accurately captures both the timing and amplitude of these fluctuations.

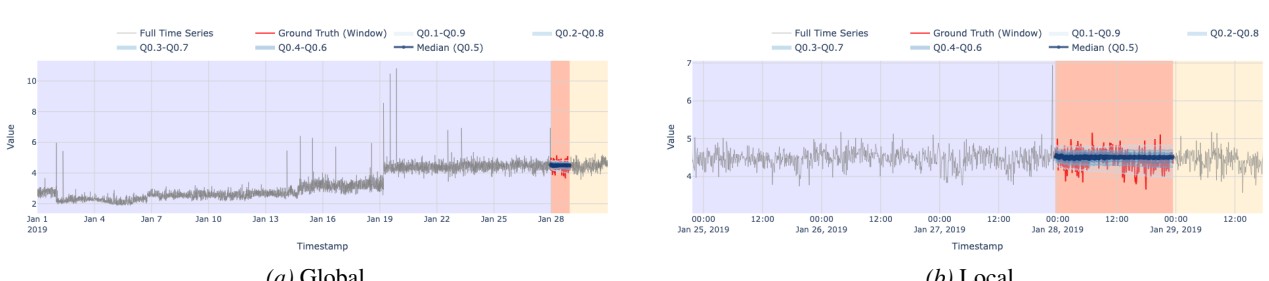

*(a)* Global          *(b)* Local

*Figure 13.* azure2019 I - item_1 - min_cpu. This series is characterized by abrupt structural level shifts and localized high-frequency fluctuations. The global view highlights the model's ability to adapt to the baseline. In the local view, the model outputs a stable, smoothed mean forecast, treating the fluctuations as irreducible noise rather than learnable patterns.

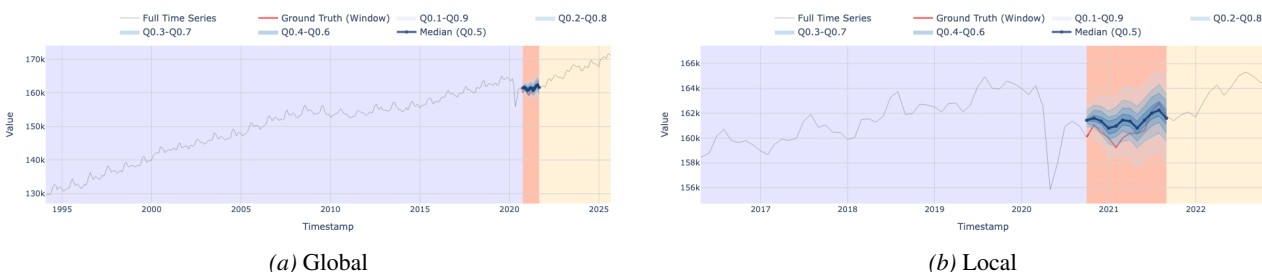

*(a)* Global

*(b)* Local

*Figure 14.* US Labor - item0 - CivilianLaborForceLevel. This series is characterized by a prominent long-term upward trend combined with regular seasonality, alongside a significant historical shock (visible around 2020). The model successfully captures both the overarching growth trajectory and the fine-grained local seasonal fluctuations within the target window.

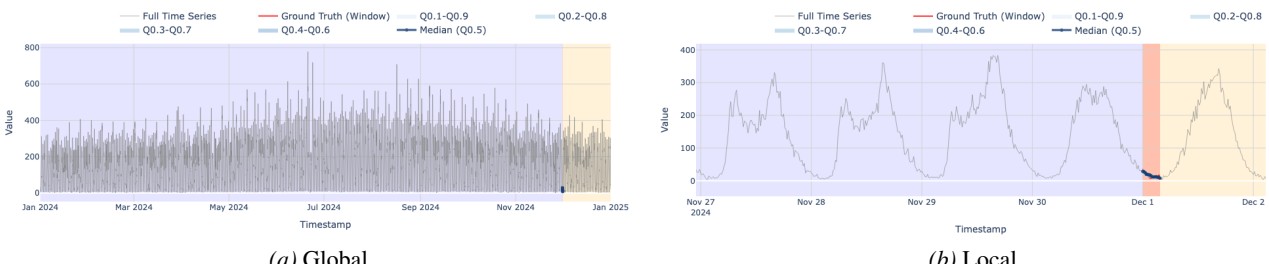

*(a)* Global

*(b)* Local

*Figure 15.* Finland Traffic - volume. While the global view displays dense and frequent spikes, the local view reveals clear seasonality that the model successfully captures and predicts.

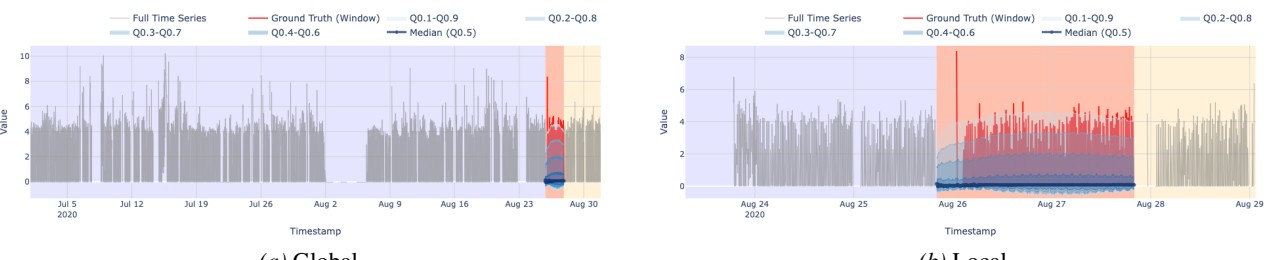

*(a)* Global

*(b)* Local

*Figure 16.* MetroPT-3 - item0 - TP2. To minimize average error metrics, the model favors a smooth forecast, failing to capture the distinct spike structure of the series.

## G. Limitations and Future Directions

**Pattern Granularity and Data Sparsity.** In our current pattern-level evaluation, we employ a median-based binary threshold to categorize feature strengths. While we explored finer-grained quantization (e.g., dividing features into 5 quintiles, representing 5 levels from Strong to Weak), we encountered a combinatorial explosion challenge: the number of subgroups grows exponentially with the number of combined patterns. This leads to severe data sparsity in intersectional groups, rendering statistical analysis unreliable. Future work will address this by expanding the scale of variates in the benchmark and developing more sophisticated pattern retrieval mechanisms to enable fine-grained analysis without compromising statistical validity.

**Task-Aligned Evaluation Metrics.** Time series forecasting fundamentally serves downstream decision-making. However, standard general-purpose metrics like MASE and CRPS often fail to provide actionable insights specific to operational contexts. For instance, in financial scenarios, the directional accuracy (i.e., predicting the correct rise or fall to guide trading actions) is often more critical than minimizing the absolute magnitude of error. Furthermore, general metrics are known to be sensitive to data processing factors such as scaling and aggregation schemes (Brigato et al., 2026). As a task-centric benchmark, our next phase aims to align evaluation protocols with specific downstream requirements, moving beyond generic error minimization to provide metrics that reflect the true actionable utility of the forecasts in real-world applications.

