# OpenReview forum: "It's TIME: Towards the Next Generation of Time Series Forecasting Benchmarks"
_ICML.cc/2026/Conference — ICML 2026 regular_

### Official Review · Reviewer_Nyob · 2026-02-26

**Soundness:** 2
**Presentation:** 2
**Significance:** 3
**Originality:** 3
**Overall Recommendation:** 4
**Confidence:** 5

**Summary:**

This paper identifies critical issues with current benchmarks in the time series forecasting domain through profound insights. It significantly advances the field by introducing more novel and reasonable dataset configurations, evaluation metrics, and other innovations.

**Compliance With Llm Reviewing Policy:**

Affirmed.

**Final Justification:**

I increase my score, as the intrinsic merit of this work is significant and the authors' rebuttal has addressed the vast majority of my concerns.

**Key Questions For Authors:**

See in Strengths And Weaknesses part

**Limitations:**

yes

**Strengths And Weaknesses:**

The work presented in this paper is highly valuable, offering deep insights and a comprehensive benchmark framework. However, the evaluated models are strictly limited to the zero-shot paradigm of Time Series Foundation Models (TSFMs), which somewhat restricts the paper's broader impact. Given the considerable effort the researchers invested in curating such detailed datasets and designing a meticulous benchmark, it is a pity to limit the evaluation to such a narrow scope of models.

**Strengths**

The paper accurately points out profound and genuinely existing problems within the current field. It curates a large number of novel datasets and presents a comprehensive performance comparison across various TSFM models.

**Weaknesses & Suggestions**

1. As mentioned earlier, the evaluated models are restricted solely to the TSFM family. However, other lightweight models or Transformer-based architectures have also demonstrated excellent performance in recent years, while requiring significantly less computational overhead. One of the paper's conclusions left a deep impression on me: "This confirms that recent advancements in TSFMs represent genuine capability improvements rather than overfitting to the data bias in established benchmarks."  However, does this conclusion also hold true for other innovative, small-parameter time series forecasting models? For example:

[1] Yue, Wenzhen, et al. "Olinear: A linear model for time series forecasting in orthogonally transformed domain." arXiv preprint arXiv:2505.08550 (2025).

[2] Qiu, Xiangfei, et al. "Duet: Dual clustering enhanced multivariate time series forecasting." Proceedings of the 31st ACM SIGKDD Conference on Knowledge Discovery and Data Mining V. 1. 2025.

[3] Tire, Kutay, et al. "Retrieval augmented time series forecasting." arXiv preprint arXiv:2411.08249 (2024).

[4] Nie, Yuqi, et al. "A time series is worth 64 words: Long-term forecasting with transformers." arXiv preprint arXiv:2211.14730 (2022).

[5] Liu, Yong, et al. "Koopa: Learning non-stationary time series dynamics with koopman predictors." Advances in neural information processing systems 36 (2023): 12271-12290.

[6] Wang, Shiyu, et al. "Timemixer: Decomposable multiscale mixing for time series forecasting." arXiv preprint arXiv:2405.14616 (2024).

2. The non-overlapping forecasting framework adopted in the paper can significantly reduce experimental costs. However, could this potentially introduce bias into the experimental results? An intuitive conclusion is that forecasting error will naturally increase the further away the prediction is from the current time step.

3. This paper has the potential to become an important foundational work in the current research field. Therefore, the data cleaning process for the benchmark datasets needs to be more transparent. For instance, in the "Automatic Screening" section, the paper mentions utilizing statistical testing methods to filter the datasets. Can the detailed results of these statistical tests be made publicly available in the appendix?

4. Similar to point 3, around line 251, how exactly is it determined that a series or variate cannot be forecasted? In other works [7], testing across all models is utilized to prove a dataset is unusable. Are there explicit, quantitative criteria here for excluding unsuitable datasets? Another minor detail: "Series-level" is conceptually a higher hierarchy than "Variate-level", therefore the word "entire" should logically modify "series-level" rather than "variates".

[7] Liu, Qinghua, and John Paparrizos. "The elephant in the room: Towards a reliable time-series anomaly detection benchmark." Advances in Neural Information Processing Systems 37 (2024): 108231-108261.

5. In the "Structural tsfeatures for Pattern Definition" section, there is a critical flaw in how the strength of autocorrelation is distinguished. Autocorrelation inherently has a direction; therefore, both strong positive and strong negative correlations should be classified as subsets with strong autocorrelation. As shown in Figure 4, the residual ACF for a massive number of series is concentrated around 0, which aligns with intuition. However, the current binarization method based strictly on the median introduces a severe logical issue, which could significantly compromise the validity of the benchmark's conclusions.

6. The authors utilized Seasonal Naive (S-Naive) as the baseline for normalizing evaluation metrics. What would the results look like if a standard Naive model [8] was used as the baseline instead?

[8] https://cbergmeir.com/talks/neurips2024/

7. In Appendix F, several case studies are visualized to demonstrate specific forecasting behaviors. However, the image captions lack the necessary descriptive context, making it difficult for readers to readily understand the specific phenomena the authors intend to illustrate.

This paper demonstrates immense potential, stemming from the authors' profound insights and their courage to confront existing systemic issues in the field head-on. However, it still contains the several shortcomings mentioned above. If the authors can adequately address these concerns, I believe this will become an exceptionally valuable piece of work for the community.

---

> ### Author Rebuttal · Authors · 2026-03-31
>
> We sincerely thank the reviewer for recognizing the potential value of our work and for the constructive suggestions. Below, we provide our responses to the raised weaknesses and suggestions.
>
> **1. Deep learning (DL)-based models**
>
> We are encouraged that the reviewer found our insight inspiring.  During the rebuttal phase, we ran experiments using five DL models: **PatchTST, DLinear, iTransformer, SparseTSF and TimeMixer** (DUET is underway). Following Gift-Eval's protocol, each model was equipped with a probabilistic head and tuned via 15 trials over a **task-shared hyperparameter (HP) space** before training with the best configuration.
>
> As shown in Fig 7 at https://anonymous.4open.science/r/icml26-rebuttal-4B18/, DL methods overall underperform TSFMs under this setup**, with some even falling behind S-Naïve. However, we emphasize that **these results do not imply DL models are inherently inferior**. Fig 8 shows that DL methods perform competitively on individual datasets. Their poor overall average is driven by **severe collapses on specific datasets** (Fig 9). We attribute these failures to the models' high sensitivity to HP configurations: a shared and constrained HP space inevitably cannot accommodate the diverse dynamics across all tasks, and maximizing DL performance requires exhaustive, dataset-specific tuning.
>
> Crucially, rather than drawing definitive conclusions about DL model capacity, these results illustrate the **immense difficulty of reliably evaluating train-from-scratch DL methods at scale.** A bottleneck is the necessity of dedicated HP searching for every individual task. In contrast, these results highlight the core advantage of TSFMs: delivering robust, zero-shot predictions that bypass massive computational overhead.
>
> **2. Potential bias introduced by non-overlapping forecasting**
>
> While we agree that differences in absolute metric values exist between overlapping and non-overlapping evaluation, we clarify that **current framework is fair and its relative rankings are reliable.
>
> As the reviewer rightly notes, this is a matter of computational tractability. At the scale of TIME (~100 forecasting tasks), stride-1 would increase the number of evaluation instances by orders of magnitude, rendering comprehensive benchmarking infeasible. To ensure fairness, all TSFMs are tested on the exact same non-overlapping windows, subjecting every model to identical positional difficulties. Furthermore, because TIME aggregates results across a massive number of series and tasks, the law of large numbers marginalizes out per-window positional variance.
>
> In fact, non-overlapping rolling evaluation is the standard adopted by recent TSFM benchmarks, including Gift-Eval and fev-bench, and is used in the official evaluation pipelines of most TSFMs.
>
> **3. Transparent data cleaning process**
>
> To ensure transparency, we will release our code alongside the specific LLM prompts, and include the detailed results in Appendix.
>
> **4. Criteria to forecastability**
>
> We thank the reviewer for the reference and agree to correct the word usage. Regarding data filtering, while testing across all baseline models is effective for anomaly detection, adopting this in a zero-shot forecasting benchmark introduces severe circular evaluation bias by inadvertently tailoring the dataset to favor existing model architectures. Furthermore, rigorously quantifying general forecastability is inherently difficult. To our knowledge, **no universal metric currently exists**, and formulating one constitutes a highly challenging independent research direction.
>
> Therefore, our pipeline (Algorithm 2) conservatively eliminates only the sequences that lack fundamental forecasting validity based on **rule-based criteria**. Specifically, we flag time series that are too short or contain excessive missing values, alongside applying statistical tests to remove trivial constant sequences, pure white noise, and series dominated by extreme outliers.
>
> **5. Binarization of ACF-1**
>
> We sincerely thank the reviewer for catching this critical flaw. We have corrected our pipeline to binarize based on the **absolute value of the residual ACF-1**. This ensures that both strong positive and strong negative correlations are accurately categorized as high autocorrelation. We have re-computed the affected pattern groupings and will update all corresponding results in Table 5.
>
> **6. Normalize with Naive**
>
> We evaluated the Naive model on TIME, observing an overall S-Naive to Naive performance ratio of 1:1.239 for MASE and 1:1.552 for CRPS. When normalizing with Naive, the overall relative rankings of the evaluated TSFMs remain unchanged. At the task level, while Naive can outperform S-Naive on specific datasets (e.g., CPHL, Port Activity), the relative model rankings are unaffected as well.
>
> **7. Image caption**
>
> We appreciate the suggestion and will include detailed descriptions in the revised version.

---

> > ### Author Rebuttal · Reviewer_Nyob · 2026-04-03
> >
> > I have decided to increase my score, as the intrinsic merit of this work is significant and the authors' rebuttal has addressed the vast majority of my concerns.
> >
> > However, I encourage the authors to further refine the manuscript to ensure a more comprehensive analysis, thereby delivering deeper insights and value to the time-series forecasting community.
> >
> > Specifically, I remain concerned about evaluating 'from-scratch' deep learning models under a non-overlapping forecasting framework. Such a framework significantly reduces the effective size of the dataset, which might be inherently unfair to models without pre-training as they may not fully exploit the available data for training. Furthermore, does the phenomenon pointed out by the authors suggest that research into deep learning models for this task is not making genuine progress? As it appears that more novel methodologies have not achieved superior performance. Addressing these broader methodological implications would further strengthen the impact of this work.

---

> > > ### Author Response · Authors · 2026-04-06
> > >
> > > We sincerely appreciate your reply and your recognition of our work. We are glad to hear that the vast majority of your concerns have been addressed. For sure, we will further refine the manuscript to ensure a more comprehensive analysis. We would like to take this opportunity to further discuss your follow-up questions:
> > >
> > > ---
> > > ### 1. Clarification on 'from-scratch' models & non-overlapping framework
> > >
> > > We want to clarify a critical detail regarding our experimental design: the **non-overlapping forecasting protocol is only applied to the testing phase, not to training**. Our benchmark enforces a fixed set of non-overlapping test windows strictly to ensure a uniform and strictly fair evaluation across all models.
> > >
> > > We do **not** mandate non-overlapping windows during the training phase. In our rebuttal implementations (following the Gift-Eval protocol), we avoid formulating a massive, static offline training set via a stride-1 sliding window. Instead, our data sampler **randomly selects windows** from all valid positions within each time series, formulating mini-batch and epoch on the fly. This explicitly allows training windows to **overlap and repeat**, generating an infinite cyclic data stream. To manage this continuous stream, we define **each epoch as a fixed number of updates** (e.g., 100 mini-batches). We continuously train the model and monitor performance, applying early stopping as soon as the validation loss ceases to decrease.
> > >
> > > We fully agree with your insight that *exposing a model to every possible training window maximizes its predictive potential*. We also concluded similar conclusions from our previous experience on training from-scratch models or finetuning TSFMs. While this deterministic stride-1 protocol is **feasible on a smaller scale** (such as the 5-6 datasets used in the LSF benchmark), we want to highlight the **extreme difficulty** of applying it across a **massive benchmark**. Pre-extracting all possible windows generates an **enormous volume** of training samples per dataset. To safely process this massive amount of data without **overfitting**, one must utilize an extremely **small learning rate**. Consequently, the model's loss converges very slowly, which drastically increases the overall training time. This makes the exhaustive stride-1 approach **computationally prohibitive when evaluating extensive from-scratch models at scale**.
> > >
> > > Therefore, our random sampling protocol serves as a necessary trade-off. To briefly summarize:
> > >
> > > * **Is the exhaustive stride-1 protocol helpful?** Yes. It maximizes data utilization and pushes models to their theoretical limits. However, the immense computational cost makes it fundamentally infeasible for benchmarking extensive models on large-scale benchmarks.
> > >
> > > * **Is our random sampling protocol fair?** Yes. While it may not extract the absolute peak performance for every individual model, it guarantees an equal sampling probability across all valid positions. Considering the strict balance between performance and training efficiency, it provides a fair and scalable evaluation framework.
> > >
> > > Ultimately, this necessary trade-off highlights a broader reality in current time series research: **evaluating the absolute best performance of from-scratch deep learning models at scale remains an extremely challenging problem**.
> > >
> > > ---
> > > ### 2. Does the rebuttal results suggest that DL models are not making genuine progress?
> > >
> > > **No**. Our results do **not** imply a lack of intrinsic capability or stagnation in DL research. Rather, they highlight the profound **difficulties of training from-scratch models across massive, diverse and newly curated datasets**.
> > >
> > > Beyond the training protocol trade-offs, a primary bottleneck is **hyperparameter (HP) optimization**. In established benchmarks like LSF, the community has identified optimal HP search spaces over time, allowing models to easily achieve strong performance. However, for our novel datasets, these default HP spaces are often **sub-optimal**. As shown in **Fig 8-9** in our anonymous URL, DL models still excel on datasets where default HPs align well, but drop significantly on datasets demanding tailored, domain-specific HP tuning.
> > >
> > > Another critical factor is potential **distribution shift** between training and testing sets. TSFMs demonstrate robustness to such shifts out-of-the-box. In contrast, DL models are inherently sensitive to these shifts because their learned representations are strictly bounded by the limited training split they observe.
> > >
> > > Therefore, our results do **not prove that DL methods are failing**. Instead, they empirically underscore a core advantage of TSFMs: **their out-of-the-box forecasting ability empowers superior robustness to HP variations and distribution shifts**. To provide a fairer conclusion, **we will conduct deeper analyses**, including **finer-grained HP searches**, on the specific datasets where DL models underperformed.

---

### Official Review · Reviewer_oRye · 2026-03-08

**Soundness:** 2
**Presentation:** 1
**Significance:** 3
**Originality:** 2
**Overall Recommendation:** 4
**Confidence:** 4

**Summary:**

The paper proposes a new benchmark for time series forecasting, together with an approach to characterize each forecasting problem based on the characteristics of the time series to be forecast. This allows for providing an in-depth analysis of empirical results.

**Compliance With Llm Reviewing Policy:**

Affirmed.

**Ethical Review Concerns:**

As the paper redistributes existing datasets and makes available new data, I believe it is necessary to ensure that all of this has been done in accordance with existing rules and guidelines.

**Ethical Review Flag:**

Flag this paper for an ethics review.

**Ethics Expertise Needed:**

["Legal Compliance (e.g., EU AI Act, GDPR, copyright, terms of use)"]

**Final Justification:**

I believe methodological contributions are relatively limited, and I have some concerns about the presentation and discussion of some aspects (e.g., use of LLMs) that could not be fully resolved in the scope of the rebuttal.

However, the contributions of the paper towards more interesting benchmarks for time series forecasting likely outweigh these negative aspects. I have increased my score to a weak accept.

**Key Questions For Authors:**

1. Please clarify how you use LLMs to filter the time series.
2. As you mention that some of the datasets were obtained from the private sector, did you obtained permession and a licence to redistribute the datasets?

**Limitations:**

I would encourage the author to discuss more on the limitations of the proposed feature-based task categorization. As an example, the Dickey–Fuller test used to detect non-stationarity operates under very precise assumptions. In particular, the discussion could be expanded to limitations in capturing nonlinear patterns.

**Strengths And Weaknesses:**

**Strengths**

* Providing more insights and fine-grained details on the data within forecasting benchmarks is an important direction for future research on the topic.
* Increasing the variety of benchmarks in ML for time series forecasting is something that the field needs.

**Major Weaknesses and discussion**

- **Presentation** Through the paper, the writing is wordy and makes ample use of terms with no clear meaning in scientific writing. As a result, it is often difficult to assess the scientific significance of the paper and of the methods being discussed. (See examples at the bottom of this block). By looking at the examples, the precise meaning and implications of all these sentences can only be guessed. These are just a few examples, but the same writing style is followed throughout the paper. Moreover, vague terms are routinely used to describe the approaches used in the papers (e.g., "rigorous" or "rigourously"), but I'd remove such terms and focus on explaining how such operations are done and why doing them differently would be problematic. In other words, saying that something was done "rigorously" is not meaningful on its own. I urge the authors to rewrite the paper, focusing on the content and making the language simpler and more direct.

Beyond presentation issues, since the paper proposes both a method to build benchmarks and a benchmark itself, I will split the following discussion into two parts. The first one focuses on the scientific merits of the proposed approach, and the second on the relevance of the introduced benchmark.

* **Relevace of the benchmark construction method** (Section 4 and 5). The method used to filter the data appears reasonable. However, many of the steps described in Section 4 involved subjective decisions from human curators. Moreover, the paper provides only a high-level description of some of the filtering steps (e.g., using LLMs to check flagged time series). The "Algoritms" reported in the appendix improve this aspect only partially. This makes this part of the paper a useful discussion on how the dataset was built, but the procedure cannot be reproduced or used to generate more benchmarks. As such, the scientific value of the methodology cannot be fully evaluated and cannot be considered a contribution of the paper. Conversely, the procedure described in Section 5 is easier to transfer and reproduce. However, it relies on well-established time series analysis tools that are routinely used to characterize time series, so the technical novelty is limited. Nonetheless, systematically including these features in analyzing forecasting results is indeed a good idea.

* **Relevance of the datasets** Providing access to a new collection of varied curated datasets is indeed a good contribution, though the number of time series in each dataset is often limited. Nonetheless, this is the strongest part of the paper and a good contribution. Can you clarify how many of these datasets were already available and how many were specifically sourced for building the benchmark?

Overall, I believe the introduced dataset could be a useful resource for ML research on time series forecasting, and I appreciate the focus on providing more detailed performance analysis. Nonetheless, the technical contributions of the paper are quite limited, and the presentation has severe issues. As such, I am leaning toward rejecting the paper in its current form.


Other comments:

* Empirical results should include error bars.

Examples of problems in the presentation:

  * "Rigid, mechanical setups detach forecasting requirements from real-world contexts, ignoring predictable horizons and variate predictability." What is a rigid mechanical setup?
  * "time-series error metrics are inherently abstract and do not directly reflect the structural fidelity"
  * "TIME enables effective pattern-based stratification and retrieval, yielding generalizable and diagnostic insights into model performance."
  * "We posit that data frequency and task settings are intrinsically subservient to the application context."
  * "[we] employ value dominance to identify uninformative
series that exhibit negligible temporal dynamics."

---

> ### Author Rebuttal · Authors · 2026-03-31
>
> We thank the reviewer for the in-depth suggestions and valuable feedback. We provide our responses to the specific weaknesses (W), questions (Q) and Limitation (L).
>
> **W1: Presentation issues**
>
> We sincerely thank the reviewer to provide this responsible and constructive feedback. We value your suggestions and believe they will significantly improve the clarity and scientific transparency of the presentation.
>
> We have conducted a **comprehensive rewrite of the paper**. We refined our language by substituting wordy phrasing with precise scientific descriptions. We also removed subjective and vague terminology, ensuring that our claims clearly describe how our procedures work and why they are implemented.
>
> Due to space constraints, we only provide our revisions to the listed examples below:
> * Classic benchmarks often use fixed prediction lengths, such as 720 steps, across diverse datasets. Such "one-size-fit-all" setups detach forecasting requirements from real-world contexts, ignoring that the valid forecasting horizon depends on the specific application scenario and data frequency.
> * Common time-series error metrics, such as MSE, only provide a scalar measurement of prediction error. A numerical improvement in these metrics does not guarantee that the model accurately captures temporal behaviors or that its predictions are reliable for actual deployment.
> * TIME can group variates based on specific time series patterns. This allows us to systematically benchmark TSFMs across these pattern groups, revealing which models are most suitable for specific data characteristics and providing concrete guidelines for model selection.
> * We argue that data frequency and horizon length are conditional upon the requirements of the specific forecasting application.
> * we filter out uninformative variates by two thresholds: a series is discarded if its top-5 value concentration exceeds 0.5 or its normalized entropy falls below 0.1.
>
> **W2 & Q2: Methodology Issues**
>
> * Benchmark Construction (Sec. 4)
>
> We are glad that the reviewer finds our filtering methodology reasonable. While curating large-scale time series data inevitably requires some subjective decisions, we are fully committed to making this process transparent. First, we will include the **specific details of our LLM usage** (see prompts in Fig1-3 at https://anonymous.4open.science/r/icml26-rebuttal-4B18/) in the revised manuscript. Furthermore, we will **open-source the data processing codebase** to ensure our pipeline can be fully reproduced and used to generate future benchmarks.
>
> * Benchmark Strategy (Sec. 5)
>
> We are encouraged that the reviewer finds this part easy to transfer and reproduce, and benchmarking based on features is a good idea. Regarding technical novelty, our core contribution lies in **providing a novel pattern-level evaluation perspective**, rather than designing new mathematical metrics to characterize time series. As a benchmark paper, we intentionally **leverage established metrics to ensure standardized and insightful comparisons** across diverse data.
>
> **W3 & Q2: Dataset source and license**
>
> We thank the reviewer for recognizing our dataset collection as a strong contribution. To clarify the sources of the 50 datasets included in TIME: **9 datasets were previously available**. The remaining **41 datasets are newly processed and curated by us**.
>
> Regarding data redistribution, we confirm that **all datasets have necessary permissions and appropriate open-source licenses to redistribute**.
>
> **W4. Empirical results should include error bars**
>
> We clarify that our benchmark focuses on zero-shot forecasting. All evaluated TSFMs are initialized from their pre-trained weights without any training. Consequently, the impact of random seeds on the final performance is minimal.
>
> To validate this, we selected **6 TSFMs and re-ran the evaluation across 3 different random seeds**. The overall results with error bars are shown below:
>
> |Model|MASE (norm.)|CRPS (norm.)|
> |:---|:---|:---|
> |Chronos-2 *|0.6603±0.0000|0.5486±0.0000|
> |TimesFM-2.5 *|0.6667±0.0000|0.5594±0.0000|
> |TiRex *|0.6812±0.0000|0.5650±0.0000|
> |Toto †|0.6957±0.0023|0.5754±0.0025|
> |Moirai2 *|0.7011±0.0000|0.5804 ± 0.0000|
> |Sundial †|0.7555±0.0079|0.6535 ± 0.0075|
>
> *( * denotes Quantile-based; † denotes Distribution-based.)*
>
> The quantile-based models yield deterministic forecasts with a variance of 0 while the distribution-based models exhibit slight variance due to the sampling mechanisms. This proves that **our zero-shot evaluation results are robust and stable**. We will include multi-seed results in the appendix.
>
> **L1: Limitation on feature-based categorization**
>
> Thanks for the suggestion. We agree that different features have its inductive bias or assumption on data. **We will expand the Limitations section** to discuss the boundaries of feature-based categorization, including its involved assumptions and difficulty in capturing nonlinear patterns.

---

> > ### Author Rebuttal · Reviewer_oRye · 2026-04-03
> >
> > Thank you for the rebuttal. I really appreciate your willingness to improve reproducibility and the presentation.
> >
> > Most of my concerns have been reasonably addressed. However, assessing whether the revision will fully address the discussed presentation issues is not possible from the rebuttal. Moreover, the soundness of relying on LLMs to filter out data and make domain-specific decisions deserves further discussion.
> >
> > Finally, while I appreciate the contribution of the paper toward increasing dataset heterogeneity and availability, I still believe the methodological contributions are quite limited.
> >
> > I will consider increasing my score, and I would not mind if the paper is accepted if the other reviewers agree that its contributions outweigh its weaknesses.

---

> > > ### Author Response · Authors · 2026-04-06
> > >
> > > We sincerely appreciate your constructive feedback and fully understand your remaining concerns. We **commit to comprehensively refining the presentation** to ensure maximum clarity in the final manuscript.
> > >
> > > Regarding **methodological contributions**, we respectfully acknowledge that our work does **not** propose a new model architecture or algorithmic design like a traditional methodology paper. However, the methodological contribution of a **benchmarking paper** fundamentally lies in establishing innovative paradigms. In this regard, beyond the newly curated datasets, we introduce a novel, **variate-level pattern-based evaluation perspective** for time series forecasting, which, to the best of our knowledge, has never been systematically explored in prior time-series research.
> > >
> > > We deeply respect your rigorous scrutiny concerning the **soundness of utilizing LLMs for data processing**. We provide a detailed discussion on this critical topic below:
> > >
> > >  ---
> > > **1. LLM assists human decision-making**
> > >
> > > We want to clarify that the **LLM does not make final decisions or filter data directly**. It provides **semantic recommendations**, while human reviewers make the final decisions. The LLM never interacts with or filters raw numerical data. As detailed in Section 2.2, the initial data quality screening on numerical time series is handled entirely by a deterministic algorithmic pipeline. The LLM only processes the resulting **Screening Reports** to offer domain-specific insights (Fig 2 in https://anonymous.4open.science/r/icml26-rebuttal-4B18/). Ultimately, the LLM serves purely as an **advisor**, and human reviewers execute the final actions after **verifying the LLM's provided rationale**.
> > >
> > > ---
> > > **2. Is such a process subjective?**
> > >
> > > Yes. However, we argue that **purely human decision-making is also inherently subjective**. In an ideal scenario, achieving true objectivity would require a panel of **multiple domain experts** to comprehensively guide and cross-validate the entire lifecycle of every single dataset. However, the human labor and communication costs required for this are prohibitively high, which severely hinders the construction of large-scale benchmarks. To overcome this bottleneck, the LLM serves as a highly scalable solution. It acts as an efficient **proxy for domain experts**, leveraging its vast prior knowledge to provide context-aware insights.
> > >
> > > ---
> > > **3. High Human-LLM Alignment**
> > >
> > > To **empirically validate** the soundness of the LLM as an expert proxy, we conducted an alignment experiment. We selected 13 datasets (comprising 33 tasks) where we have direct connection to domain experts. We then compared the LLM-generated recommendations (data filtering & horizon selection) against the decisions made independently by these human experts. The results demonstrated a **high Human-LLM alignment**, achieving a **93.9%** agreement rate (31 out of 33 tasks). This confirms that the LLM is a reliable semantic advisor specifically for our benchmark's data screening and task formulation pipeline.
> > >
> > > ---
> > > **4. Robustness across LLMs**
> > >
> > > To evaluate the stability of our LLM-assisted pipeline, we tested it across different top-tier LLMs. Specifically, we employed **Claude Opus 4.5** to perform the horizon selection task on 20 diverse datasets (comprising 60 tasks). We then compared its outputs with the results generated by Gemini 3 Pro (the model used in our original manuscript). The two models demonstrated high alignment, achieving a **93.3% consistency rate** (examples are shown in **Fig. 10-11** in https://anonymous.4open.science/r/icml26-rebuttal-4B18/). This empirically proves that high-performance LLMs can provide robust and consistent recommendations.
> > >
> > > ---
> > >
> > > Finally, we agree that determining how to systematically ensure the soundness of LLMs is a **critical, emerging challenge** across the AI community.  Recently, the time-series community has successfully begun utilizing LLMs for complex domain reasoning and QA tasks [1,2], demonstrating strong semantic capabilities. To rigorously verify the soundness of such LLM applications, an ideal methodology would involve adopting emerging automated evaluation frameworks like *Agent-as-a-Judge* [3]. However, comprehensively building such an advanced verification framework represents a research direction that falls out of the scope of our current benchmarking focus. Therefore, we rely on our empirical human-in-the-loop safeguards for this work, and we will explicitly discuss this LLM soundness topic in the **Limitations and Future Work** section of our revised manuscript.
> > >
> > > [1] TimeOmni-1: Incentivizing Complex Reasoning with Time Series in Large Language Models, ICLR 2026
> > >
> > > [2] Time-MQA: Time Series Multi-Task Question Answering with Context Enhancement, ACL 2025
> > >
> > > [3] A Survey on Agent-as-a-Judge, arxiv 2026

---

### Official Review · Reviewer_irvg · 2026-03-11

**Soundness:** 3
**Presentation:** 2
**Significance:** 3
**Originality:** 2
**Overall Recommendation:** 4
**Confidence:** 4

**Summary:**

This paper introduces a task-centric benchmark for evaluating time series foundation models, comprising 50 fresh datasets and 98 forecasting tasks designed to eliminate data leakage risks inherent in existing benchmarks. The work addresses several limitations of existing benchmarks: constrained data coverage, compromised data integrity, misaligned task formulations, and rigid analysis perspectives. A human-in-the-loop construction pipeline integrates automated quality screening with LLM-assisted expert judgment to ensure data integrity and align forecasting configurations with real-world operational requirements. The paper's key methodological contribution is a pattern-level evaluation framework that extracts structural time series features via STL decomposition, encodes them as binary pattern codes, and retrieves matching variates across datasets for cross-domain performance analysis. Evaluation of 12 TSFMs reveals that model advantages are most pronounced on non-stationary series and those with unstable seasonality, while high complexity compresses inter-model performance differences.

**Compliance With Llm Reviewing Policy:**

Affirmed.

**Final Justification:**

The authors' response addresses most of my concerns. Overall, my initial score is relatively positive, and I maintain a positive view of the paper.

**Key Questions For Authors:**

At line 150, the authors mentioned that exogenous covariates are excluded entirely, and all variates are treated as prediction targets. In my understanding, multivariate dependency modeling is therefore largely ignored, any reason why doing this?

**Limitations:**

Yes.

**Strengths And Weaknesses:**

**Strengths**

1. The problem is critical: legacy dataset contamination and misaligned task formulations risk rendering TSFM benchmarks unreliable, while plateauing performance obscures whether gains are genuine.

2. The insight that competitive MASE scores can coexist with structurally poor predictions (flat-line forecasts) is an interesting finding that could inspire development of structure-aware evaluation metrics.

3. The experiment design is compelling: 12 architecturally diverse TSFMs evaluated across multi-granular views, with the pattern-specific analysis effectively revealing model-specific strengths invisible in aggregate rankings, while newer models consistently outperforming predecessors validates benchmark rationality.

** Weaknesses**

1. The core data quality screening steps (e.g. outlier detection via IQR, white noise testing via Ljung-Box, correlation checking) are standard preprocessing techniques rather than methodological innovations.

2. The STL decomposition assumes additive structure and a single dominant seasonal period, which fails for multiplicative dynamics or series with multiple seasonalities common in many domains.

3. At line 320, the author mentioned that the binary encoding scheme is employed. However, the 7 binary features yield 128 possible pattern codes, but the paper does not report the actual population distribution across codes. Many combinations likely have insufficient variates for statistically reliable aggregation.

---

> ### Author Rebuttal · Authors · 2026-03-31
>
> We sincerely thank the reviewer for the valuable feedback and positive evaluation of our work. We provide detailed responses to the weaknesses (W) and questions (Q) raised by the reviewer.
>
> **W1: Data quality screening steps are standard preprocessing techniques**
>
> We agree with the reviewer that the individual screening steps are standard techniques. However, our core contribution is the **systematic, automated pipeline architecture**, rather than the invention of new standalone mathematical tests.
>
> While these techniques are common in isolation, no existing time series benchmark has successfully formulated them into a comprehensive, end-to-end automated screening workflow. This historical lack of systematic curation is exactly why many current benchmarks suffer from the data quality issues we highlighted in the paper.
>
> Our methodological innovation lies in the **system design and integration**, transforming fragmented standard steps into a robust, scalable pipeline that establishes a rigorous new standard for large-scale benchmark data curation.
>
>
> **W2: The STL decomposition fails for multiplicative dynamics or series with multiple seasonalities**
>
> We acknowledge the reviewer's point regarding single-period STL. Our initial approach followed standard practices in established libraries like tsfeatures: https://github.com/Nixtla/tsfeatures/blob/main/tsfeatures/tsfeatures.py#L681
>
> To rigorously address your concern, we re-implemented our feature extraction using MSTL (handling up to 3 seasonal periods detected via FFT). The average per-pattern reclassification rate is minor (11.6%), primarily affecting seasonal strength (22.4%) and residual ACF1 (25.8%). Crucially,  **the final model rankings remain exceptionally stable.** Across selected pattern filter combinations (14 single-pattern conditions derived from 7 features each set to =0 and =1, plus 8 representative dual-pattern combinations), the Spearman rank correlation between STL- and MSTL-based leaderboards averages 0.996, maintaining 100% agreement for the top-1 and top-5 models. This confirms **our benchmark conclusions are highly robust to the decomposition method**.
>
> Regarding multiplicative dynamics, while applying a log-transformation is a potential solution, altering the data scale in a broad benchmark compromises the evaluation of models designed for raw values. Therefore, we maintain the raw scale to ensure fair evaluation. **We will incorporate this MSTL robustness analysis into Appendix and discuss the boundary of additive decomposition in the Limitations section.**
>
> **W3: population distribution across possible pattern codes**
>
> We appreciate the reviewer’s rigorous observation. Providing 7 binary features was intended to offer a comprehensive taxonomy for diverse analytical angles, rather than mandating a rigid 128-bin grouping mechanism for all scenarios. In practice, to ensure statistical reliability, **we strongly recommend that users filter the per-pattern leaderboard using no more than 5 features** tailored to their specific analytical goals.
>
> Regarding conditioning on all 7 features, we agree it creates a highly granular space, leading to sparsity in certain codes. We have analyzed the empirical population distribution across all 128 codes. The results reveal a natural **long-tail distribution**, which shows properties below:
>
> * The top 33 pattern combinations account for **over 80%** of the entire benchmark dataset, ensuring highly robust sample sizes and statistical reliability for these dominant categories.
>
> * 18 theoretical combinations contain **zero variates**. This naturally occurs because certain mathematical properties are mutually exclusive or extremely rare in physical reality.
>
> In the revised manuscript, we will emphasize the specific usage guidelines and include the complete population distribution histogram.
>
> **Q1:  Exogenous covariates are excluded thus multivariate dependency modeling is largely ignored.**
>
> We thank the reviewer for raising this point and would like to clarify our setup. While we currently exclude exogenous covariates, our benchmark **explicitly allows TSFMs to model cross-variate dependencies during multivariate forecasting tasks**. We intentionally excluded exogenous covariates (auxiliary inputs that only serve as features and do not require prediction) in this initial version for two practical reasons:
>
> * Definitively separating covariates from target variables across massive, multi-domain datasets requires **extensive domain-specific expertise**, making rigorous, large-scale automated curation highly challenging.
>
> * Native architectural support for exogenous covariates remains **relatively sparse among current zero-shot TSFMs**.
>
> We agree that covariate-assisted forecasting holds immense practical value. **We view this as a crucial next step for the field and will list the integration of exogenous covariate as a future direction.**

---

> > ### Author Rebuttal · Reviewer_irvg · 2026-04-03
> >
> > I thank the authors for the efforts in rebuttal. I will maintain my positive score.

---

> > > ### Author Response · Authors · 2026-04-04
> > >
> > > We sincerely thank you for evaluating our work and highlighting its strengths. Your insights are highly valuable to us, and we are glad to know that your concerns have been fully resolved. We will thoroughly update our manuscript to incorporate your constructive points.

---

### Official Review · Reviewer_EG45 · 2026-03-11

**Soundness:** 3
**Presentation:** 4
**Significance:** 3
**Originality:** 3
**Overall Recommendation:** 5
**Confidence:** 4

**Summary:**

This work examines several systemic issues in existing benchmarks for time series forecasting (TSF), particularly in the evaluation of time series foundation models. The authors argue that current benchmarks suffer from several key limitations, including the reliance on legacy datasets, data quality concerns, unrealistic task configurations, and coarse-grained evaluation protocols.

To address these issues, the paper introduces a new benchmark consisting of 50 datasets and 98 forecasting tasks, designed for strict zero-shot evaluation in order to reduce potential contamination between training and testing data. The authors further propose a human–LLM collaborative pipeline for benchmark construction, which includes data collection, data quality assessment, expert review, LLM-assisted analysis, and task configuration. In addition, the benchmark evaluates models at the pattern level, grouping time series according to their structural characteristics. The benchmark includes evaluations of 12 time series foundation models.

**Compliance With Llm Reviewing Policy:**

Affirmed.

**Final Justification:**

My concerns have been answered, so I've changed some of the scores.

**Key Questions For Authors:**

1 I found parts of Weaknesses 1 and 2 somewhat confusing during my reading and would appreciate further clarification from the authors.

2 Have the authors considered issues related to the long-term maintenance and extension of the benchmark?

**Limitations:**

Regarding Weaknesses 3–5, I believe there is still room for improvement in future versions of the benchmark.

**Strengths And Weaknesses:**

Strengths

1 The paper focuses on the increasingly important problem of time series forecasting and identifies several genuine issues in existing TSF benchmarks. By proposing a new benchmark, the work provides a potentially valuable resource contribution to the community.

2 The idea of pattern-level evaluation is interesting and potentially meaningful. Evaluating models based on temporal patterns may provide a more informative comparison of model performance across different types of time series.

3 The inclusion of both human experts and LLMs in the benchmark construction pipeline is a thoughtful design choice and may improve the reliability of the data curation and task configuration process.

Weaknesses

1 The use of STL decomposition together with handcrafted features would benefit from additional theoretical justification. In particular, the feature selection process does not appear to be fully systematic, and it is unclear why the chosen features are sufficient, complete, and unbiased. In addition, the binary encoding used for pattern representation may cause different time series to be mapped to the same pattern. The definition of patterns could be clarified further, as this part of the methodology is somewhat difficult to interpret.

2 The paper does not provide sufficient details about the LLM used in the LLM-assisted component of the pipeline. More information about the specific model, prompts, and settings would help improve reproducibility.

3 The benchmark already includes a large number of foundation models. However, including additional recent models such as MOMENT [1] and Time-MoE [2] could further strengthen the evaluation.

4 It is unclear whether the benchmark configuration includes sensitivity analysis. For example, it would be useful to examine whether different forecasting horizons or context lengths would lead to changes in the relative ranking of models.

5 While the benchmark focuses on foundation models, including a few strong traditional TSF baselines could make the comparison more comprehensive. For example, models such as TimeMixer++ [3], PatchTST [4], or iTransformer [5] could provide strong competitive baselines. In addition, lightweight models such as DLinear [6] or SparseTSF [7] could also serve as useful references. If the goal of the benchmark is to provide a comprehensive evaluation, expanding the coverage of models may further improve the study.

6 The experimental results table lacks bolding and underlining, making it difficult to distinguish between the best and second-best models.

[1] Goswami M, Szafer K, Choudhry A, et al. MOMENT: A Family of Open Time-series Foundation Models[C]//Forty-first International Conference on Machine Learning.

[2] Shi X, Wang S, Nie Y, et al. Time-MoE: Billion-Scale Time Series Foundation Models with Mixture of Experts[C]//The Thirteenth International Conference on Learning Representations.

[3] Wang S, Jiawei L I, Shi X, et al. TimeMixer++: A General Time Series Pattern Machine for Universal Predictive Analysis[C]//The Thirteenth International Conference on Learning Representations.

[4] Nie Y, Nguyen N H, Sinthong P, et al. A Time Series is Worth 64 Words: Long-term Forecasting with Transformers[C]//The Eleventh International Conference on Learning Representations.

[5] Liu Y, Hu T, Zhang H, et al. iTransformer: Inverted Transformers Are Effective for Time Series Forecasting[C]//The Twelfth International Conference on Learning Representations.

[6] Zeng A, Chen M, Zhang L, et al. Are transformers effective for time series forecasting?[C]//Proceedings of the AAAI conference on artificial intelligence. 2023, 37(9): 11121-11128.

[7] Lin S, Lin W, Wu W, et al. SparseTSF: Modeling Long-term Time Series Forecasting with* 1k* Parameters[C]//Forty-first International Conference on Machine Learning.

---

> ### Author Rebuttal · Authors · 2026-03-31
>
> We thank the reviewer for the recognition and constructive feedback. We now provide our responses to the specific weaknesses (W) and questions (Q).
>
> **W1&Q1: Further clarification on patterns**
>
> We want to clarify that defining a sufficient, complete, and unbiased feature representation for time series is a challenging open problem. As discussed in Sec. 2.2 and App. D.1, existing time series characterization predominantly relies on empirically validated handcrafted features rather than strict theoretical guarantees. We empirically validate our selection in App. D.3, showing that our curated features possess strong discriminative power and low-to-moderate correlations.
>
> Our objective is not complete or lossless representation, but rather practical and interpretable benchmarking. Thus, in our framework, *a "pattern" is defined as a macro-level temporal characterization of a time series, represented by a set of curated features*. We acknowledge that binary encoding is a relatively coarse-grained abstraction. As discussed in App. G, this is an engineering trade-off designed to ensure robust evaluation. Implementing finer-grained, multi-level binning would lead to a combinatorial explosion of patterns, resulting in highly sparse bins with too few variates to support reliable benchmarking insights. However, applying **multi-level binning to individual features** remains feasible, and we will incorporate this direction into future versions of TIME.
>
> **W2: Details of LLM usage**
>
> We utilized Gemini 3 Pro (standard web interface) to assist in the following tasks:
>
> * **Dataset Pruning**: Assist the final data pruning step based on the data quality report from the automatic screening phase.
>
> * **Horizon Configuration**: Determining practical and realistic forecasting window lengths tailored to actual real-world deployment scenarios.
>
> We provide an example of data report and complete prompt templates in Fig 1-3 at https://anonymous.4open.science/r/icml26-rebuttal-4B18/. We will include these details in the revised version.
>
> **W3: Inclusion of additional TSFMs (MOMENT and Time-MoE)**
>
> We appreciate the reviewer's suggestion. We originally excluded Time-MoE and MOMENT because they are **restricted to point forecasting** and do not support probabilistic forecasting. Also, as noted in the original paper, **MOMENT requires task-specific fine-tuning** for its forecasting head and lacks out-of-the-box zero-shot capabilities.
>
> To comprehensively address your feedback, we evaluated both models on the TIME benchmark for point forecasting. The results are summarized below:
> |Model|MASE (norm.) |MAE (norm.)|MSE (norm.)|
> |:--- |:---|:---|:---|
> | Time-MoE (50m) |0.780|0.775|0.594|
> | Moment (large) |1.347|1.295|1.353|
>
> Time-MoE achieves performance positioned between Sundial and TimesFM 1.0. This outcome is **consistent with the general developmental timeline** of TSFMs. MOMENT demonstrates **severely degraded** performance in a zero-shot setting, underperforming even the S-Naive baseline.
>
> **W4: Sensitivity analysis on horizons or context lengths**
>
> * **Forecasting Horizons:** TIME inherently include short, medium, and long-term horizons. To check the effect of horizon, we filter the evaluation results by specific horizon to generate three horizon-specific leaderboards (Fig 4-6). We found that **top model rankings are consistent while others exhibit variability**.
>
> * **Context Length**: We clarify that **context length is an intrinsic model hyperparameter, not a universal benchmark setting**. Different TSFMs have various context capacity ranges. To evaluate the "out-of-the-box" forecasting ability of TSFMs, we utilize the default context lengths recommended by the official model releases or demos.
> To further address this question, we conducted a **sensitivity analysis under varying context lengths** using 3 top-performing models.
>
> |Model|Context|MASE (norm.)|CRPS (norm.)|
> |:---|:---|:---|:---|
> |Chronos2|1024|0.675|0.563|
> ||2048|0.659|0.550|
> ||4096|**0.657**|**0.547**|
> ||8192|0.660|0.549|
> |TimesFM-2.5|1024|0.686|0.579|
> ||2048|0.674|0.565|
> ||4096|0.667|0.559|
> ||8192|**0.664**|**0.557**|
> |Tirex|512|0.721|0.592|
> ||1024|0.692|0.575|
> ||2048|**0.681**|**0.565**|
>
> *(After submission, we identified and corrected minor implementation issues in the evaluation of Chronos2 and Tirex. The results are updated.)*
>
> These models exhibit varying degrees of sensitivity to context length, but their **recommended default values consistently deliver strong results**. This reflects that a good TSFM can directly utilize its default context length to achieve excellent performance.
>
> **W5: Inclusion of deep-learning-based methods**
>
> Please refer to our response to Q1 of Reviewer *Nyob*.
>
> **W6: Bolding & underlining in tables**
>
>  We will fix this in the revised version.
>
> **Q2: Will TIME be maintained/extended?**
>
> Yes, all evaluation scripts and datasets will be publicly released. We will accept and integrate community submissions of new models and datasets.

---

> > ### Author Rebuttal · Reviewer_EG45 · 2026-04-03
> >
> > Thanks for the rebuttal. My concerns are all solved, and I will adjust my score.

---

> > > ### Author Response · Authors · 2026-04-04
> > >
> > > Thanks for your recognition and careful review. We greatly appreciate the time and effort you invested in providing such constructive feedback. We will revise our manuscripts accordingly.

---

### Decision · Program_Chairs · 2026-04-30

**Decision:**

Accept (regular)

**Comment:**

The reviewers agree that there is clear merit and value to this work: the goal of creating a new forecasting benchmark using fresh data unencumbered by some of the issues the plague legacy datasets is highly relevant, the design choices made by the authors during benchmark construction are largely considered reasonable, and the experimental evaluation is comprehensive and insightful. The reviewers also appreciate the authors' commitment to release the benchmark construction code, improving transparency and the potential for reuse.
The reviewers point out some areas for improvement, that we hope the authors can address for the camera-ready version:
* The reviewers suggest that the paper could be strengthed by making the writing simpler and more direct and avoiding unnecessary fluff. The authors committed to address this during the rebuttal.
* The reviewers see the most of the value in the created dataset and the benchmarking study the authors performed. Perhaps the space allocation in the paper can be shifted to spend more space describing the properties of the dataset and the results of the benchmarking study, while shrinking some of the other sections.
* Please ensure that all data included in the dataset is appropriately licensed, and provide a clear description of the terms under which the benchmark dataset can be used, redistributed, or modified. This is critical for the wider adoption of this benchmark.